# PlotCraft: Pushing the Limits of LLMs for Complex and Interactive Data Visualization

Jiajun Zhang [* 1 2]   Jianke Zhang [* 3]   Zeyu Cui [4]   Jiaxi Yang [5]   Lei Zhang [5]   Zilei Wang [1]   Qiang Liu [2]
Liang Wang [2]   Binyuan Hui [4]   Junyang Lin [4]

## Abstract

Recent Large Language Models (LLMs) have demonstrated remarkable proficiency in code generation. However, their ability to create complex visualizations for scaled and structured data remains largely unevaluated and underdeveloped. To address this gap, we introduce **PlotCraft**, a new benchmark featuring 1k challenging visualization tasks that cover a wide range of topics, such as finance, scientific research, and sociology. The benchmark is structured around seven high-level visualization tasks and encompasses 48 distinct chart types. Crucially, it is the first to systematically evaluate both single-turn generation and multi-turn refinement across a diverse spectrum of task complexities. Our comprehensive evaluation of 23 leading LLMs on PlotCraft reveals obvious performance deficiencies in handling sophisticated visualization tasks. To bridge this performance gap, we construct **SynthVis-30K**, a large-scale, high-quality dataset of complex visualization code synthesized via a collaborative agent framework. Building upon this dataset, we develop **PlotCraftor**, a novel code generation model that achieves strong capabilities in complex data visualization with a remarkably small size. Across VisEval, PandasPlotBench, and our proposed PlotCraft, PlotCraftor shows performance comparable to that of leading proprietary approaches. Especially on hard tasks, our model achieves over 50% performance improvement. We will release the benchmark, dataset, and code at PlotCraft Benchmark.

[1]University of Science and Technology of China, Hefei, China [2]Institute of Automation, Chinese Academy of Sciences, Beijing, China [3]Tsinghua University, Beijing, China [4]Alibaba Group, Hangzhou, China [5]Shenzhen Institute of Advanced Technology, Chinese Academy of Sciences, Shenzhen, China. Correspondence to: Qiang Liu <qiang.liu@nlpr.ia.ac.cn>.

*Proceedings of the $43^{rd}$ International Conference on Machine Learning*, Seoul, South Korea. PMLR 306, 2026. Copyright 2026 by the author(s).

## 1. Introduction

Recent advancements in AI research have demonstrated the powerful code generation capabilities of LLMs (OpenAI, 2023; 2025; Anthropic, 2023; Team, 2024; Rozière et al., 2023; Hui et al., 2024; MistralAI, 2024; Team et al., 2025b; Cao et al., 2026; Team, 2026; Zhang et al., 2026a), which have solved coding challenges in domains such as software engineering (Jimenez et al., 2023; Zhang et al., 2025; Pan et al., 2025b), code completion (Ding et al., 2023; Yang et al., 2024a; Gong et al., 2024), and algorithmic problem-solving (Chen et al., 2021a; Zhuo et al., 2025; Jain et al., 2024). However, in the domain of data visualization, the capabilities of LLM have yet to be fully explored. We observe that while LLMs excel at creating simple, single-panel charts, they struggle to generate plots with multiple subplots and intricate composite layouts from large, complex structured data. As illustrated in Figure 2, when faced with these complex plotting tasks, LLMs often generate code that results in chaotic layouts, overlapping elements, and obscured axis labels, or they completely ignore instructions about certain areas of plots.

Prior works on chart generation have primarily focused on relatively simple, text-to-visualization (Text2Vis) tasks (e.g., VisEval (Chen et al., 2024)/PandasPlotBench (Galimzyanov et al., 2025)) or on understanding and reproducing existing data charts (Chart2Code) (e.g., Plot2Code (Wu et al., 2024), ChartMimic (Yang et al., 2025b)). However, these efforts do not adequately assess the ability of LLMs to synthesize complex visualization code from structured raw data. To systematically evaluate these capabilities of LLMs, we introduce **PlotCraft**, a large-scale benchmark comprising 982 instances. PlotCraft is characterized by its use of instructional prompts of varying difficulties, a diverse range of chart types, and multi-level evaluation metrics. Specifically, we design and collect tasks of varying difficulty based on number of subplots to be generated, chart complexity, data volume, and level of detail in the instructions. Furthermore, we introduce an additional multi-turn refinement task, which allows the model to refine the chart over multiple conversational turns, thereby assessing its ability to debug code and perform iterative optimization based on user feedback.

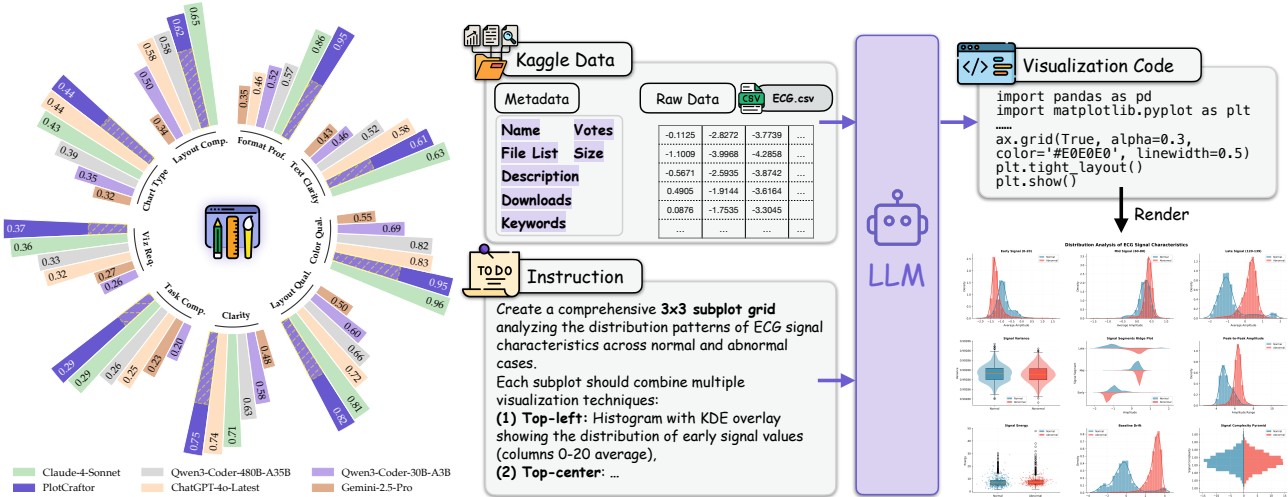

*Figure 1.* An overview of the PlotCraft benchmark and the performance of several leading LLMs and our model, PlotCraftor. **(Left)** A polar bar chart compares the performance of PlotCraftor against five leading baseline models across all of our proposed sub-metrics. The purple area explicitly highlights the performance gains of PlotCraftor relative to its base model, Qwen3-Coder-30B-A3B. **(Right)** An example task from PlotCraft, which requires an LLM to process raw Kaggle data and a complex, human-written instruction to generate visualization code, which is then rendered into the final chart. PlotCraft benchmark comprises 1k high-quality evaluation instances.

We evaluate 23 prominent LLMs on PlotCraft benchmark, including 6 proprietary models and 17 open-weight models. We observe that most models perform well on simple chart generation tasks, but fail on medium and hard tasks, which require processing larger data and generating composite plots with multiple, properly arranged chart types. Results show that existing LLMs have limited capabilities in complex scientific visualization code generation tasks. To further validate our findings, we score a set of results with human evaluation and a correlation analysis (Section 4.3) demonstrates a high correlation between our multi-level metrics and human evaluation.

To address the limited visualization code generation capabilities, we construct a new dataset **SynthVis-30K**, and a novel model **PlotCraftor**. Specifically, as shown in Fig 3, we collect 30k multi-modal chart data instances covering 31 topics, 48 chart types, and 7 tasks. Each data instance contains a human instruction, some data files for visualization (CSV/XLSX), the resulting chart image, and the corresponding source code. Based on SynthVis-30K, we construct PlotCraftor that achieves strong capabilities in complex data visualization with a minimal model size.

Experimental results demonstrate that PlotCraftor improves performance on PlotCraft by 25%, achieving performance comparable to leading proprietary LLMs. Concurrently, we adapt other benchmarks to our task settings, including VisEval (Chen et al., 2024), PandasPlotBench (Galimzyanov et al., 2025). PlotCraftor demonstrates a 7% higher performance compared to the much larger Qwen3-Coder-480B. These results validate PlotCraftor's powerful abilities in different text-to-visualization code generation tasks.

## 2. The PlotCraft Benchmark

### 2.1. Task Definition

We define the data visualization task as a conditional code generation problem. Formally, given a natural language instruction $I$, metadata $M$, and a raw dataset $D$, a Large Language Model ($\mathcal{F}$) must generate an executable code snippet $C$. This code must use $D$ to render a visualization that satisfies all requirements outlined in $I$. This task is formulated as $C = \mathcal{F}(I, M, D)$. The PlotCraft benchmark evaluates models on two distinct variants of this core task.

The two variants are: **(1) Single-Turn Generation**, where the model must generate the complete visualization code in a single step from the initial instruction, thereby measuring its ability to execute complex requests from scratch; and **(2) Multi-Turn Refinement**, which assesses a model's ability to debug and iteratively improve existing code based on a conversation history and a new modification request, mirroring a realistic user interaction workflow.

### 2.2. Benchmark Coverage Analysis

The PlotCraft benchmark comprises 982 evaluation instances, evenly divided into 491 for single-turn generation and 491 for multi-turn refinement. Both single-turn and multi-turn tasks are based on the same underlying visualization goal. We describe the benchmark's coverage across three dimensions: Chart Types, Thematic Topics, and Task Complexity (the same as shown in Figure 3).

**Chart Types**  PlotCraft encompasses 48 distinct plot types, including scatter plots, bubble plots, dendrograms, and violin plots. Since a significant portion of tasks involves composite visualizations, single exclusive labels are often

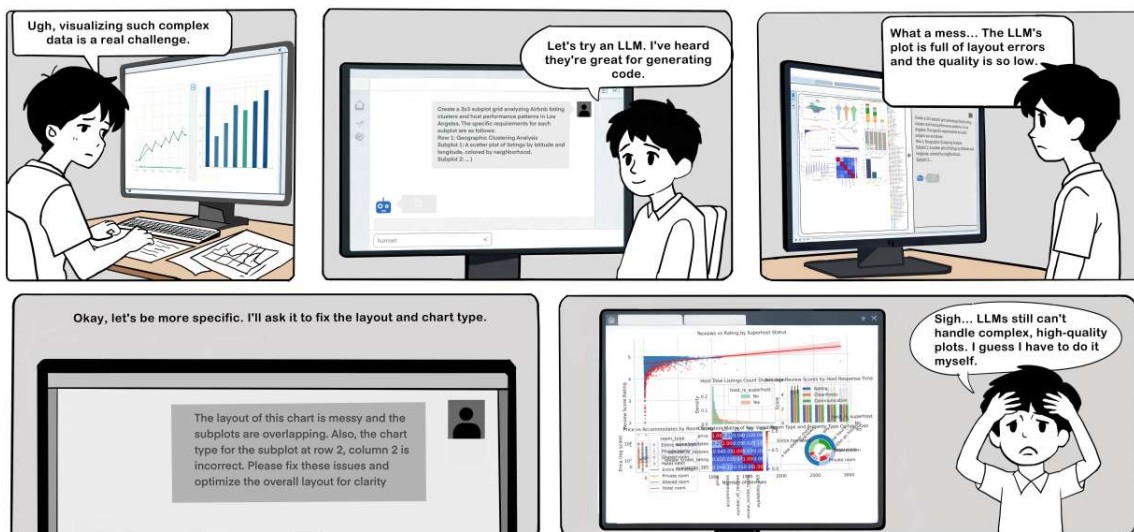

*Figure 2.* A real-world example illustrating the limitations of LLMs on complex visualization tasks. When presented with a sophisticated request, the model generates a low-quality output and struggles to make effective improvements during the subsequent refinement process.

inapplicable; thus, we focus on the diversity of graphical elements. Illustrative examples of these categories are provided in Appendix §A.1, and a detailed breakdown of the benchmark's thematic coverage—spanning domains is presented in Appendix §A.2.

**Cross-Chart Semantic Coherence** Beyond structural complexity, PlotCraft emphasizes semantic logicality. All multi-subplot tasks are grounded in a unified high-level visualization goal chosen from seven primary categories: Correlation, Deviation, Ranking, Distribution, Composition, Change, and Groups. These tasks exhibit intrinsic cross-chart semantic coherence and insight-driven coordination, where every subplot contributes to a holistic analytical narrative rather than serving as an isolated entity. For instance, as illustrated in Appendix Figure 7, all generated subplots are coordinated to collectively visualize the relationship between economic indicators and Olympic performance.

**Task Complexity** To systematically evaluate model capabilities, tasks are stratified into three levels of complexity: (1) Easy: The task requires generating a single, standard chart (e.g., a bar chart or a line plot) to visualize the data. (2) Medium: The task involves creating a composite visualization, which could be either a single chart combining multiple plot types or a subplot grid where each individual subplot is of a simple nature. (3) Hard: The task demands the creation of a complex subplot grid where each individual subplot is itself a composite chart, requiring advanced spatial and logical reasoning. We provide illustrative examples for different complexities in Appendix §A.3.

### 2.3. Data Curation Process

The construction of PlotCraft adheres to four guiding principles: (1) Grounded in Real Data: All tasks are based on

real-world datasets to avoid artifacts of synthetic data. (2) Built from Scratch: All tasks and reference solutions are newly created to prevent data leakage from existing sources. (3) Zero-Reference Generation: Tasks provide no sample images or code, requiring models to generate visuals from abstract instructions. (4) Compositional Complexity: The benchmark spans a wide spectrum of complexities, including tasks with intricate layouts and multiple chart types within a single figure. Adhering to these principles, we curate the benchmark through a 4-step pipeline, which we overview here. Further details are provided in Appendix §B.

**Data Filtering** We source open-source datasets from Kaggle. The use of these datasets in our work is for scientific research purposes only; other uses are subject to their original licenses. Using metrics such as vote counts, download counts, and usability ratings, we performed an initial screening of 7,162 datasets, comprising over 95,000 files and 25.6 billion data rows. From this pool, we conduct a second filtering stage to select 140 core datasets for the benchmark. This selection was curated to ensure diverse topics, a wide distribution of data volume and complexity, and minimal overlap with datasets used in other visualization tasks to mitigate data leakage. The final collection of benchmark datasets contains 1,874 raw data files and approximately 462 million data rows.

**Task and Instruction Writing** Using the selected benchmark datasets, we authored 491 unique visualization tasks. The design of these tasks was guided by a combination of seven high-level visualization intents and 48 distinct chart types. We ensure a roughly balanced distribution of tasks across three complexity levels: simple (159), medium (163), and hard (169). The accompanying instructions are written to be abstract, specifying only the visualization requirements without providing any guidance on code implementation.

*Table 1.* A comparison of PlotCraft with existing benchmarks. A ✓indicates the presence of a feature, while a ✗indicates its absence.

| Benchmarks | # of Test Instances | Composite Types | Multiple Subplots | Multi-Turn | Evaluation Metric |
|---|---|---|---|---|---|
| *Chart Understanding Benchmarks* | | | | | |
| ChartQA (Masry et al., 2022) | 10K | ✗ | ✗ | ✗ | Accuracy |
| ChartSumm (Rahman et al., 2023) | 84K | ✗ | ✗ | ✗ | Match-based |
| CharArXiv (Wang et al., 2024) | 93K | ✗ | ✓ | ✗ | MLLM Score |
| ChartX (Xia et al., 2025) | 1152 | ✓ | ✗ | ✗ | Multi-Level |
| *Chart to Code Benchmarks* | | | | | |
| Plot2Code (Wu et al., 2024) | 132 | ✗ | ✓ | ✗ | MLLM Score |
| Design2Code (Si et al., 2025) | 484 | ✗ | ✗ | ✗ | Multi-Level |
| ChartMimic (Yang et al., 2025b) | 4,800 | ✓ | ✓ | ✗ | Multi-Level |
| *Text to Visualization Benchmarks* | | | | | |
| MatPlotBench (Yang et al., 2024b) | 100 | ✗ | ✗ | ✗ | MLLM Score |
| VisEval (Chen et al., 2024) | 2300 | ✗ | ✗ | ✗ | Multi-Level |
| PandasPlotBench (Galimzyanov et al., 2025) | 150 | ✗ | ✗ | ✗ | Multi-Level |
| **PlotCraft** (Ours) | 982 | ✓ | ✓ | ✓ | Multi-Level |

**Reference Code Writing**  For each task, a reference solution is implemented by a team of five senior Python developers using Matplotlib and its associated libraries. It is important to note that this code serves as a valid reference implementation that achieves a high-quality result, rather than a single, optimal solution. The details of the sandboxed execution environment used for our evaluation are provided in Appendix §A.4.

**Multi-turn Conversation Synthesis**  To simulate realistic refinement scenarios, we first generate an initial code draft for each task using a less capable model. Human experts then review these drafts and intentionally introduce common, realistic errors—such as incorrect chart types or overlapping visual components—to create a faulty, low-quality implementation. Examples of such faulty code are provided in Appendix B.4. This faulty code, along with its rendered image and the original instruction, was presented to human annotators who then wrote a natural language modification request to correct the errors. Each multi-turn task instance is thus composed of the original instruction, data metadata, the faulty code, and this human-authored refinement prompt. This process yields 491 multi-turn tasks, which, combined with the 491 single-turn tasks, constitute the complete PlotCraft benchmark of 982 instances.

## 2.4. Evaluation Metrics

Our evaluation pipeline begins with a strict assessment of functional correctness. To be eligible for scoring, generated code must execute successfully within a sandboxed environment and produce a valid, non-empty image; any code failing this step receives a score of zero. For all valid outputs, we align with established practices by employing a **multi-judge ensemble** for automated visual assessment. While we acknowledge the inherent challenges and potential biases in automated visual and aesthetic judgment (Xie et al., 2025; Pan et al., 2025a; Shi et al., 2025), this ensemble approach aims to mitigate individual model variance and improve evaluation robustness. Indeed, as demonstrated by the eval-

uation results in Appendix §G using three distinct VLMs as judges, the ranking of chart generation performance remains consistent across different evaluators. The ensemble evaluates charts based on a two-dimensional framework: ❶ Task Compliance: This dimension is decomposed into four sub-metrics evaluated on a binary scale (0=Fail, 1=Pass): Layout Compliance, Chart Type Compliance, Requirement Fulfillment, and Complete Task Fulfillment. ❷ Chart Quality: This dimension assesses aesthetic and functional fidelity on a 3-point scale (0, 1, or 2) across five criteria: Clarity (e.g., avoidance of element occlusion), Layout Quality, Color Quality, Text Readability, and Professional Formatting. Detailed scoring rubrics, ensemble prompting strategies, and specific judgment cases are provided in Appendix §C.

## 3. The PlotCraftor

To address the performance gap observed in our benchmark evaluations, we developed PlotCraftor, a model specifically fine-tuned for complex data visualization. The development process consists of two key stages: the creation of a large-scale, high-quality synthetic dataset, which we call SynthVis-30K, and the subsequent Supervised Fine-Tuning (SFT) of a base model on this data.

### 3.1. SynthVis-30K Dataset

The SynthVis-30K dataset was created using a multi-agent framework, illustrated in Figure 3, which comprises two main stages: Task Generation and Code Generation. Details are in Appendix §E.

**Task Generation.**  We first sourced a collection of Kaggle datasets (with CC BY 4.0 or Apache 2.0 licenses), ensuring no overlap with those used in the PlotCraft benchmark to prevent data contamination. A Data Analyzer agent processes each dataset to extract structured metadata, including data types, column names, and sample rows. This formatted data then enters a Task Cycle, an iterative loop between a Task Generator and a Task Judge. The Task Generator, prompted with few-shot examples from PlotCraft, proposes 3-6 visu-

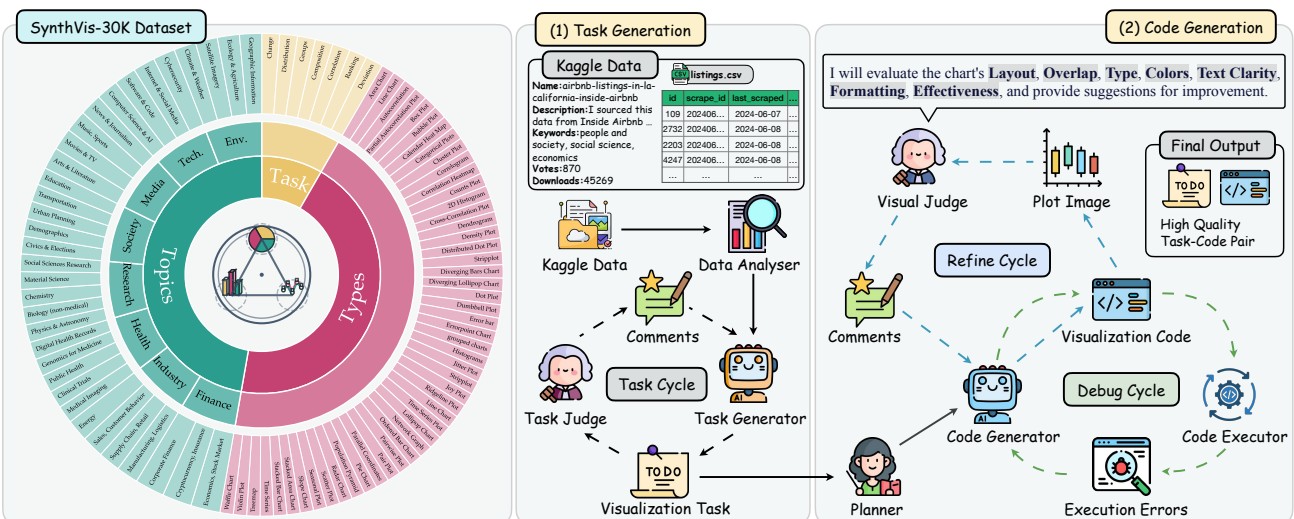

*Figure 3.* An overview of the SynthVis-30K dataset, detailing its coverage and the multi-agent framework used for its creation. **(Left)** A hierarchical chart illustrates the dataset's comprehensive coverage across three dimensions: thematic Topics, chart Types, and visualization Tasks. **(Right)** A schematic of our multi-agent data synthesis pipeline. This framework consists of two primary stages, Task Generation and Code Generation, which process raw Kaggle data to produce complete, multi-modal visualization instances. Each instance comprises structured data, a natural language instruction, the visualization code, and the corresponding rendered image.

alization tasks of varying difficulty for a given dataset. The Task Judge assesses these tasks for feasibility and coherence, providing feedback for refinement. This cycle repeats until a high-quality visualization task is finalized.

**Code Generation.** The generated task is then passed to a Planner agent, which creates a high-level plan for implementation. Following this plan, a Code Generator agent produces the visualization code through two iterative feedback loops: a Debug Cycle and a Refine Cycle. In the Debug Cycle, the code is passed to an Executor—a sandboxed Python environment (Appendix §A.4). If the code fails, the resulting error message is returned to the Code Generator for debugging until the code is executable. Subsequently, in the Refine Cycle, the error-free code is used to render an image. A Visual Judge evaluates this image on multiple criteria (Layout, Overlap, Type, Colors, Text Clarity, Formatting, and Effectiveness) and provides natural language feedback. The Code Generator refines the code based on this feedback. These cycles continue until the code is both executable and produces a high-quality visual, resulting in a final task-code pair.

**SFT Trajectory Synthesis.** Finally, we convert the curated task-code pairs into training instances for SFT. For single-turn data, we use an external LLM (Claude) to generate a concise Chain-of-Thought (CoT) rationale that explains the steps from the task instruction to the final code. The instruction, CoT, and code are then combined into a single-turn training trajectory. For multi-turn data, we extract high-quality interactions from the Refine Cycle logs, specifically selecting instances where the Visual Judge's feedback led to a significant improvement in the code's quality score. These successful interactions form our multi-turn refinement trajectories.

### 3.2. Model Training

We develop PlotCraftor by Supervised Fine-Tuning (SFT) on the Qwen3-Coder-30B-A3B model using our SynthVis-30K dataset. The training is configured with a context length of 131,072 tokens and proceeded for 3 epochs. We use a global batch size of 64 and a micro-batch size of 1. The learning rate is $7 \times 10^{-6}$, which includes a warm-up phase of 30 steps. The learning rate is subsequently decreased via a linear decay schedule to a minimum of $7 \times 10^{-7}$. We utilize BF16 for mixed-precision training to enhance efficiency.

## 4. Experiments

### 4.1. Experiments Setup

**Baseline Setup** We evaluate a comprehensive suite of 24 models, comprising our proposed **PlotCraftor** and 23 leading proprietary and open-source Large Language Models (LLMs). This selection covers a wide spectrum of capabilities and parameter sizes, ranging from 7B to over 1T parameters, to ensure a robust comparative analysis. A detailed enumeration of all baseline models is provided in Appendix §I.

**Benchmark Setup** To ensure a comprehensive evaluation, in addition to our primary PlotCraft benchmark, we assessed model performance on two other established text2vis benchmarks. ❶ VisEval (Chen et al., 2024), is a large-scale benchmark composed of high-quality visualization tasks curated from nvBench (Luo et al., 2025). It employs a multi-faceted evaluation protocol that includes execution pass rate,

*Table 2.* Quantitative results on PlotCraft for 16 primary LLMs across two settings: Single-Turn Generation and Multi-Turn Refinement. Task-Comp. and Quality denote the total scores for the Task Compliance (out of a maximum of 4) and Chart Quality (out of a maximum of 10) sub-metrics, respectively. AVG score is the average of these scores across both single-turn and multi-turn evaluations. Within each model category, the best score is **bolded** and the second-best is underlined.

| Model | Single-Turn Generation | | | Multi-Turn Refinement | | | AVG score |
|---|---|---|---|---|---|---|---|
| | Pass Rate (%) | Task-Comp. | Quality | Pass Rate (%) | Task-Comp. | Quality | |
| *Closed-source LLMs* | | | | | | | |
| Claude-4.1-Opus (Anthropic, 2023) | **76.50** | **1.95** | **4.19** | **81.74** | **2.07** | **5.21** | **6.72** |
| Claude-4-Sonnet (Anthropic, 2023) | 69.14 | 1.75 | 3.98 | 78.71 | 1.90 | 4.79 | 6.22 |
| Gemini-2.5-Pro (Team, 2024) | 41.64 | 1.17 | 2.30 | 59.16 | 1.53 | 3.79 | 4.41 |
| ChatGPT-4o-Latest (OpenAI, 2023) | 63.84 | 1.62 | 3.32 | 69.55 | 1.53 | 4.22 | 5.35 |
| GPT-5 (OpenAI, 2025) | 70.16 | 1.78 | 2.86 | 74.43 | 1.82 | 4.32 | 5.40 |
| Grok-4 (xAI, 2025) | 64.82 | 1.65 | 3.65 | 70.54 | 1.63 | 4.41 | 5.67 |
| *Open-source LLMs* | | | | | | | |
| Kimi-K2 (Team et al., 2025b) | 60.43 | 1.54 | 3.35 | 61.33 | 1.51 | 4.04 | 5.23 |
| DeepSeek-V3.1 (DeepSeek-AI & etc., 2024) | 56.01 | 1.51 | 3.19 | 57.16 | 1.52 | 3.98 | 5.11 |
| GLM-4.5 (Team et al., 2025a) | 43.68 | 1.27 | 2.47 | 65.27 | 1.56 | 3.90 | 4.61 |
| GPT-oss-120B (Agarwal et al., 2025) | 48.53 | 1.34 | 2.67 | 29.99 | 0.79 | 1.86 | 3.34 |
| GPT-oss-20B (Agarwal et al., 2025) | 44.98 | 1.23 | 2.42 | 34.32 | 0.91 | 2.13 | 3.36 |
| Seed-Coder-8B (Seed et al., 2025) | 32.68 | 0.89 | 1.73 | 57.33 | 1.28 | 3.21 | 3.57 |
| VisCoder-7B (Ni et al., 2025) | 25.76 | 0.69 | 1.49 | 52.03 | 1.01 | 2.95 | 3.08 |
| Qwen3-Coder-480B-A35B (Hui et al., 2024) | 61.60 | 1.58 | 3.20 | 76.27 | 1.77 | 4.45 | 5.51 |
| Qwen3-Coder-30B-A3B (Hui et al., 2024) | 52.85 | 1.34 | 2.84 | 73.42 | 1.57 | 4.17 | 4.97 |
| **PlotCraftor-30B-A3B** (Ours) | **64.66** | **1.75** | **4.08** | **77.41** | **1.78** | **4.73** | **6.18** |

*Table 3.* Quantitative results for 16 LLMs across two benchmarks, VisEval and PandasPlotBench.

| Model | VisEval | | | | | PandasPlotBench | | |
|---|---|---|---|---|---|---|---|---|
| | Invalid Rate (%) | Illegal Rate (%) | Pass Rate (%) | Readability | Quality | Pass Rate (%) | Vis | Task |
| *Closed-source LLMs* | | | | | | | | |
| Claude-4.1-Opus (Anthropic, 2023) | **0.87** | **5.42** | **96.14** | **4.39** | **3.91** | 100 | 78 | 95 |
| Claude-4-Sonnet (Anthropic, 2023) | 1.82 | 6.78 | 94.51 | 4.27 | 3.85 | 98.3 | 75 | 92 |
| Gemini-2.5-Pro (Team, 2024) | 6.79 | 15.83 | 83.71 | 3.76 | 3.09 | 85.1 | 66 | 77 |
| ChatGPT-4o-Latest (OpenAI, 2023) | 3.39 | 14.54 | 84.47 | 4.01 | 3.34 | 93.7 | 72 | 89 |
| GPT-5 (OpenAI, 2025) | 5.01 | 12.98 | 87.71 | 4.22 | 3.52 | 82.9 | 66 | 79 |
| Grok-4 (xAI, 2025) | 4.02 | 8.53 | 94.41 | 4.30 | 3.69 | 94.6 | 75 | 88 |
| *Open-source LLMs* | | | | | | | | |
| Kimi-K2 (Team et al., 2025b) | **2.45** | **8.91** | 93.62 | 4.24 | 3.78 | 95.4 | **76** | 91 |
| DeepSeek-V3.1 (DeepSeek-AI & etc., 2024) | 5.94 | 9.82 | 92.31 | 4.17 | 3.59 | 93.7 | 74 | 86 |
| GLM-4.5 (Team et al., 2025a) | 5.03 | 13.07 | 88.54 | 4.19 | 3.48 | 47.4 | 39 | 44 |
| GPT-oss-120B (Agarwal et al., 2025) | 3.91 | 14.88 | 90.13 | 4.03 | 3.35 | 92.6 | 71 | 88 |
| GPT-oss-20B (Agarwal et al., 2025) | 5.23 | 15.12 | 86.34 | 3.95 | 3.28 | 85.7 | 69 | 87 |
| Seed-Coder-8B (Seed et al., 2025) | 8.15 | 18.25 | 75.40 | 3.45 | 2.81 | 54.3 | 65 | 80 |
| VisCoder-7B (Ni et al., 2025) | 6.45 | 16.05 | 82.88 | 3.81 | 3.12 | 87.7 | 66 | 78 |
| Qwen3-Coder-480B-A35B (Hui et al., 2024) | 2.89 | 11.53 | 91.75 | 4.18 | 3.55 | **97.1** | 73 | 87 |
| Qwen3-Coder-30B-A3B (Hui et al., 2024) | 3.55 | 13.90 | 89.67 | 4.10 | 3.41 | 94.3 | 71 | 86 |
| **PlotCraftor-30B-A3B** (ours) | 2.61 | 9.76 | **93.81** | **4.26** | **3.80** | 96.0 | 74 | **91** |

GPT-4 based scoring, and SVG-based layout checks. ❷ PandasPlotBench (Galimzyanov et al., 2025) is a benchmark containing 175 test cases specifically designed to evaluate code generation for Matplotlib and its related libraries.

**Evaluation Details** All experiments were conducted by sending requests in the standard OpenAI API format, with the chat structure following the ChatML format (OpenAI, 2022). We employed a greedy decoding strategy and set the maximum output length to 131,072 tokens; if a model did not support this length, we used its specified maximum limit. The final reported results are the average of three experimental runs. For proprietary models, we queried their official APIs directly. For open-weight models, we utilized the vLLM framework for serving. It is important to note a

potential limitation regarding the GPT-oss series of models: as they are trained on the harmony format (Agarwal et al., 2025), which natively incorporates multi-turn reasoning and tool invocation, our standardized ChatML-based setup may not fully leverage their intended capabilities. Additional evaluation details and specific prompts are in Appendix H.

### 4.2. Main Results

This section presents the results of 16 leading LLMs on the PlotCraft, VisEval, and PandasPlotBench benchmarks. PlotCraft scores reported in Table 2 represent the average of a multi-judge ensemble comprising Gemini-2.5-Pro, Gemini-2.5-Flash, and GPT-4o; detailed breakdowns for each judge are provided in Appendix §G. Com-

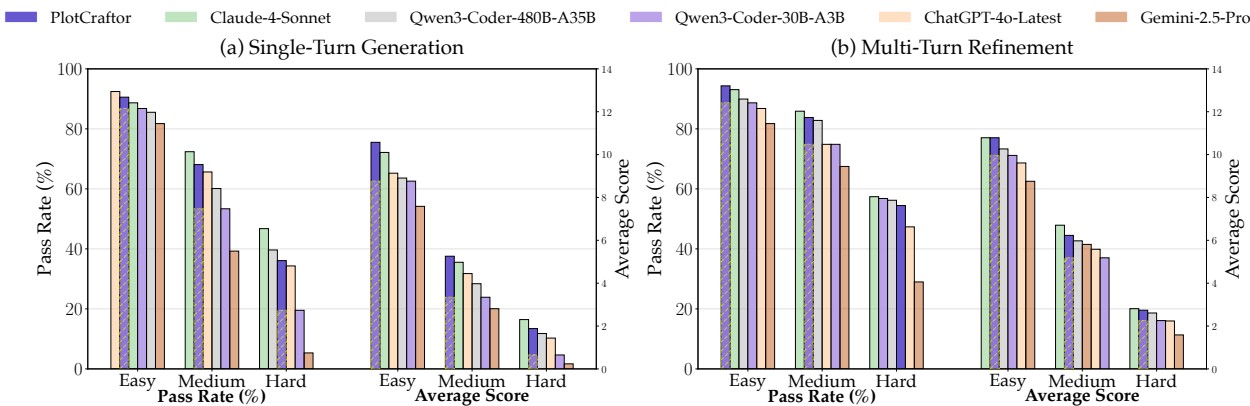

*Figure 4.* Performance comparison of PlotCraftor and five leading LLMs on tasks of varying difficulty. The figure is split into two subplots: **(a)** Single-Turn Generation and **(b)** Multi-Turn Refinement. Within each subplot, we report the Pass Rate (%) and Average Score for tasks categorized as Easy, Medium, and Hard. The yellow hatched area within the PlotCraftor bars indicates the score contribution from its base model, Qwen3-Coder-30B-A3B.

prehensive results for all 24 evaluated models are available in Appendix Table 7, while comparisons on VisEval and PandasPlotBench are shown in Table 3. For a granular analysis, Figures 1 and 4 decompose performance by sub-metrics and task difficulty. The key findings are as follows.

**Claude-4.1-Opus leads proprietary models, while PlotCraftor is the top-performing open-weight model, achieving results comparable to Claude-4-Sonnet.** Among proprietary models, Claude-4.1-Opus demonstrates a significant advantage across all tasks. It achieved an average of 6.72 on PlotCraft and a 100% pass rate on PandasPlotBench, substantially outperforming other models.

Among open-weight models, our PlotCraftor achieves SOTA performance on PlotCraft. It surpasses its base model, Qwen3-Coder-30B-A3B, by a margin of 25% and nearly matches the performance of the top-tier Claude 4 Sonnet, with a score difference of less than 1%. PlotCraftor also secures leading results among open models on the VisEval and PandasPlotBench benchmarks. An analysis of performance by task difficulty, focusing on 6 key models (4 widely-used models, our baseline, and PlotCraftor), reveals that while GPT-4o excels on simple, single-turn tasks, Claude 4.1 Opus shows superior performance on more complex and demanding tasks. A more detailed discussion and comparison between different models is available in Appendix §J.

**Strong performance on simple benchmarks does not guarantee success on complex visualization tasks.** We observe a significant performance disparity between models on different benchmarks. Models such as Kimi-K2 and GPT-4o, which perform exceptionally well on the simpler tasks found in VisEval and PandasPlotBench, experience a notable performance degradation on the complex, multifaceted tasks within PlotCraft. In contrast, the Claude series and our PlotCraftor demonstrate consistently high performance across all three benchmarks, indicating their robust capability in handling both simple and sophisticated

visualization challenges.

**Task difficulty significantly impacts the effectiveness of model scaling and fine-tuning.** As shown in Figure 5, the benefits of model scaling are highly dependent on task difficulty. For models under 100B parameters, performance on Easy tasks scales rapidly with size, while performance on Hard tasks remains flat, only improving for models beyond the 100B threshold. This disparity is mirrored in SFT: smaller models can be fine-tuned to near-proprietary levels on Easy tasks, yet SFT provides minimal benefit for Hard tasks. This indicates that solving complex visualization challenges may rely more on the emergent reasoning abilities that come with scale than on task-specific fine-tuning.

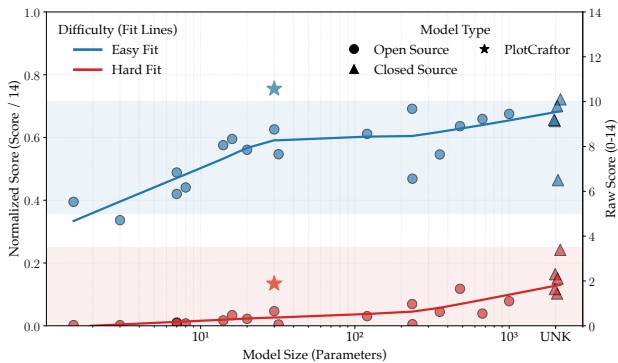

*Figure 5.* Performance scaling on Easy vs. Hard.

### 4.3. Correlation with Human Evaluation

To validate the reliability of our proposed multi-dimensional metrics, we measured their correlation with human expert judgments. We conducted a scoring experiment on 500 charts randomly sampled from the output of PlotCraftor. For each chart, three human annotators provided scores for all metrics, with the final ground-truth score determined by a majority vote. Detailed descriptions of the human evalua-

tion scale and reliability analysis are provided in Appendix §F. We then calculated the Cohen's Kappa coefficient to assess the inter-rater agreement between these human-derived scores and the automated scores provided by our **multi-judge ensemble**. As presented in Table 4, the average Kappa score across all evaluation metrics reached a substantial level of agreement, demonstrating the reliability of our automated evaluation framework. A comparative analysis of other models as evaluators is detailed in Appendix K, where we found that individual models were significantly less effective at replicating human scoring patterns compared to the ensemble approach.

*Table 4.* Cohen's Kappa scores indicating the agreement between our automated **multi-judge ensemble** and human evaluations. By organizing metrics into Compliance and Quality dimensions, we observe substantial alignment across both categories.

| Compliance Metrics | | Quality Metrics | |
|---|---|---|---|
| Metric | $\kappa$ | Metric | $\kappa$ |
| Layout | 0.90 | Clarity | 0.73 |
| Type | 0.78 | Layout | 0.71 |
| Visual | 0.72 | Color | 0.69 |
| Task | 0.80 | Text | 0.65 |
| | | Format | 0.73 |

### 4.4. Error Analysis

To understand the primary failure modes on the PlotCraft benchmark, we analyzed common errors and categorized them into three main types. **(1) Code-Level Errors** includes functionally flawed code that causes runtime exceptions or critical rendering failures, such as using non-existent object attributes or creating conflicts between incompatible layout managers (e.g., `constrained_layout` with `fig.legend()`). **(2) Task Compliance Failures** occur when the visual output disregards explicit instructions, such as generating an incorrect subplot layout or using the wrong chart types. **(3) Deficiencies in Chart Quality** refers to aesthetic issues that degrade the visualization's readability and professional appearance, including element overlap, suboptimal color choices, and incorrect text. A more detailed, case-by-case analysis of these errors is provided in Appendix §L.

## 5. Related Works

### 5.1. Code Generation

Recent advances in LLMs, including general-purpose models (e.g., GPT (OpenAI, 2023), Claude (Anthropic, 2023), Gemini (Team, 2024)) and specialized code models (e.g., Qwen-Coder (Hui et al., 2024), DeepSeek-Coder (Guo et al., 2024), Codestral (MistralAI, 2024)), have demonstrated powerful coding capabilities (DeepSeek-AI & etc., 2024;

Rozière et al., 2023; Team et al., 2025b;a). While conventional tasks like algorithmic problem-solving (Chen et al., 2021a; Zhuo et al., 2025) and software engineering (Jimenez et al., 2023; Zhang et al., 2025) are evaluated on functional correctness (Chen et al., 2021b), this paper focuses on data visualization, a domain where generated code must produce a visually accurate output, a requirement shared by front-end design (Xu et al., 2025; Lu et al., 2025) and SVG generation (Xing et al., 2025).

### 5.2. Data Visualization

Prior LLM-based data visualization research spans three areas: The first, chart understanding (e.g., QA and captioning), focuses on interpreting visual information from plots, such as for question answering or summary generation (Li et al., 2024b; Zeng et al., 2024; Rahman et al., 2023; Kantharaj et al., 2022; Jia et al., 2025; Li et al., 2024a; Wang et al., 2025). The second, Chart-to-Code (Chart2Code) generation, involves reverse-engineering a visualization by generating the code required to replicate it (Wu et al., 2024; Yang et al., 2025b; Zhao et al., 2025; Zhang et al., 2026b; Sun et al., 2025; Chen et al., 2025), and the third, Text-to-Visualization (Text2Vis), concerns generating visualization specifications or code from natural language descriptions (Luo et al., 2025; Galimzyanov et al., 2025; Ni et al., 2025). However, these works predominantly focus on simple, single-panel plots, offering limited differentiation for advanced LLMs' ability to handle complex layouts and high information density. We introduce PlotCraft to address this critical evaluation gap.

## 6. Conclusion and Future Direction

In this work, we addressed the significant gap in the ability of Large Language Models to handle complex data visualization. We introduced PlotCraft, a novel and challenging benchmark designed to evaluate both single-turn generation and multi-turn refinement capabilities. Our comprehensive evaluation of 23 leading models on PlotCraft demonstrated that current LLMs struggle with sophisticated visualization tasks. To bridge this performance gap, we developed the SynthVis-30K dataset and used it to train PlotCraftor, a novel and efficient model. Experimental results show that PlotCraftor achieves state-of-the-art performance among open-weight models, delivering results comparable to leading proprietary LLMs. Meanwhile, we observed that there are still limitations in data generation and evaluation. The synthesis of high-quality data relies on costly multi-agent systems, and MLLM-based judges may fail to detect subtle visual errors, like minor overlaps or color style. This might be an area for future research. Furthermore, while PlotCraft focuses on detailed instructions to rigorously test capability limits, exploring model performance under abstract or open-ended goals remains a promising direction for future

work.

## Impact Statement

This paper presents work whose goal is to advance the field of Machine Learning. There are many potential societal consequences of our work, none which we feel must be specifically highlighted here.

## Acknowledgments

This work is supported by Strategic Priority Research Program of Chinese Academy of Sciences (XDA0480101), National Natural Science Foundation of China (92570204, 62576339)

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

# A. Benchmark Coverage Details

This section provides a detailed breakdown of the PlotCraft benchmark's coverage. Table 5 presents the frequency distribution of 48 distinct chart types as they are mentioned within the instructions for each visualization task in the PlotCraft dataset. This distribution highlights the diversity of chart creation tasks included in the benchmark.

*Table 5.* This table illustrates the frequency distribution of 48 chart types mentioned within the instructions of visualization tasks from the PlotCraft dataset. To prevent data skew from repetition, each distinct task was analyzed and counted a single time, given that the same set of tasks is utilized for both single-turn and multi-turn interactions.

| Type | Count | Type | Count | Type | Count | Type | Count |
|---|---|---|---|---|---|---|---|
| Line Chart | 279 | Scatter Plot | 225 | Time Series | 179 | Area Chart | 164 |
| Correlation Heatmap | 145 | Violin Plot | 132 | Box Plot | 130 | Stacked Area Chart | 127 |
| Error bar | 96 | Bubble Plot | 91 | Time Series Plot | 88 | Density Plot | 85 |
| Stacked Bar Chart | 79 | Network Graph | 75 | Parallel Coordinates | 71 | Histograms | 70 |
| Pie Chart | 68 | Radar Chart | 67 | grouped charts | 67 | Counts Plot | 57 |
| Dendrogram | 56 | Slope Chart | 56 | Seasonal Plot | 54 | Treemap | 54 |
| Cluster Plot | 44 | Population Pyramid | 24 | Pair Plot | 17 | Lollipop Chart | 15 |
| Dumbbell Plot | 14 | Autocorrelation | 10 | Cross-Correlation Plot | 9 | Waffle Chart | 9 |
| 2D Histogram | 7 | Diverging Lollipop Chart | 7 | Pairwise Plot | 7 | Dot Plot | 6 |
| Jitter Plot | 6 | Partial Autocorrelation Plot | 6 | Categorical Plots | 4 | Joy Plot | 4 |
| Diverging Bars Chart | 3 | Ridgeline Plot | 2 | Stripplot | 2 | Calendar Heat Map | 1 |
| Correlogram | 1 | Distributed Dot Plot | 1 | Errorpoint Chart | 1 | Ordered Bar Chart | 1 |

## A.1. Chart Types

**Distribution**   Tasks in this category require the visualization of data spread and grouping. Figure 6 presents a representative example of a class distribution task. The associated instruction demands the generation of a complex, multi-panel figure: "Create a comprehensive 3x3 subplot grid analyzing the distribution patterns of ECG signal characteristics across normal and abnormal cases."

**Correlation**   This category focuses on tasks that require visualizing the relationship and interdependence between two or more variables. Figure 7 illustrates a representative example, for which the model must generate a complex multi-panel visualization. The instruction for this task is: "Create a comprehensive 3x3 subplot grid analyzing the relationship between economic indicators and Olympic performance across different regions."

**Groups**   This category involves tasks that require visualizing clusters, hierarchies, and other logical groupings within a dataset. Figure 8 shows a complex example where the model is instructed to represent intricate relationships with the prompt: "Create a comprehensive 3x3 subplot grid analyzing the clustering patterns and hierarchical relationships within the Brawl Stars competitive ecosystem."

**Change**   Tasks in the Change category focus on visualizing trends, time-series data, and the evolution of metrics over a period. A representative example is presented in Figure 9, where the instruction requires a multi-faceted temporal analysis: "Create a comprehensive 3x2 subplot grid analyzing the temporal evolution of NIRF rankings across different educational categories from 2016-2021."

**Composition**   This category requires visualizing the composition of a whole, illustrating the proportions and breakdown of its constituent parts. Figure 10 provides an example of a multi-dimensional composition task, with the instruction: "Create a composite visualization showing Netflix content composition across different dimensions, designed as a 2x2 subplot grid."

**Ranking**   Ranking tasks involve comparing and ordering items based on one or more quantitative metrics. Figure 11 showcases an example where the model is prompted to perform a comparative ranking with the instruction: "Create a comprehensive ranking visualization that compares Udemy course performance across different metrics. Design a 2x2 subplot grid."

**Deviation**  The Deviation category includes tasks that visualize the difference or variance of data points against a fixed baseline or between different groups. An example is shown in Figure 12, where the task is to analyze prediction errors: "Create a 3x3 grid of composite visualizations analyzing the deviation patterns in Titanic passenger survival predictions."

## A.2. Thematic Coverage

The benchmark's thematic coverage is structured around eight high-level domains: **Finance** (Business & Finance), **Industry** (Industry & E-commerce), **Health** (Health & Medicine), **Research** (Science & Research), **Society** (Government & Society), **Media** (Culture & Media), **Tech.** (Technology & Computing), and **Env.** (Environment & Geospatial). These domains are further subdivided into 31 fine-grained sub-topics. A comprehensive list of all included sub-topics is presented in Table 6.

*Table 6.* This table illustrates the coverage of 31 fine-grained thematic topics within the PlotCraft dataset.

| Type | Count | Type | Count | Type | Count | Type | Count |
|---|---|---|---|---|---|---|---|
| Manufacturing, Logistics | 18 | Supply Chain, Retail | 17 | Sales, Customer Behavior | 22 | Energy | 11 |
| Medical Imaging | 25 | Clinical Trials | 16 | Public Health | 20 | Genomics for Medicine | 11 |
| Digital Health Records | 19 | Physics & Astronomy | 14 | Biology (non-medical) | 15 | Chemistry | 13 |
| Material Science | 12 | Social Sciences Research | 15 | Civics & Elections | 10 | Demographics | 12 |
| Urban Planning | 14 | Transportation | 16 | Education | 18 | Arts & Literature | 11 |
| Movies & TV | 13 | Music, Sports | 12 | News & Journalism | 19 | Computer Science & AI | 44 |
| Software & Code | 28 | Internet & Social Media | 21 | Cybersecurity | 14 | Climate & Weather | 12 |
| Satellite Imagery | 7 | Ecology & Agriculture | 12 | Geographic Information | – | – | – |

## A.3. Task Complexity

Tasks within the PlotCraft benchmark are stratified into three distinct levels of complexity:

- **Easy**: Tasks require the generation of a single, standard chart (e.g., one bar chart, one line chart, or one scatter plot) to visualize the data.

- **Medium**: Tasks involve creating a composite visualization. This can be either: (a) a single figure that integrates multiple plot types or variables (e.g., a bar chart with a line plot overlay), or (b) a multi-panel grid (e.g., 2x1, 2x2) composed of simple, individual plots.

- **Hard**: Tasks demand the creation of a complex, multi-panel grid (e.g., 2x2, 3x3) wherein each individual subplot is itself a composite chart. For example, a 2x2 grid where each of the four plots contains both a histogram and a Kernel Density Estimation (KDE) curve.

To illustrate these levels, Figure 13 displays the instructions and raw data for three tasks of varying difficulty, all derived from the `firethorn-10-world-co2-emission-analysis` dataset. Figure 14 then shows the corresponding ground-truth reference images for each of these three instructions.

## A.4. Sandboxed Envirnment

To ensure the safe, consistent, and reproducible execution of all model-generated code, we constructed a dedicated sandboxed environment containerized using Docker. This approach provides an isolated and standardized platform, guaranteeing that all models are evaluated under identical conditions.

The environment is based on Python 3.13 and comes pre-installed with a comprehensive suite of libraries essential for data analysis and visualization. The foundational libraries include Pandas for data manipulation and NumPy for numerical operations. For visualization, the environment is equipped with Matplotlib as the primary plotting library, complemented by a wide array of higher-level and specialized libraries to support diverse charting requirements. These include:

- **Seaborn**: For high-level statistical graphics.

- **Plotly**: For interactive charts.

- **Squarify**: For creating treemaps.

- **scikit-learn**: For generating machine learning-related plots like confusion matrices.

- **statsmodels**: For advanced statistical visualizations.

All evaluations are conducted on a server equipped with 128 CPU cores and 1024 GB of RAM. Each code execution is performed within its container without network access and is subject to a strict execution timeout of 120 seconds to prevent runaway processes. This fully-specified, sandboxed setup eliminates variability from system configurations and provides a fair and secure assessment of each model's code generation capabilities.

## B. Benchmark Data Curation Details

### B.1. Design Principles

The PlotCraft benchmark was designed to address critical gaps in existing evaluation methodologies, guided by four core principles for a more realistic and comprehensive assessment of LLM visualization capabilities.

- **Grounded in Real Data**: To ensure practical relevance and ecological validity, all tasks are constructed using authentic, real-world datasets. This principle allows the benchmark to circumvent the limitations and potential artifacts of synthetic data, which often lacks the noise, complexity, and inherent quirks found in genuine data sources. By grounding tasks in reality, PlotCraft provides a more robust evaluation of a model's ability to handle practical data visualization scenarios.

- **Built from Scratch**: To mitigate data contamination and prevent benchmark leakage, all tasks and corresponding code in PlotCraft are built from scratch. We utilize open-source datasets from Kaggle to generate novel visualization challenges, ensuring no overlap with existing benchmarks or code repositories. This approach guarantees a more accurate assessment of a model's true generalization capabilities.

- **Zero-Reference Generation**: All tasks are initiated from text-only instructions, with no reference images or code provided. This setup compels the model to generate a visualization from an abstract concept, mirroring the creative and interpretive workflow of a data analyst, rather than the simpler task of replicating a known visual pattern.

- **Compositional Complexity**: PlotCraft's tasks feature a wide spectrum of complexity, requiring the generation of multi-panel plots with intricate layouts (e.g., 2×2, 3×3 grids), shared axes, complex legends, and the combination of multiple chart types within a single figure. This focus on composition directly probes the spatial and logical reasoning skills that are undertested by current benchmarks.

### B.2. Data Filtering

Our data curation process began with sourcing datasets from Kaggle. This process was structured into two distinct phases to ensure the final benchmark was of high quality, diverse, and robust.

The first phase consisted of a large-scale, automated pre-screening of an initial pool of 7,162 candidate datasets, which collectively comprised over 95,000 files and 25.6 billion data rows. To filter for quality and relevance, we applied quantitative thresholds for community engagement metrics (vote counts ¿ 20 and download frequency $\geq$ 100) and official Kaggle usability ratings. Datasets that did not meet these minimum criteria for community validation and documentation quality were programmatically excluded.

In the second phase, the resulting pool of high-quality datasets underwent a rigorous manual curation to select the final 140 core datasets for the benchmark. This selection was guided by three key principles. First, to ensure thematic diversity, we employed a stratified approach based on domain tags, guaranteeing broad coverage across topics like finance, healthcare, and technology. Second, we deliberately selected for a wide distribution of data volume and complexity, ensuring the inclusion of everything from small, clean tables to large, multi-file relational datasets. Finally, to mitigate data leakage, we conducted a thorough review to exclude datasets known to be prevalent in major pre-training corpora or other common visualization benchmarks.

The final curated collection for PlotCraft consists of 1,874 raw data files, totaling approximately 462 million data rows, providing a robust and novel foundation for evaluating visualization models.

## B.3. Task and Instruction Writing

The creation of our benchmark's 491 unique visualization tasks was a meticulous, two-phase process designed to ensure depth, diversity, and a rigorous test of model capabilities.

The first phase, Comprehensive Task Generation, was conducted by a team of three data visualization experts. Adopting a systematic approach, the experts were tasked with authoring prompts for each of the 140 curated datasets. For each dataset, they targeted seven distinct, high-level visualization intents (Correlation, Deviation, Ranking, etc.) and aimed to create a variant for each of the three complexity levels (Easy, Medium, and Hard). To guide this process, the experts utilized a rich taxonomy of over 50 distinct chart types. This initial generation phase resulted in a large candidate pool of nearly 2,940 tasks (140 datasets × 7 intents × 3 complexity levels), providing comprehensive coverage of datasets, analytical goals, and difficulty.

The second phase, Collaborative Curation and Refinement, involved multiple rounds of review where the expert team discussed and filtered the large initial pool down to the final 491 tasks. Tasks were selected based on several criteria, including the clarity of the prompt, the feasibility of the visualization, and its analytical value. A key goal of this curation was to select the highest-quality examples while maintaining the balanced distribution across the three complexity levels established during generation. The final set reflects this, with 159 Easy, 163 Medium, and 169 Hard tasks.

Crucially, all final instructions were refined to be abstract and goal-oriented. They articulate the analytical requirements of the visualization, such as the desired layout, markings, and data transformations—without providing any guidance on code implementation. This approach compels the model to reason about the task from first principles, mirroring a more realistic human workflow.

The seven high-level visualization intents, along with the extensive range of chart types considered for each, are detailed below:

- **Correlation**: Scatter Plot, Bubble Plot, Scatter with Best Fit Line, Jitter Plot, Counts Plot, Scatter with Marginal Plots, Correlogram, Pairwise Plot, Network Graph, and Cluster Plot.

- **Deviation**: Diverging Bars Chart, Diverging Lollipop Chart, Slope Chart, Dumbbell Plot, Area Chart, Radar Chart, and Errorbar / Errorpoint Chart.

- **Ranking**: Ordered Bar Chart, Lollipop Chart, Dot Plot, Slope Chart, Dumbbell Plot, Stacked Bar Chart, and Diverging Lollipop Chart.

- **Distribution**: Histogram, Density Plot, Box Plot, Violin Plot, Joy Plot / Ridgeline Plot, Population Pyramid, Jitter Plot / Stripplot, and Categorical Plots.

- **Composition**: Stacked Bar Chart, Stacked Area Chart, Pie Chart, Treemap, and Waffle Chart.

- **Change**: Line Chart / Time Series Plot, Area Chart, Time Series with Error Bands, Calendar Heatmap, Seasonal Plot, Slope Chart, Dumbbell Plot, and Time Series Decomposition Plot.

- **Groups**: Dendrogram, Cluster Plot, Network Graph, Radar Chart, Treemap, Parallel Coordinates Plot, and Grouped Charts.

## B.4. Multi-turn Coversation

The following figures 15, 16, 17, and 18 provide several examples of the setup for our multi-turn refinement tasks. Each example consists of two key components: (1) the rendered image produced by an initial, intentionally flawed code implementation, and (2) the corresponding human-authored natural language request for its modification.

## B.5. Quality Control

To guarantee the robustness of PlotCraft, we implemented a multi-stage quality control protocol involving cross-validation and consensus-based adjudication.

**Multi-turn Refinement Verification**   Ensuring the correctness of the multi-turn tasks required verifying the logical consistency between the faulty image, the error description, and the refinement instruction. We employed a \*\*triple-verification strategy\*\* for this phase:

1. **Triplet Validation**: Each generated triplet (Original Instruction, Faulty Image, Refinement Request) was reviewed by three independent annotators.

2. **Factuality & Consistency Check**: Annotators verified two specific conditions: First, *Error Existence*—confirming that the errors described in the refinement request (e.g., "the text overlaps") were visually present in the rendered image. Second, *Instruction Consistency*—ensuring the refinement request did not contradict the original visualization goal.

3. **Majority Vote & Consensus**: A task was only accepted if it received a unanimous "Pass" vote. In cases of partial agreement (2/3), the triplet underwent a consensus discussion phase to determine if the refinement prompt could be polished. Triplets failing to reach consensus were discarded.

This strict verification loop ensures that all multi-turn interactions in PlotCraft reflect grounded, logical, and realistic user behaviors.

## Example 1 - Distribution

### Instruction

Create a comprehensive 3x3 subplot grid analyzing the distribution patterns of ECG signal characteristics across normal and abnormal cases. Each subplot should combine multiple visualization techniques: (1) Top-left: Histogram with KDE overlay showing the distribution of early signal values (columns 0-20 average), (2) Top-center: Histogram with KDE overlay for mid-signal values (columns 60-80 average), (3) Top-right: Histogram with KDE overlay for late signal values (columns 120-139 average), (4) Middle-left: Box plot with violin plot overlay comparing signal variance across normal vs abnormal cases, (5) Middle-center: Ridge plot showing signal amplitude distributions at different time segments (early, mid, late) for both classes, (6) Middle-right: Histogram with KDE overlay of peak-to-peak amplitude differences, (7) Bottom-left: Strip plot with box plot overlay showing signal energy distribution by class, (8) Bottom-center: Histogram with KDE overlay of signal baseline drift (difference between first and last 10 values), (9) Bottom-right: Population pyramid comparing the distribution of signal complexity measures between normal and abnormal ECG readings. Use different colors for normal (0) and abnormal (1) cases throughout all subplots.

### Files

📄 **ecg.csv**

### Reference Image

*Figure 6.* An example of a complex distribution visualization task from the PlotCraft benchmark. The model is required to generate a 3x3 grid of plots to compare ECG signal distributions, showcasing the benchmark's focus on sophisticated, multi-panel figures.

**Example 2 - Correlation**

**Instruction**

Create a comprehensive 3x3 subplot grid analyzing the relationship between economic indicators and Olympic performance across different regions. Each subplot should be a composite visualization combining multiple chart types:

Top row: (1) Scatter plot with regression line showing GDP vs Total Medals, overlaid with bubble sizes representing population and color-coded by region, plus marginal histograms; (2) Correlation heatmap of all numerical variables (gold, silver, bronze, total, gdp, population) with annotated correlation coefficients; (3) Scatter plot matrix (pair plot) of GDP, Total Medals, and Population with different colors for each region.

Middle row: (4) Box plots showing Total Medals distribution by region overlaid with individual data points as a strip plot; (5) Violin plots of GDP distribution by region combined with box plots inside; (6) Scatter plot of GDP vs Gold medals with best-fit lines for each region separately, including confidence intervals.

Bottom row: (7) Bubble chart showing Silver vs Bronze medals where bubble size represents GDP and colors represent regions, with trend lines; (8) Hexbin plot (2D histogram) of GDP vs Total Medals overlaid with scatter points colored by region; (9) Parallel coordinates plot showing standardized values of GDP, Population, Gold, Silver, Bronze, and Total medals with lines colored by region.

**Files**

📄 olympics.csv

**Reference Image**

*Figure 7.* An example of a complex correlation task from the PlotCraft benchmark. The prompt requires a 3x3 subplot grid to visualize the relationship between economic indicators and Olympic performance, testing the model's ability to generate detailed, multi-faceted comparative analyses.

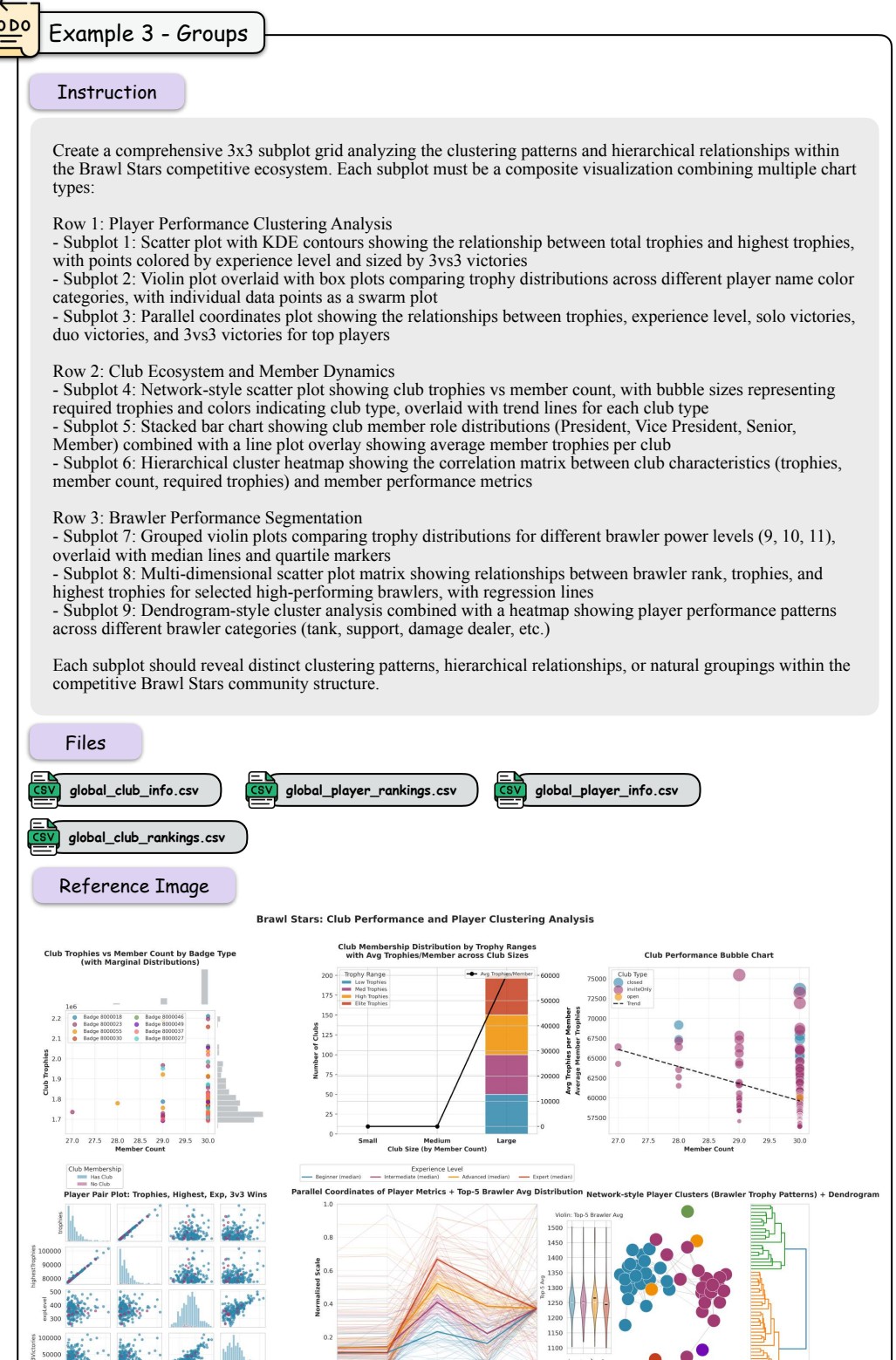

*Figure 8.* An example of a complex Groups visualization task. The model must generate a 3x3 grid to display clustering and hierarchical structures, testing its ability to render complex relational plots like dendrograms or network graphs.

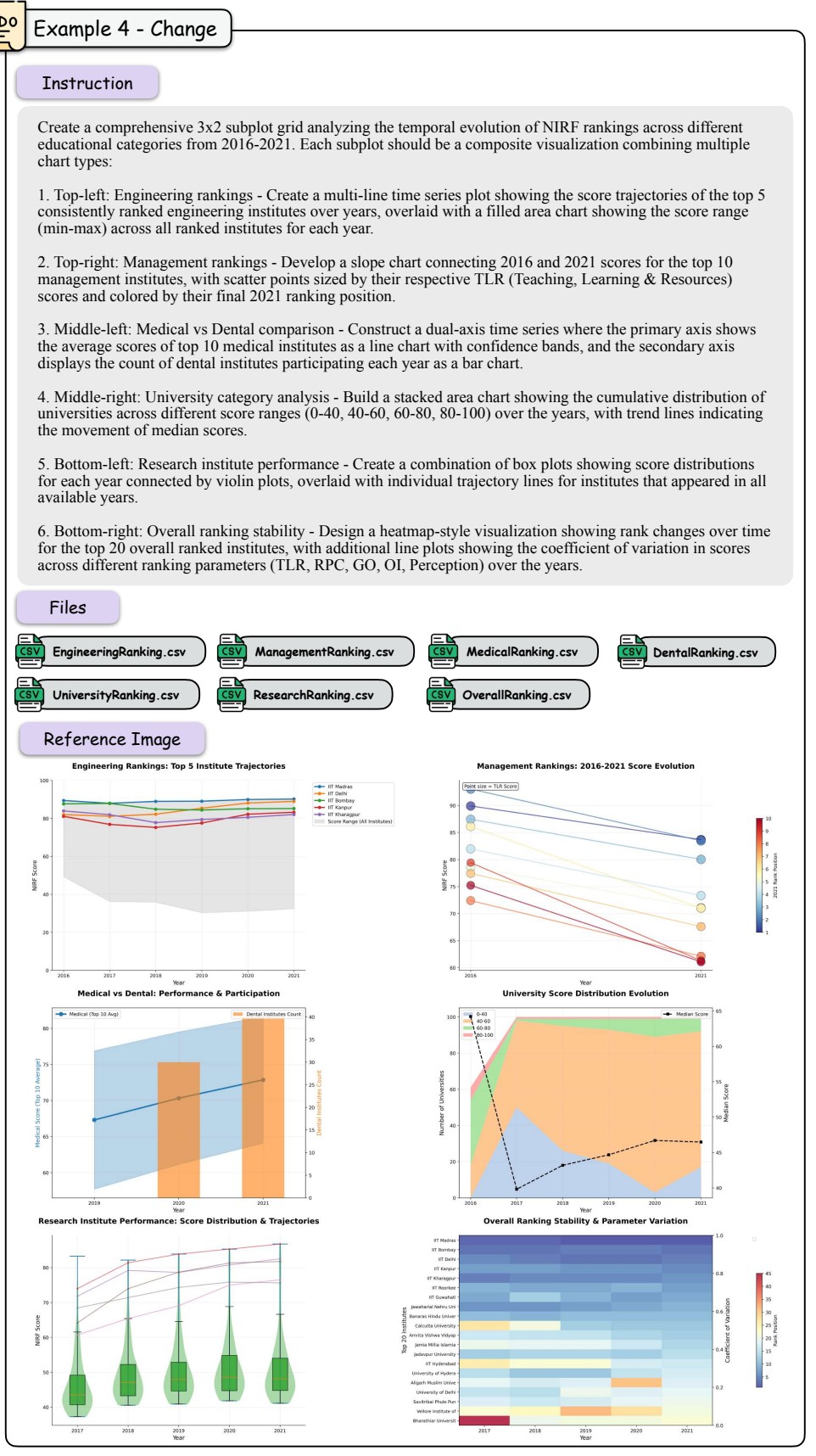

*Figure 9.* An example of a Change visualization task from the benchmark. This prompt requires a 3x2 subplot grid to track the temporal evolution of rankings, evaluating the model's ability to create detailed time-series visualizations.

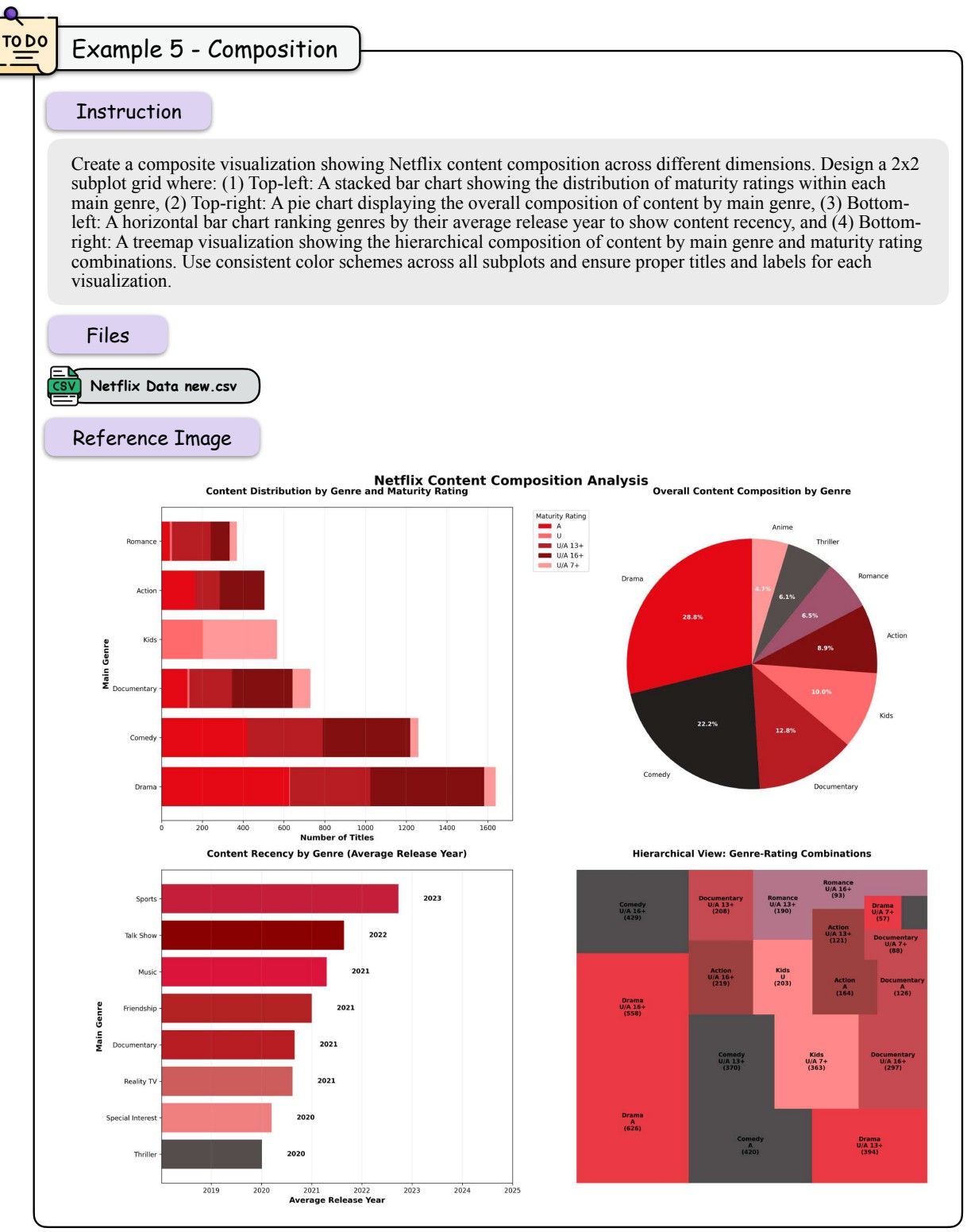

*Figure 10.* A Composition task example requiring the model to break down a dataset into its constituent parts. The instruction to use a 2x2 grid tests the ability to create comparative part-to-whole visualizations across different categories.

**TO DO**

**Example 6 - Ranking**

**Instruction**

Create a comprehensive ranking visualization that compares Udemy course performance across different metrics. Design a 2x2 subplot grid where each subplot combines multiple visualization elements:

1. Top-left: Create a horizontal bar chart showing the top 15 courses by number of subscribers, with bars colored by rating (use a color gradient from red for lower ratings to green for higher ratings). Overlay scatter points at the end of each bar to show the number of reviews.

2. Top-right: Build a lollipop chart displaying the top 12 courses by rating, where the stem length represents the rating value and the circle size represents the number of subscribers. Color-code the circles by instructional level.

3. Bottom-left: Design a stacked horizontal bar chart showing the top 10 instructors by total subscribers across all their courses. Each bar should be segmented to show the contribution of individual courses, with different colors for each course.

4. Bottom-right: Create a slope chart comparing the top 8 courses, showing the relationship between their ranking by subscribers (left side) versus their ranking by number of reviews (right side). Connect corresponding courses with lines, using different line styles or colors to highlight courses that maintain similar rankings versus those that change significantly.

Include appropriate titles, labels, and legends for each subplot. Use consistent color schemes and ensure all text is readable.

**Files**

**CSV** udemy_courses.csv

**Reference Image**

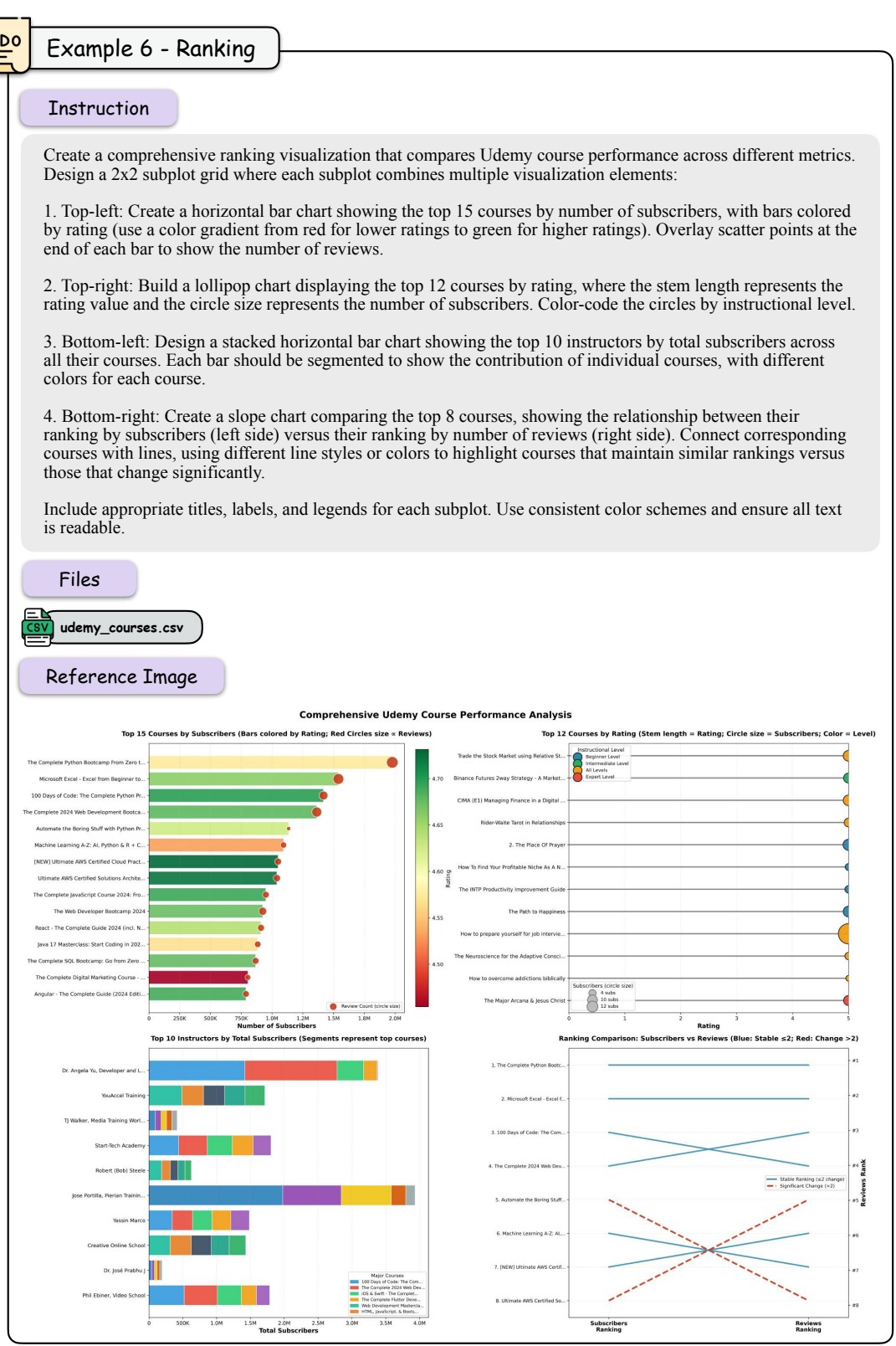

*Figure 11.* An example of a Ranking visualization task. The model is required to generate a 2x2 grid to compare and rank items across multiple metrics, assessing its ability to create clear and effective comparative bar charts or ordered plots.

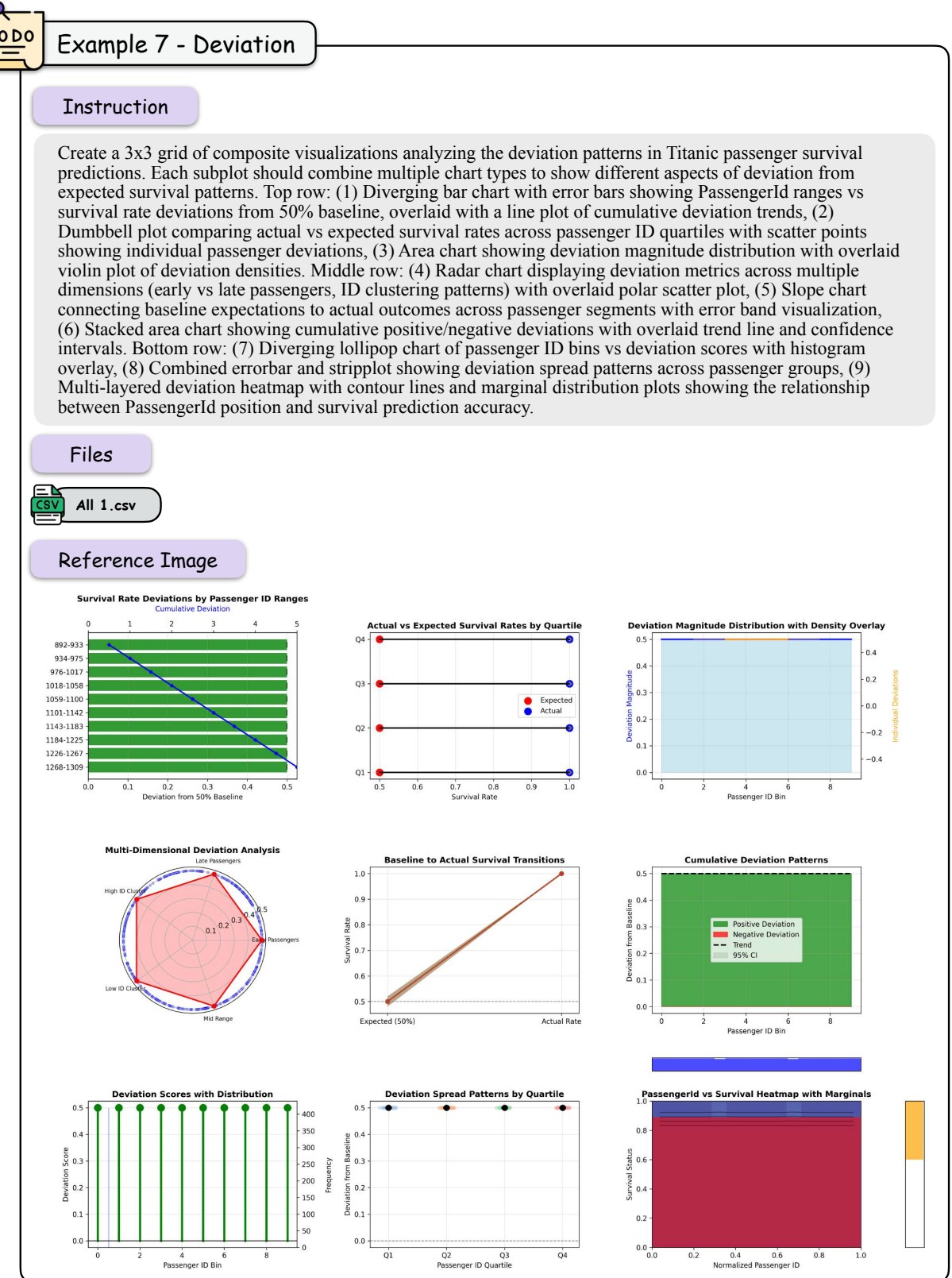

*Figure 12.* A Deviation task from the benchmark, requiring the analysis of prediction errors. The instruction to create a 3x3 grid tests the model's ability to visualize variance and differences, such as in divergence bars or error plots.

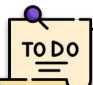

## Difficulty Comparison - Part 1

### Dataset Name

aniruddha1995/firethorn-10-world-co2-emission-analysis

### Easy Instruction

Create a line chart showing the evolution of CO2 emissions per capita over time from 1990 to 2020 for all four countries in the dataset. Extract and clean the CO2 emissions per capita data from the appropriate series, handle missing values by interpolation where reasonable, and display each country as a separate colored line with proper labels, legend, and title.

### Medium

Create a composite visualization showing the temporal evolution of CO2 emissions across different countries. Design a subplot layout with two complementary charts: (1) a line chart displaying CO2 emissions trends over time (1990-2020) for the top 3 countries by total emissions, and (2) a stacked area chart showing the cumulative contribution of these same countries to global CO2 emissions over the same time period. Both charts should highlight how emission patterns have changed over the three-decade span and reveal which countries have been the dominant contributors to global CO2 emissions.

### Hard Instruction

Create a comprehensive 3x3 subplot grid analyzing the temporal evolution of CO2 emissions and environmental indicators across different countries from 1990-2020. Each subplot must be a composite visualization combining multiple chart types:

Top row: (1) Line chart with confidence bands showing CO2 emissions per capita trends overlaid with scatter points for key milestone years, (2) Stacked area chart showing composition of different emission sources with trend lines for total emissions, (3) Dual-axis plot combining bar chart of annual emission changes with line plot of cumulative emissions.

Middle row: (4) Time series decomposition plot showing seasonal patterns in methane emissions with moving averages and trend components, (5) Slope chart connecting 1990 and 2020 values with intermediate milestone markers overlaid on a background heatmap of emission intensity, (6) Multi-line time series with error bands comparing agricultural vs energy sector emissions with filled areas showing the gap between them.

Bottom row: (7) Calendar heatmap of monthly emission data overlaid with box plots showing quarterly distributions, (8) Autocorrelation plot combined with partial autocorrelation analysis showing emission pattern dependencies over time, (9) Cross-correlation matrix heatmap between different emission types with time-lagged correlation coefficients displayed as a network graph overlay.

Each subplot must include country-specific trend analysis, statistical significance indicators, and highlight periods of major environmental policy changes or economic events that influenced emission patterns.

### Files

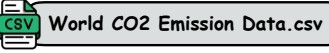 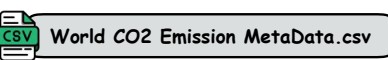

*Figure 13.* Example tasks of varying complexity levels from the PlotCraft benchmark. For a single dataset (`firethorn-10-world-co2-emission-analysis`), this figure shows the raw data and the natural language instructions for an Easy, Medium, and Hard task.

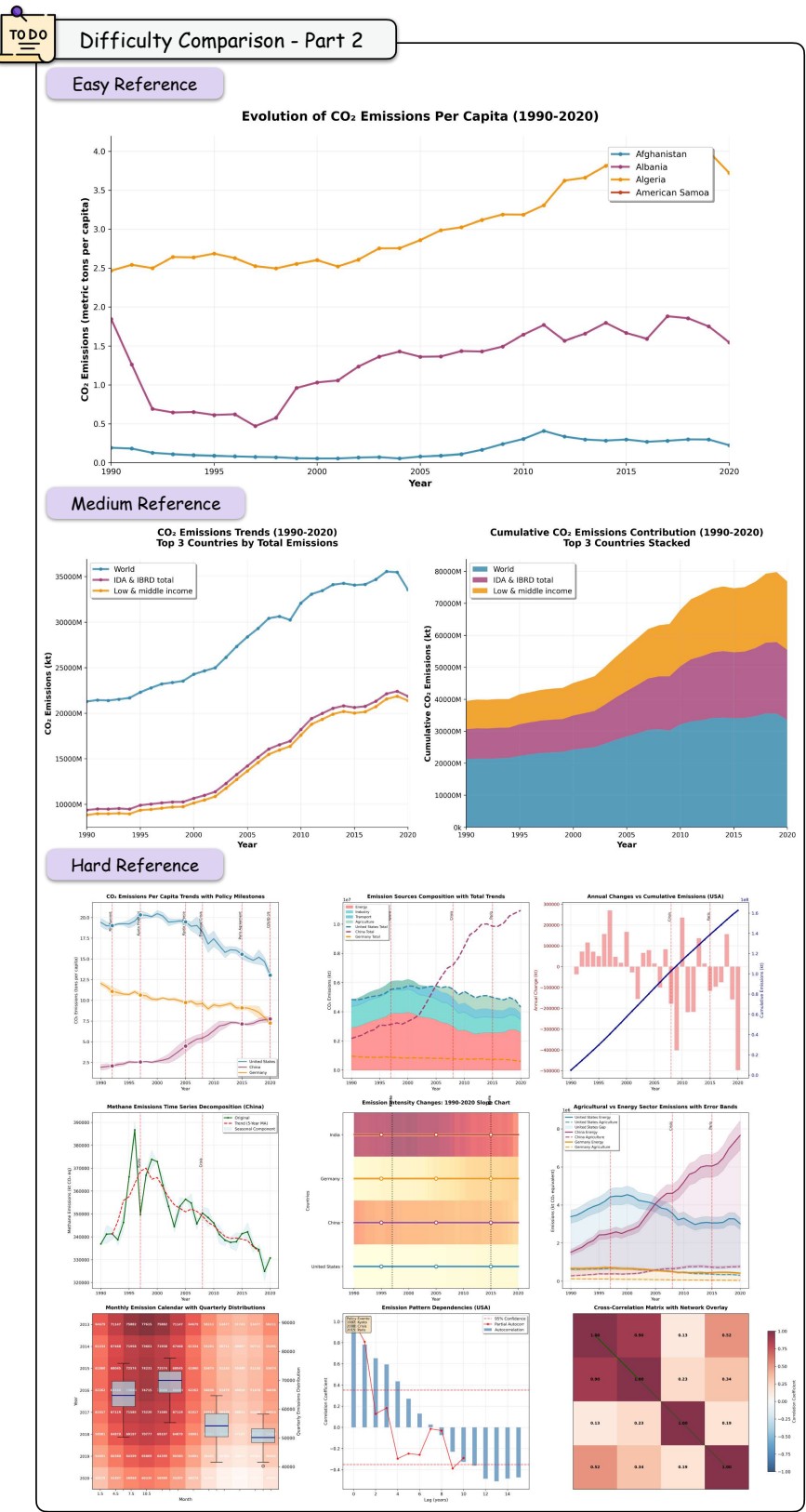

*Figure 14.* Reference images corresponding to the three task instructions shown in Figure 13. These images represent the ground-truth visualizations for the Easy, Medium, and Hard tasks, respectively, illustrating the expected output for each complexity level.

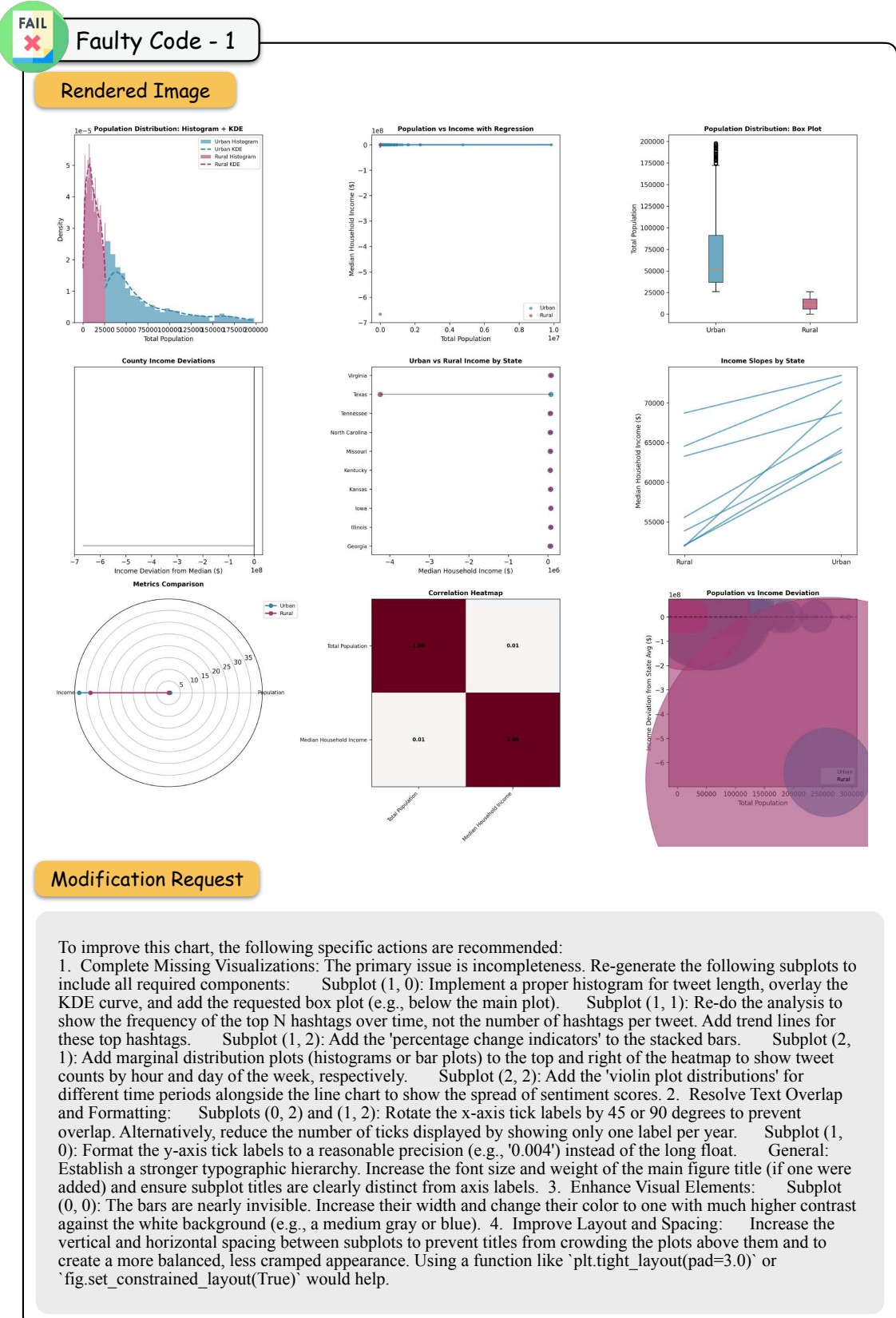

**Faulty Code - 1**

**Rendered Image**

**Modification Request**

To improve this chart, the following specific actions are recommended:
1. Complete Missing Visualizations: The primary issue is incompleteness. Re-generate the following subplots to include all required components: Subplot (1, 0): Implement a proper histogram for tweet length, overlay the KDE curve, and add the requested box plot (e.g., below the main plot). Subplot (1, 1): Re-do the analysis to show the frequency of the top N hashtags over time, not the number of hashtags per tweet. Add trend lines for these top hashtags. Subplot (1, 2): Add the 'percentage change indicators' to the stacked bars. Subplot (2, 1): Add marginal distribution plots (histograms or bar plots) to the top and right of the heatmap to show tweet counts by hour and day of the week, respectively. Subplot (2, 2): Add the 'violin plot distributions' for different time periods alongside the line chart to show the spread of sentiment scores. 2. Resolve Text Overlap and Formatting: Subplots (0, 2) and (1, 2): Rotate the x-axis tick labels by 45 or 90 degrees to prevent overlap. Alternatively, reduce the number of ticks displayed by showing only one label per year. Subplot (1, 0): Format the y-axis tick labels to a reasonable precision (e.g., '0.004') instead of the long float. General: Establish a stronger typographic hierarchy. Increase the font size and weight of the main figure title (if one were added) and ensure subplot titles are clearly distinct from axis labels. 3. Enhance Visual Elements: Subplot (0, 0): The bars are nearly invisible. Increase their width and change their color to one with much higher contrast against the white background (e.g., a medium gray or blue). 4. Improve Layout and Spacing: Increase the vertical and horizontal spacing between subplots to prevent titles from crowding the plots above them and to create a more balanced, less cramped appearance. Using a function like `plt.tight_layout(pad=3.0)` or `fig.set_constrained_layout(True)` would help.

*Figure 15.* A multi-turn task example with multiple errors. Subplot (1, 0) fails to implement the required histogram, KDE curve, and boxplot. Furthermore, subplot (2, 2) exhibits severe rendering artifacts, including element overlap and content extending beyond the subplot boundaries.

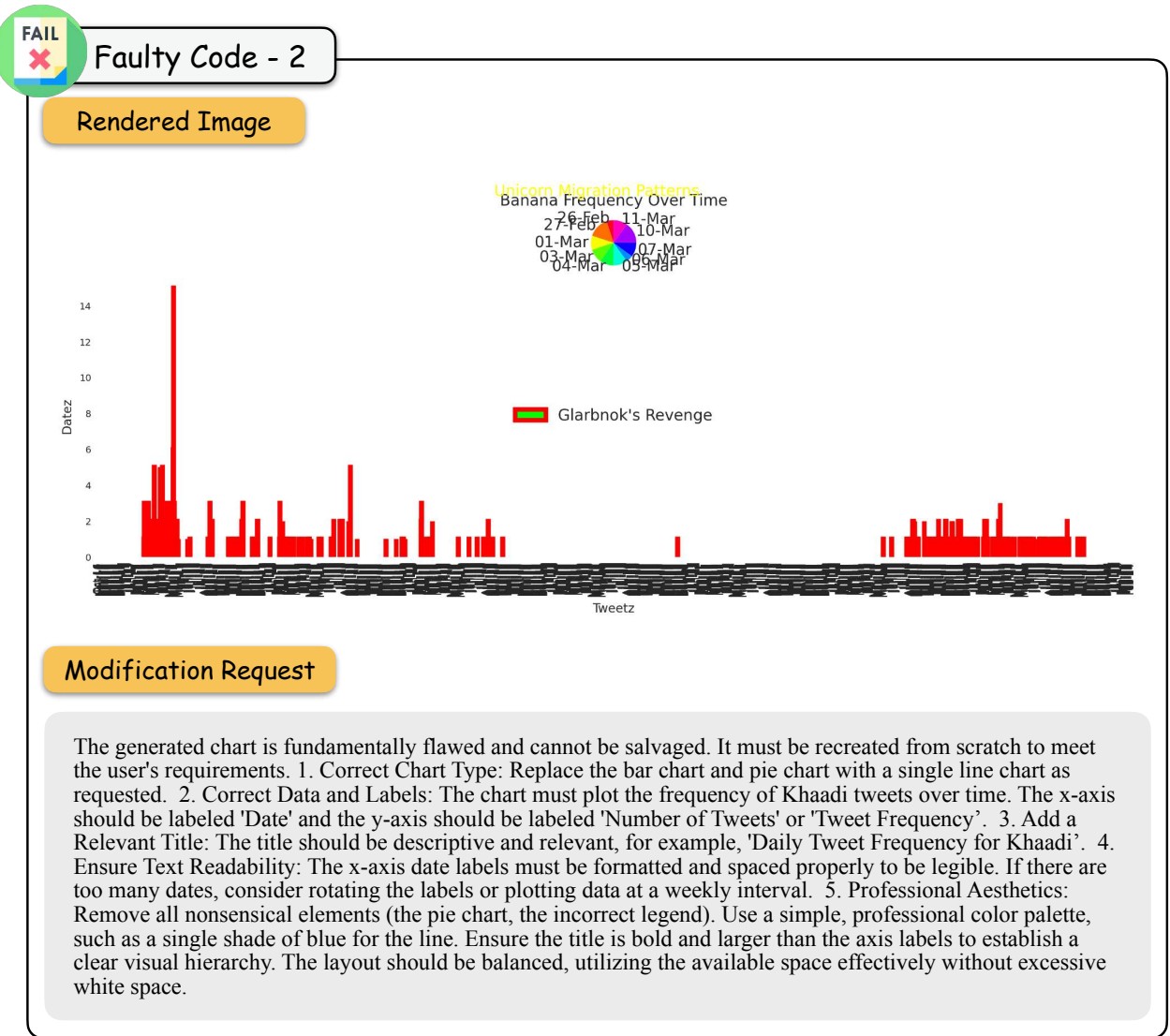

*Figure 16.* An example of incorrect chart type and poor aesthetics. The generated code produces pie charts, which violates the instruction, and utilizes a color palette with excessively high contrast, diminishing the visual quality.

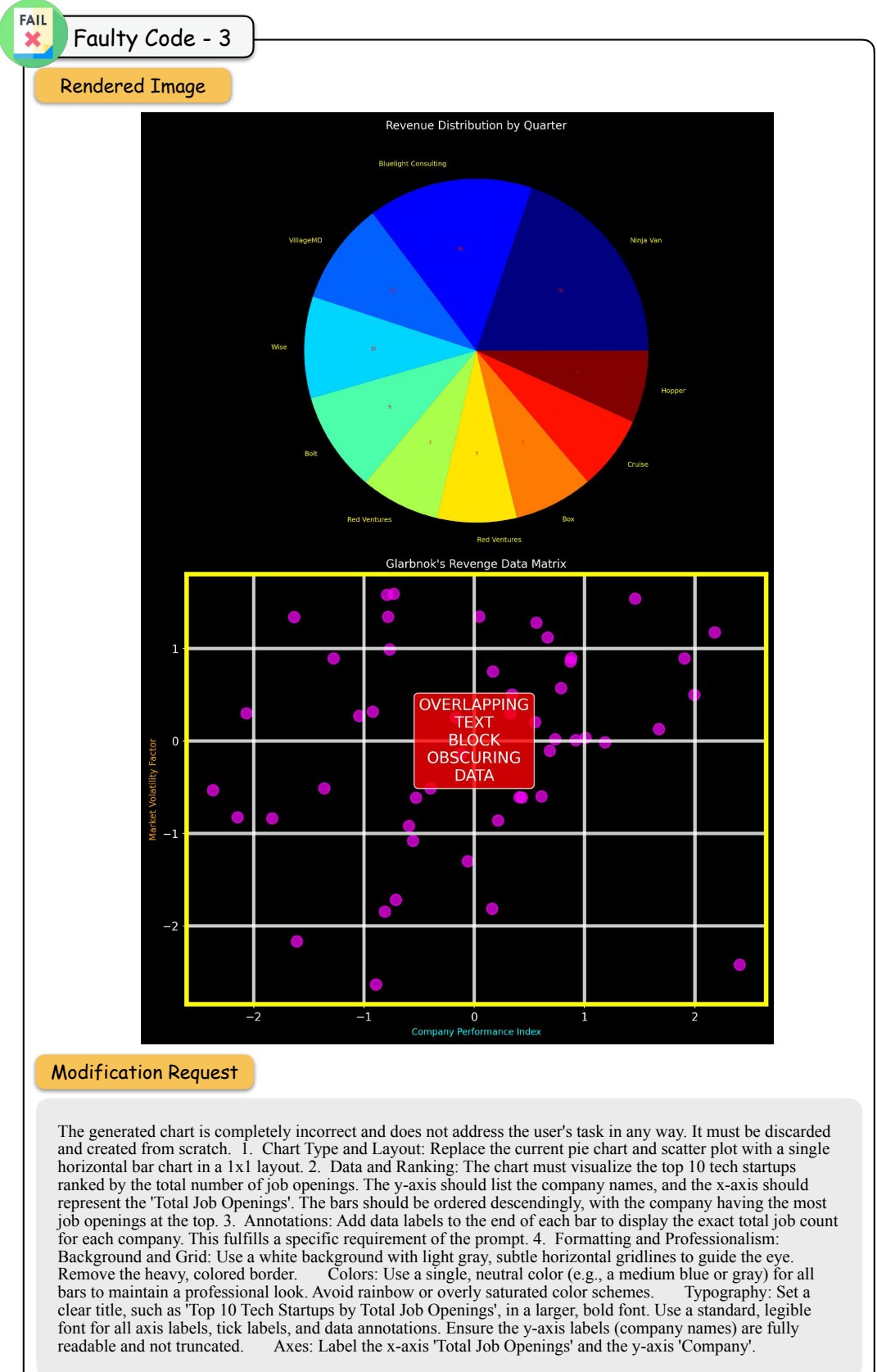

**Modification Request**

The generated chart is completely incorrect and does not address the user's task in any way. It must be discarded and created from scratch. 1. Chart Type and Layout: Replace the current pie chart and scatter plot with a single horizontal bar chart in a 1x1 layout. 2. Data and Ranking: The chart must visualize the top 10 tech startups ranked by the total number of job openings. The y-axis should list the company names, and the x-axis should represent the 'Total Job Openings'. The bars should be ordered descendingly, with the company having the most job openings at the top. 3. Annotations: Add data labels to the end of each bar to display the exact total job count for each company. This fulfills a specific requirement of the prompt. 4. Formatting and Professionalism: Background and Grid: Use a white background with light gray, subtle horizontal gridlines to guide the eye. Remove the heavy, colored border.     Colors: Use a single, neutral color (e.g., a medium blue or gray) for all bars to maintain a professional look. Avoid rainbow or overly saturated color schemes.     Typography: Set a clear title, such as 'Top 10 Tech Startups by Total Job Openings', in a larger, bold font. Use a standard, legible font for all axis labels, tick labels, and data annotations. Ensure the y-axis labels (company names) are fully readable and not truncated.     Axes: Label the x-axis 'Total Job Openings' and the y-axis 'Company'.

*Figure 17.* An example of incorrect layout and suboptimal color choice. The code fails to adhere to the specified 1x1 grid layout, and the visualization suffers from an oversaturated color scheme that compromises professional appearance.

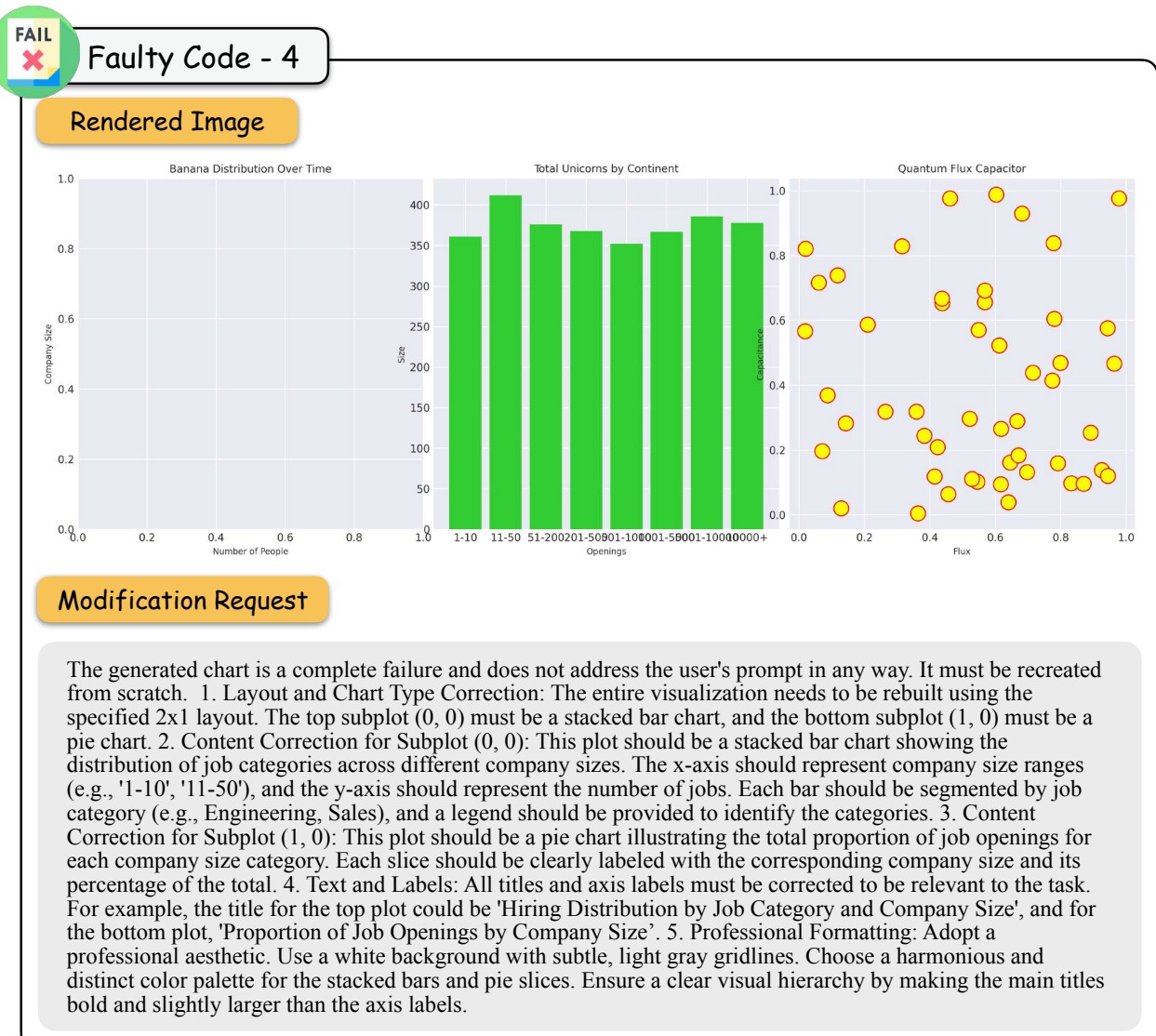

*Figure 18.* An example where chart types are correct but formatting is poor. The visualization suffers from overlapping x-axis tick labels, which impairs readability, and uses a default gray background that lacks professional polish.

# C. Evaluation Metrics

This section details the comprehensive prompt in Figure C, C, C, C and scoring rubric used to evaluate the generated visualizations. Our evaluation is structured around two primary dimensions: **Task Compliance** and **Chart Quality**, each with a set of granular sub-metrics as defined below.

## 1. Task Compliance (Binary Scoring: 0/1)

For each criterion, a binary score is assigned: 1 for compliance (requirement met) or 0 for non-compliance (requirement not met).

1. **Layout Compliance:**
   - *Question:* Does the chart follow the required layout specification (e.g., 1x1, 2x2)?
   - *Score:* 1 if the layout matches the requirement exactly; 0 otherwise.

2. **Chart Type Compliance:**
   - *Question:* Does the chart use the correct specified chart type(s)?
   - *Score:* 1 if all chart types match the requirement; 0 otherwise.

3. **Visualization Requirement Fulfillment:**
   - *Question:* Does the chart fulfill the core visualization goal (e.g., showing a relationship, displaying a trend)?
   - *Score:* 1 if the core visualization requirement is met; 0 otherwise.

4. **Complete Task Fulfillment:**
   - *Question:* Are all specified requirements from the task instruction completed?
   - *Score:* 1 if all requirements are fulfilled; 0 if any requirement is missing.

## 2. Chart Quality (3-Level Scoring: 0/1/2)

For each criterion, a score from 0 to 2 is assigned based on the following rubric.

1. **Clarity (No Overlap):**

   **Score 2** No overlap exists between any elements; all components are clearly separated.
   **Score 1** Minor overlap exists but does not significantly impact readability.
   **Score 0** Significant overlap exists, severely affecting the chart's readability.

2. **Layout Quality:**

   **Score 2** Excellent layout with well-proportioned elements and optimal spacing.
   **Score 1** Good layout with acceptable element sizes and spacing, but with minor imperfections.
   **Score 0** Poor layout with inappropriately sized, cramped, or poorly spaced elements.

3. **Color Quality:**

   **Score 2** Excellent, harmonious color scheme with appropriate contrast and visual appeal.
   **Score 1** Good color scheme with acceptable harmony and contrast, but with minor issues.
   **Score 0** Poor color scheme with clashing, harsh, or overly dull colors.

4. **Text Readability:**

   **Score 2** All text content is correct, appropriately sized, and clearly legible.
   **Score 1** Text contains minor issues (e.g., small font size, minor typos) that do not significantly impair understanding.
   **Score 0** Text has significant correctness or legibility issues (e.g., wrong labels, unreadable font).

5. **Professional Formatting:**

**Score 2 (Publication-Ready)**  The chart is highly professional and polished, adhering to formal publication standards (e.g., white background, subtle gridlines, clear typographic hierarchy).

**Score 1 (Needs Revision)**  The chart is functional but lacks professional refinement. It may use a plain default style, have a weak visual hierarchy, or heavy borders.

**Score 0 (Unacceptable)**  The chart uses a non-standard, themed style inappropriate for formal contexts (e.g., non-white background, high-contrast gridlines).

**Prompt C.1: Evaluation Prompt - Part 1**

```
**Role and Goal:**
You are a meticulous data visualization quality inspector. Your task is to
    evaluate a generated chart based on a user's task prompt and the chart's
    visual properties. The evaluation is divided into two main categories: Task
     Compliance and Chart Quality.

**Task Prompt Format:**
The user's request follows this format:
```
# Task
Category: {task_category}
Instruction: {task_instruction}
```

**Evaluation Categories:**

## 1. Task Compliance (Binary Scoring: 0/1)
For each criterion, provide a binary score: '1' for compliance (requirement met
    ) or '0' for non-compliance (requirement not met).

1. **Layout Compliance:**
   - **Question**: Does the chart follow the required layout specification (e.g
     ., 1x1, 2x2, 2x3)?
   - **Score**: '1' if the layout matches the requirement exactly. '0' if the
     layout differs from what was specified.

2. **Chart Type Compliance:**
   - **Question**: Does the chart use the correct chart type as specified (e.g
     ., bar chart, line chart, scatter plot)?
   - **Score**: '1' if the chart type matches the requirement. '0' if a
     different chart type was used.

3. **Visualization Requirement Fulfillment:**
   - **Question**: Does the chart fulfill the specific visualization
     requirement (e.g., showing relationship between two variables,
     displaying trends over time)?
   - **Score**: '1' if the core visualization requirement is met. '0' if the
     requirement is not addressed.

4. **Complete Task Fulfillment:**
   - **Question**: Are all specified requirements from the task instruction
     completed?
   - **Score**: '1' if all requirements are fulfilled. '0' if any requirement
     is missing or incomplete.
```

**Prompt C.2: Evaluation Prompt - Part 2**

```
## 2. Chart Quality (3-Level Scoring: 0/1/2)
For each criterion, provide a score from 0-2 with detailed explanations for
    each level.

1. **Clarity (No Overlap):**
    - **Score 2**: No overlap exists between any elements; all subplots, titles,
        axis labels, tick marks, legends, and text boxes are clearly separated.
    - **Score 1**: Minor overlap between text boxes or legends with plot content
        (data points, lines, bars) or border lines, but doesn't significantly
        impact readability.
    - **Score 0**: Significant overlap between subplots, titles, axis labels,
        tick marks, or other text elements that severely affects readability.

2. **Layout Quality:**
    - **Score 2**: Excellent layout with well-proportioned elements, optimal
        spacing, balanced white space distribution, and outstanding overall
        visual appeal.
    - **Score 1**: Good layout with reasonable element sizes and spacing,
        acceptable visual balance with minor imperfections.
    - **Score 0**: Poor layout with inappropriately sized elements, cramped or
        excessive spacing, unbalanced composition, or unappealing visual
        presentation.

3. **Color Quality:**
    - **Score 2**: Excellent color scheme with harmonious palette, appropriate
        contrast, visually appealing combinations, and effective use of distinct
        colors for differentiation.
    - **Score 1**: Good color scheme with acceptable harmony and contrast, minor
        issues that don't significantly impact aesthetics.
    - **Score 0**: Poor color scheme with clashing colors, excessive harsh
        contrasts, overly dull/muted colors, or ineffective use of similar
        colors that lack distinction.

4. **Text Clarity:**
    - **Score 2**: All text content is correct and appropriate, including
        accurate axis labels, proper titles, correct tick mark text, clear
        legends, and accurate text box content.
    - **Score 1**: Most text content is correct with minor issues that don't
        significantly impair understanding or convey wrong information.
    - **Score 0**: Text content has significant correctness issues including
        incorrect axis labels, inappropriate titles, wrong tick marks, unclear
        legends, or inaccurate text box content.
```

**Prompt C.3: Evaluation Prompt - Part 3**

```
5. **Formatting and Professional Standards:**
  - **Score 2**: (Excellent / Publication-Ready): The chart's formatting is
     highly professional, clean, and adheres strictly to formal publication
     standards. It appears polished and intentionally designed, not like a
     default software output. Key characteristics include:
    - A white background is used.
    - Gridlines, if present, are subtle (thin, light gray) and do not distract
       from the data.
    - A clear typographic hierarchy is established, with the main title being
       visually distinct (e.g., bolded and/or larger) from axis labels and
       other text.
    - All non-data elements (axes, ticks) are appropriately weighted and do not
        appear heavy or clumsy.
  - **Score 1**: (Good / Needs Revision): The chart is functional but lacks
     professional refinement and contains minor stylistic issues. It is clear
     but would require formatting adjustments before formal publication. Key
     characteristics include:
    - The background is white, but the overall aesthetic is plain or unpolished
       .
    - The title lacks emphasis (e.g., is not bolded), making the visual
       hierarchy weak.
    - It may have slightly heavy chart borders (spines) or default styling that
        feels more like a "first draft" than a final product.
  - **Score 0**: (Poor / Unacceptable for Formal Use): The chart uses a non-
     standard, themed style that is inappropriate for academic or professional
      contexts. It is immediately identifiable as a default output from an
     analysis tool. Key characteristics include:
    - It features a non-white background (e.g., gray, beige).
    - It employs high-contrast, distracting elements like white gridlines on a
       colored background.
    - The overall visual style is cluttered or stylized in a way that detracts
        from a formal, serious tone.
```

**Prompt C.4: Evaluation Prompt - Part 4**

```
**Output Format:**
Your response **MUST** be a single, valid JSON object, without any additional
    text before or after it. Use the exact structure below:

```json
{
  "task_compliance": {
    "layout_compliance": {
      "score": <0_or_1>,
      "reason": "<State whether the layout matches requirements and specify
          deviations if score is 0.>"
    },
    "chart_type_compliance": {
      "score": <0_or_1>,
      "reason": "<State whether the chart type matches requirements and specify
          what was expected vs. actual if score is 0.>"
    },
    "visualization_requirement_fulfillment": {
      "score": <0_or_1>,
      "reason": "<State whether the core visualization requirement is met and
          specify what's missing if score is 0.>"
    },
    "complete_task_fulfillment": {
      "score": <0_or_1>,
      "reason": "<State whether all requirements are completed and list missing
          items if score is 0.>"
    }
  },
  "chart_quality": {
    "clarity_no_overlap": {
      "score": <0_1_or_2>,
      "reason": "<Describe the overlap situation and justify the score level.>"
    },
    "layout_quality": {
      "score": <0_1_or_2>,
      "reason": "<Evaluate element sizing, spacing, and overall visual balance
          .>"
    },
    "color_quality": {
      "score": <0_1_or_2>,
      "reason": "<Assess color harmony, contrast, and aesthetic appeal.>"
    },
    "text_clarity": {
      "score": <0_1_or_2>,
      "reason": "<Evaluate text readability, correctness, and positioning.>"
    },
    "formatting_and_professional_standards": {
      "score": <0_1_or_2>,
      "reason": "Evaluate formatting and professional standards.>"
    }
  }
}
```
```

## C.1. LLM Judge Cases

**Case - 1**  This case, presented in Figure 19, illustrates a complete evaluation performed by our Gemini-2.5-Pro judge, including the original instruction, the model-generated image, and the resulting scores. The example is notable because the generated visualization is of high aesthetic quality but exhibits significant Task Compliance failures. This highlights the importance of decoupling the evaluation of quality from correctness.

Specifically, several subplots do not adhere to the prompt's requirements:

- Subplot (1,0) is incorrectly implemented as a scatter plot instead of the required dumbbell plot.

- Subplot (1,1) is rendered as a line chart with error bands rather than the specified area chart.

- Subplots (0,0) and (0,1) fail to meet the "diverging" chart criteria, as they employ a monochromatic color scheme that does not differentiate between positive and negative values.

**Case - 2**  This case, presented in Figure 20, illustrates a generated visualization with a cascade of failures across multiple evaluation criteria, rendering it both incorrect and uninterpretable. The primary issues are categorized below.

- **Chart Type Non-Compliance:** The model fails to implement the specified chart types correctly. In subplot (1, 0), the bubbles are of uniform size and do not encode the transaction amount as required for a proper bubble chart. In subplot (2, 0), the treemap omits the required embedded bar charts, merely subdividing the areas instead.

- **Severe Element Overlap:** The visualization suffers from pervasive element occlusion that severely impacts readability. This includes illegible, overlapping x-axis tick labels in subplots (1, 0) and (2, 2); a legend in subplot (2, 1) that obstructs data points; and a dendrogram in subplot (2, 2) that clashes with its corresponding heatmap.

- **Inconsistent Color Scheme:** The use of color is critically flawed and misleading. In the first row, 'Fraud' is represented by red, but this is reversed in the parallel coordinates plot (1, 1), where 'Fraud' is incorrectly labeled as light blue. This semantic inconsistency makes the chart actively deceptive.

- **Poor Text Readability:** The chart has significant text-related issues. Multiple axis labels are illegible due to overlap, text within the treemap (2, 0) is too small to read, and some axes use raw, unformatted variable names (e.g., 'INIT_BALANCE'), which is unprofessional.

**Case - 3**  This case, presented in Figure 21, showcases a generated visualization that is highly successful in terms of both task compliance and overall chart quality. Its sole deficiency lies in the Professional Formatting sub-metric. The chart utilizes a non-standard, themed style that is inappropriate for formal publication; it features a gray background with high-contrast white gridlines, which is characteristic of a default software output (e.g., from Seaborn) rather than a polished, professional graphic. For publication-ready figures, a clean white background with subtle, non-distracting gridlines is the expected standard. This example underscores the importance of evaluating not just correctness, but also the fine-grained stylistic details that separate a functional plot from a professional one.

**Case - 4**  This case, presented in Figure 22, highlights how an otherwise high-quality visualization can be penalized for subtle but important flaws in its layout and clarity. While the chart successfully fulfills the core task requirements, its final score is reduced due to two specific issues. First, the layout is suboptimal, with excessive vertical white space between the main figure title and the subplot grid, creating a disjointed appearance. Second, the chart suffers from a minor clarity issue, as several x-axis tick labels exhibit slight overlap, which can impede readability. This example demonstrates the importance of meticulous polishing, a nuance that even capable models can overlook.

# D. Additional Results

Table 7 presents the complete quantitative results for all 24 evaluated LLMs on the PlotCraft benchmark. The data reinforces the findings from our main analysis and further highlights the strong performance of our model.

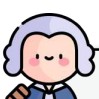

## Judgement - 1

### Instruction

Create a comprehensive pricing deviation analysis across multiple e-commerce platforms by combining data from three key files. Design a 2x2 subplot grid where each subplot shows the deviation of platform-specific MRPs from a baseline reference price. In the top-left, create a diverging bar chart showing how Amazon MRP deviates from the average MRP across all platforms for each product category. In the top-right, display a diverging lollipop chart comparing Myntra MRP deviations from the Final MRP Old baseline. In the bottom-left, construct a dumbbell plot showing the price range between the lowest platform MRP and highest platform MRP for each Style ID, highlighting the pricing spread. In the bottom-right, create an area chart with error bands showing the deviation of actual sale amounts from expected amounts (based on MRP) over different order statuses, using the Amazon sales data. Each subplot should use different colors to distinguish positive and negative deviations, and include reference lines at zero deviation.

### Generation

**Comprehensive Pricing Deviation Analysis Across E-commerce Platforms**

### Judgement

| | | | |
|---|---|---|---|
| **Layout Compliance** | 1 | **Clarity** | 2 |
| **Chart Type Compliance** | 0 | **Layout Quality** | 1 |
| **Requirement Fulfillment** | 0 | **Color Quality** | 2 |
| **Complete Task Fulfillment** | 0 | **Text Readability** | 2 |
| | | **Professional Formatting** | 1 |

38

*Figure 19.* A complete evaluation case from our Gemini-2.5-Pro judge, displaying the original instruction, a generated image, and the final scores. This example highlights a common failure mode where a model produces a high-quality, aesthetically pleasing chart that

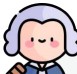

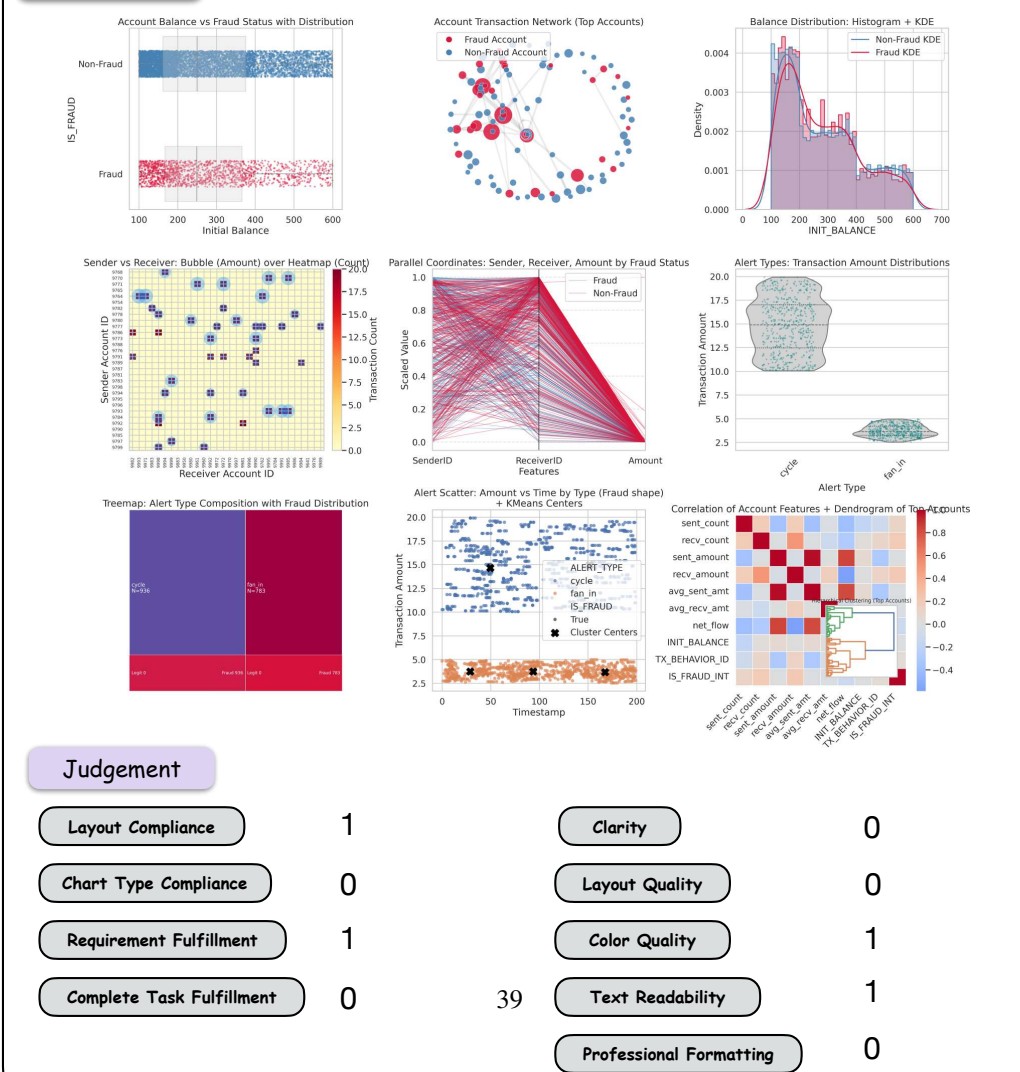

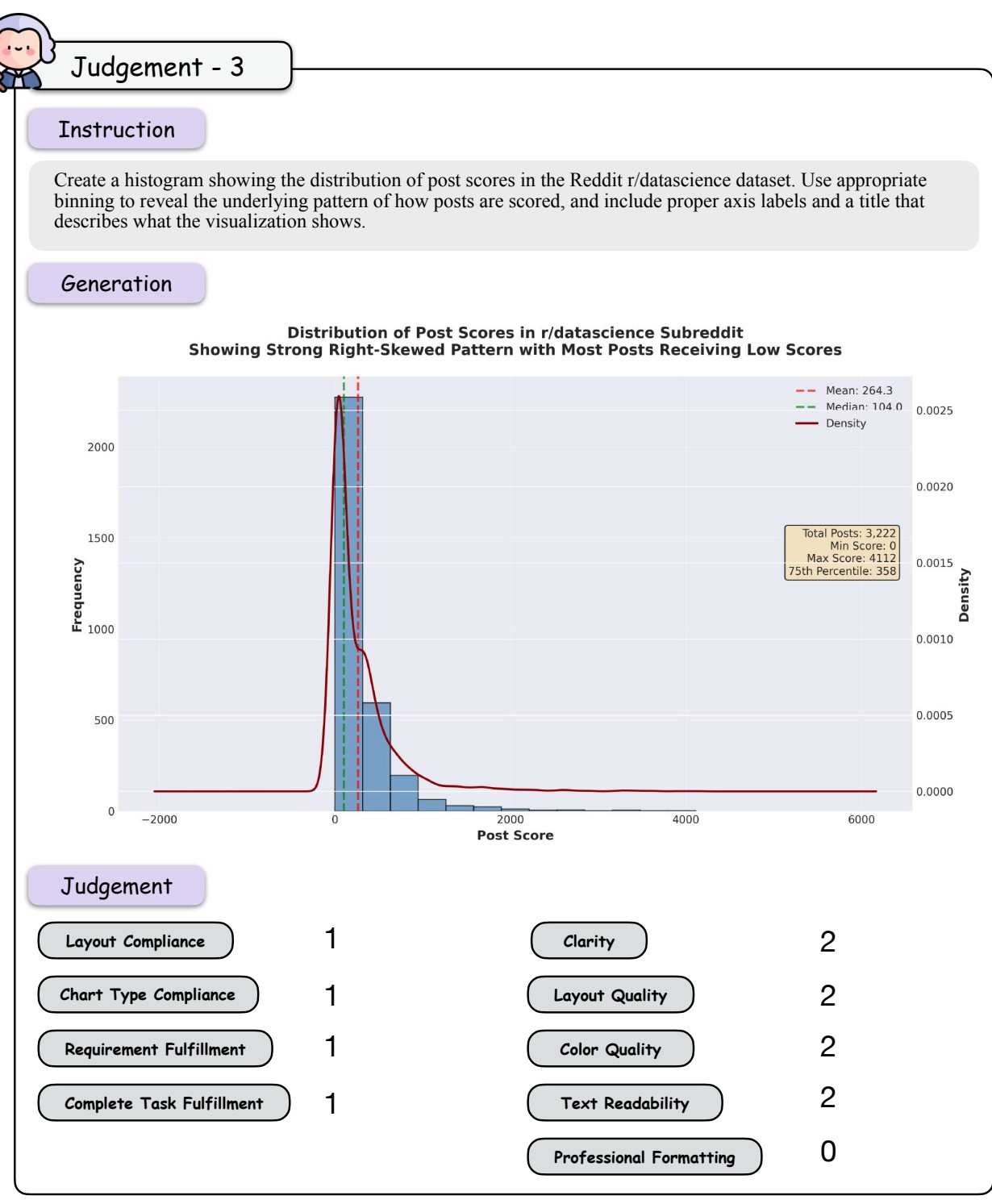

*Figure 21.* An example of a high-quality visualization that is penalized for a lack of professional formatting. While the chart correctly adheres to all task requirements, its use of a default gray background and high-contrast gridlines prevents it from meeting publication-ready standards.

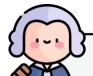

## Judgement - 4

### Instruction

Create a comprehensive 3x3 subplot grid analyzing university performance clusters and regional patterns across Asian countries. Each subplot should be a composite visualization: (1) Top-left: Scatter plot with Academic Reputation vs Employer Reputation, overlaid with country-based color coding and bubble sizes representing Overall Score, plus marginal histograms showing distribution of each metric; (2) Top-center: Stacked bar chart showing count of universities by country, overlaid with a line plot showing average Overall Score per country; (3) Top-right: Box plot showing Citations per Paper distribution by country, overlaid with violin plots to show density distributions; (4) Middle-left: Radar chart comparing average performance metrics (Academic Reputation, Employer Reputation, International Students, International Faculty, Faculty Student Ratio) for top 5 countries by university count; (5) Middle-center: Heatmap showing correlation matrix of all numerical performance metrics, overlaid with hierarchical clustering dendrogram; (6) Middle-right: Parallel coordinates plot showing performance profiles of top 20 universities across key metrics (Academic Reputation, Employer Reputation, Citations per Paper, International Students), with lines colored by country; (7) Bottom-left: Treemap showing university count by country and city hierarchy, with cell sizes representing total universities and colors representing average Overall Score; (8) Bottom-center: Network graph showing country relationships based on similarity in performance metrics, with node sizes representing university count and edge weights representing similarity scores; (9) Bottom-right: Cluster scatter plot using PCA on all performance metrics, with points colored by country and shaped by ranking tiers (1-20, 21-50, 51-100), overlaid with cluster boundaries and centroids.

### Generation

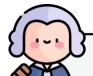

### Judgement

| | | | |
|---|---|---|---|
| Layout Compliance | 1 | Clarity | 1 |
| Chart Type Compliance | 1 | Layout Quality | 1 |
| Requirement Fulfillment | 1 | Color Quality | 2 |
| Complete Task Fulfillment | 1 | Text Readability | 1 |
| | 41 | Professional Formatting | 1 |

*Figure 22.* An example of a high-quality visualization penalized for subtle layout and clarity issues. The excessive white space between the title and the plots, along with slightly overlapping x-axis labels, detracts from its overall professional quality.

| Model | Single-Turn Generation | | | Multi-Turn Refinement | | | AVG score |
|---|---|---|---|---|---|---|---|
| | Pass Rate (%) | Task-Comp. | Quality | Pass Rate (%) | Task-Comp. | Quality | |
| *Closed-source LLMs* | | | | | | | |
| Claude-4.1-Opus (Anthropic, 2023) | 76.20 | 1.93 | 4.20 | 81.44 | 2.05 | 5.22 | 6.70 |
| Claude-4-Sonnet (Anthropic, 2023) | 68.84 | 1.73 | 3.99 | 78.41 | 1.88 | 4.80 | 6.20 |
| Gemini-2.5-Pro (Team, 2024) | 41.34 | 1.15 | 2.31 | 58.86 | 1.51 | 3.80 | 4.39 |
| ChatGPT-4o-Latest (OpenAI, 2023) | 63.54 | 1.60 | 3.33 | 69.25 | 1.51 | 4.23 | 5.33 |
| GPT-5 (OpenAI, 2025) | 69.86 | 1.76 | 2.87 | 74.13 | 1.80 | 4.33 | 5.38 |
| Grok-4 (xAI, 2025) | 64.52 | 1.63 | 3.66 | 70.24 | 1.61 | 4.42 | 5.65 |
| *Open-source LLMs* | | | | | | | |
| Kimi-K2 (Team et al., 2025b) | 60.13 | 1.52 | 3.36 | 61.03 | 1.49 | 4.05 | 5.21 |
| DeepSeek-V3.1 (DeepSeek-AI & etc., 2024) | 55.71 | 1.49 | 3.20 | 56.86 | 1.50 | 3.99 | 5.09 |
| DeepSeek-Coder-V2 (Guo et al., 2024) | 47.45 | 1.23 | 2.68 | 68.23 | 1.47 | 4.12 | 4.76 |
| DeepSeek-Coder-V2-Lite (Guo et al., 2024) | 32.79 | 0.77 | 1.90 | 47.45 | 0.97 | 2.73 | 3.19 |
| GLM-4.5 (Team et al., 2025a) | 43.38 | 1.25 | 2.48 | 64.97 | 1.54 | 3.91 | 4.59 |
| GPT-oss-120B (Agarwal et al., 2025) | 48.23 | 1.32 | 2.68 | 29.69 | 0.77 | 1.87 | 3.32 |
| GPT-oss-20B (Agarwal et al., 2025) | 44.68 | 1.21 | 2.43 | 34.02 | 0.89 | 2.14 | 3.34 |
| Seed-Coder-8B (Seed et al., 2025) | 32.38 | 0.87 | 1.74 | 57.03 | 1.26 | 3.22 | 3.55 |
| VisCoder-7B (Ni et al., 2025) | 25.46 | 0.67 | 1.50 | 51.73 | 0.99 | 2.96 | 3.06 |
| Qwen2.5-Coder-1.5B (Hui et al., 2024) | 22.81 | 0.55 | 1.38 | 36.86 | 0.66 | 1.71 | 2.15 |
| Qwen2.5-Coder-3B (Hui et al., 2024) | 17.92 | 0.50 | 1.17 | 36.46 | 0.77 | 2.00 | 2.22 |
| Qwen2.5-Coder-7B (Hui et al., 2024) | 29.94 | 0.79 | 1.79 | 46.64 | 0.98 | 2.65 | 3.10 |
| Qwen2.5-Coder-14B (Hui et al., 2024) | 38.29 | 1.09 | 2.29 | 51.53 | 1.17 | 3.04 | 3.80 |
| Qwen2.5-Coder-32B (Hui et al., 2024) | 37.68 | 1.05 | 2.21 | 48.47 | 1.20 | 3.11 | 3.79 |
| Qwen3-235B-A22B-2507 (Yang et al., 2025a) | 56.01 | 1.53 | 3.31 | 71.69 | 1.70 | 4.49 | 5.51 |
| Qwen3-Coder-480B-A35B (Hui et al., 2024) | 61.30 | 1.56 | 3.21 | 75.97 | 1.75 | 4.46 | 5.49 |
| Qwen3-Coder-30B-A3B (Hui et al., 2024) | 52.55 | 1.32 | 2.85 | 73.12 | 1.55 | 4.18 | 4.95 |
| **PlotCraftor-30B-A3B** (Ours) | 64.36 | 1.73 | 4.09 | 77.11 | 1.76 | 4.74 | 6.16 |

*Table 7.* The complete quantitative results on PlotCraft for 24 LLMs across two settings: Single-Turn Generation and Multi-Turn Refinement.

Among all open-source models, PlotCraftor demonstrates a clear advantage, achieving an average score of 6.16. This result significantly surpasses other leading open-weight models, including the much larger Qwen3-235B (5.51) and Qwen3-Coder-480B (5.49). More importantly, PlotCraftor's performance closes the gap with top-tier proprietary systems, achieving a score nearly identical to that of Claude-4-Sonnet (6.20). These comprehensive results validate that PlotCraftor provides SOTA capabilities within the open-source community for complex data visualization tasks.

# E. SynthVis-30K Details

This section provides further implementation details for the multi-agent framework used to create the SynthVis-30K dataset.

**Task Generation Agents and Criteria.** The roles of both the Task Generator and the Task Judge were fulfilled by an ensemble of Large Language Models, primarily consisting of Claude-4-Sonnet and Qwen3-Coder-480B-A22B. The iterative refinement cycle for a given task was designed to be rigorous; a task was only finalized and accepted when the Task Judge agent could no longer identify any logical inconsistencies or feasibility issues with the proposed instructions.

**Code Generation Agents and Termination Criteria.** The Code Generator agent was also comprised of an ensemble of Claude-4-Sonnet and Qwen3-Coder-480B-A22B. The crucial role of the Visual Judge, responsible for assessing the quality of the rendered images, was performed by Gemini-2.5-Pro. The code generation process was constrained to a maximum of 10 refinement iterations. The cycle was considered successful and terminated early if the generated visualization achieved a perfect Task Compliance score (4 out of 4) and a Chart Quality score of at least 6 (out of 10), with the additional constraint that each of the five quality sub-metrics must score at least 1. If these criteria were not met within the 10-iteration limit, the entire task-code pair was discarded to maintain the high-quality standard of the final dataset.

**SFT Trajectory Synthesis Details.** The Chain-of-Thought (CoT) rationales for the single-turn training instances were generated using Claude-4-Sonnet. The final SynthVis-30K dataset is composed of 30,000 instances with a 2:1 split between single-turn and multi-turn trajectories, resulting in 20,000 single-turn generation instances and 10,000 multi-turn refinement instances.

# F. Human Evaluation Scale and Reliability

To establish a high-quality ground truth for our correlation analysis (Section 4.3), we conducted a rigorous human evaluation campaign. We randomly sampled 500 charts from the generation outputs (stratified into 163 Easy, 167 Medium, and 170 Hard instances). Each chart was independently scored by three expert annotators. The final ground truth for each metric was determined via majority vote. This section analyzes the reliability and statistical validity of these human labels.

## F.1. Inter-Rater Agreement

We quantify the consistency among annotators using Fleiss' Kappa ($\kappa$). As shown in Table 8, the overall average $\kappa$ across all tasks is **0.8884**, which falls within the "Almost Perfect" agreement range ($0.81 - 1.00$).

Breakdown by difficulty reveals robust consistency:

- **EASY** and **MEDIUM** tasks achieved exceptional agreement ($\kappa > 0.90$), indicating that for standard and composite charts, human experts are highly aligned.

- **HARD** tasks, despite their complexity, maintained a strong agreement level ($\kappa \approx 0.78$), confirming that our rubrics remain sufficiently objective even for intricate, multi-subplot layouts.

Notably, the **Layout Compliance** metric achieved perfect agreement ($\kappa = 1.00$) across all levels, reflecting the binary and objective nature of layout verification.

*Table 8.* Fleiss' Kappa ($\kappa$) coefficients for human inter-rater agreement. The results demonstrate strong consensus across all dimensions, validating the reliability of the human ground truth.

| Metric | Overall $\kappa$ | EASY $\kappa$ | MEDIUM $\kappa$ | HARD $\kappa$ |
|---|---|---|---|---|
| *Compliance Metrics (Binary 0/1)* | | | | |
| **Average** | **0.9237** | **0.9184** | **0.9396** | **0.7852** |
| Layout | 1.0000 | 1.0000 | 1.0000 | 1.0000 |
| Type | 0.9093 | 0.8504 | 0.9507 | 0.7158 |
| Visual | 0.8880 | 0.9061 | 0.9155 | 0.7358 |
| Task | 0.8974 | 0.9170 | 0.8921 | 0.6891 |
| *Quality Metrics (Scale 0/1/2)* | | | | |
| **Average** | **0.8602** | **0.9093** | **0.8738** | **0.7697** |
| Clarity | 0.8846 | 0.9094 | 0.8796 | 0.8331 |
| Layout | 0.8608 | 0.9182 | 0.7814 | 0.8504 |
| Color | 0.8764 | 0.9487 | 0.9386 | 0.7343 |
| Text | 0.8333 | 0.8859 | 0.8830 | 0.6740 |
| Format | 0.8459 | 0.8841 | 0.8864 | 0.7569 |
| **Global Average** | **0.8884** | **0.9133** | **0.9030** | **0.7766** |

## F.2. Confidence Intervals and Validity

To assess the precision of the human evaluation scores, we calculated the 95% Confidence Intervals (CI) for the mean scores across all metrics and difficulties. The results, detailed in Table 9, exhibit narrow confidence intervals. This low variance indicates that our sample size ($N = 500$) is sufficient to provide a representative and statistically significant estimation of true model performance, minimizing the risk of sampling bias.

# G. Different Judge Results

To verify the robustness of our evaluation framework and ensure that our findings are not artifacts of a specific evaluator, we present the performance of 16 primary LLMs as scored independently by each member of our judge ensemble. Table 10, Table 11, and Table 12 detail the results using `Gemini-2.5-Pro`, `GPT-4o`, and `Gemini-2.5-Flash` as the sole judge, respectively. Across all three evaluators, the relative performance rankings—specifically the leadership of Claude-4.1-Opus and the strong performance of PlotCraftor among open-source models—remain highly consistent, demonstrating the reliability of our evaluation metrics.

*Table 9.* Mean scores and 95% Confidence Intervals (CI) for human evaluations across three difficulty levels. The narrow intervals indicate high statistical precision.

| Metric | EASY ($N = 163$) Mean | 95% CI | MEDIUM ($N = 167$) Mean | 95% CI | HARD ($N = 170$) Mean | 95% CI |
|---|---|---|---|---|---|---|
| **Compliance Metrics (Score range: 0-1)** | | | | | | |
| Layout | 0.8834 | [0.834, 0.928] | 0.6946 | [0.625, 0.765] | 0.6078 | [0.535, 0.680] |
| Type | 0.8446 | [0.787, 0.898] | 0.5130 | [0.437, 0.589] | 0.0725 | [0.037, 0.114] |
| Visual | 0.6871 | [0.616, 0.755] | 0.4072 | [0.333, 0.479] | 0.1118 | [0.069, 0.159] |
| Task | 0.5685 | [0.493, 0.642] | 0.3353 | [0.268, 0.405] | 0.0529 | [0.024, 0.086] |
| **Quality Metrics (Score range: 0-2)** | | | | | | |
| Clarity | 1.2311 | [1.098, 1.360] | 0.7345 | [0.613, 0.858] | 0.4294 | [0.341, 0.522] |
| Layout | 1.3395 | [1.219, 1.452] | 0.9681 | [0.844, 1.092] | 0.6176 | [0.529, 0.706] |
| Color | 1.3047 | [1.194, 1.413] | 0.9681 | [0.850, 1.086] | 0.7216 | [0.620, 0.824] |
| Text | 1.1350 | [0.994, 1.274] | 0.5788 | [0.467, 0.701] | 0.4000 | [0.316, 0.488] |
| Format | 0.9673 | [0.834, 1.096] | 0.8583 | [0.735, 0.982] | 0.6275 | [0.535, 0.722] |

| Model | Single-Turn Generation Pass Rate (%) | Task-Comp. | Quality | Multi-Turn Refinement Pass Rate (%) | Task-Comp. | Quality | AVG score |
|---|---|---|---|---|---|---|---|
| *Closed-source LLMs* | | | | | | | |
| Claude-4.1-Opus (Anthropic, 2023) | **76.20** | **1.93** | **4.20** | **81.44** | **2.05** | **5.22** | **6.70** |
| Claude-4-Sonnet (Anthropic, 2023) | 68.84 | 1.73 | 3.99 | 78.41 | 1.88 | 4.80 | 6.20 |
| Gemini-2.5-Pro (Team, 2024) | 41.34 | 1.15 | 2.31 | 58.86 | 1.51 | 3.80 | 4.39 |
| ChatGPT-4o-Latest (OpenAI, 2023) | 63.54 | 1.60 | 3.33 | 69.25 | 1.51 | 4.23 | 5.33 |
| GPT-5 (OpenAI, 2025) | 69.86 | 1.76 | 2.87 | 74.13 | 1.80 | 4.33 | 5.38 |
| Grok-4 (xAI, 2025) | 64.52 | 1.63 | 3.66 | 70.24 | 1.61 | 4.42 | 5.65 |
| *Open-source LLMs* | | | | | | | |
| Kimi-K2 (Team et al., 2025b) | 60.13 | 1.52 | 3.36 | 61.03 | 1.49 | 4.05 | 5.21 |
| DeepSeek-V3.1 (DeepSeek-AI & etc., 2024) | 55.71 | 1.49 | 3.20 | 56.86 | 1.50 | 3.99 | 5.09 |
| GLM-4.5 (Team et al., 2025a) | 43.38 | 1.25 | 2.48 | 64.97 | 1.54 | 3.91 | 4.59 |
| GPT-oss-120B (Agarwal et al., 2025) | 48.23 | 1.32 | 2.68 | 29.69 | 0.77 | 1.87 | 3.32 |
| GPT-oss-20B (Agarwal et al., 2025) | 44.68 | 1.21 | 2.43 | 34.02 | 0.89 | 2.14 | 3.34 |
| Seed-Coder-8B (Seed et al., 2025) | 32.38 | 0.87 | 1.74 | 57.03 | 1.26 | 3.22 | 3.55 |
| VisCoder-7B (Ni et al., 2025) | 25.46 | 0.67 | 1.50 | 51.73 | 0.99 | 2.96 | 3.06 |
| Qwen3-Coder-480B-A35B (Hui et al., 2024) | 61.30 | 1.56 | 3.21 | 75.97 | 1.75 | 4.46 | 5.49 |
| Qwen3-Coder-30B-A3B (Hui et al., 2024) | 52.55 | 1.32 | 2.85 | 73.12 | 1.55 | 4.18 | 4.95 |
| **PlotCraftor-30B-A3B (Ours)** | **64.36** | **1.73** | **4.09** | **77.11** | **1.76** | **4.74** | **6.16** |

*Table 10.* Quantitative results on PlotCraft for 16 primary LLMs by `Gemini-2.5-Pro` Judge

# H. Evaluation Details

**PlotCraft Evaluation.** For the evaluation on our PlotCraft benchmark, all models were prompted using a standardized format. For the single-turn generation tasks, we used the prompt detailed in Prompt H. For the multi-turn refinement tasks, this same prompt served as the initial user turn in the conversation, followed by the specific refinement request.

**VisEval and PandasPlotBench Evaluation.** For our evaluation on the VisEval benchmark, we adopted the CoML table format and restricted our analysis to tasks specifically designed for the Matplotlib library. Similarly, for the PandasPlotBench benchmark, we exclusively evaluated the Matplotlib-based tasks. To ensure consistency with established practices for these benchmarks, the performance of the generated visualizations for both VisEval and PandasPlotBench was assessed using ChatGPT-4o-Latest as the automated judge.

| Model | Single-Turn Generation | | | Multi-Turn Refinement | | | AVG score |
|---|---|---|---|---|---|---|---|
| | Pass Rate (%) | Task-Comp. | Quality | Pass Rate (%) | Task-Comp. | Quality | |
| *Closed-source LLMs* | | | | | | | |
| Claude-4.0-Opus (Anthropic, 2023) | **77.40** | **1.94** | **4.24** | **80.54** | **2.03** | **5.12** | **6.74** |
| Claude-4-Sonnet (Anthropic, 2023) | 69.25 | 1.71 | 3.91 | 78.43 | 1.86 | 4.71 | 6.18 |
| Gemini-2.5-Pro (Team, 2024) | 41.67 | 1.16 | 2.26 | 59.59 | 1.52 | 3.76 | 4.40 |
| ChatGPT-4o-Latest (OpenAI, 2023) | 64.32 | 1.61 | 3.38 | 68.13 | 1.49 | 4.20 | 5.43 |
| GPT-5 (OpenAI, 2025) | 70.19 | 1.78 | 2.88 | 73.77 | 1.83 | 4.34 | 5.43 |
| Grok-4 (xAI, 2025) | 63.44 | 1.62 | 3.62 | 68.96 | 1.62 | 4.43 | 5.73 |
| *Open-source LLMs* | | | | | | | |
| Kimi-K2 (Team et al., 2025b) | 59.43 | 1.51 | 3.34 | 61.36 | 1.48 | 3.99 | 5.15 |
| DeepSeek-V3.1 (DeepSeek-AI & etc., 2024) | 54.96 | 1.50 | 3.16 | 57.11 | 1.51 | 4.06 | 5.04 |
| GLM-4.5 (?) | 42.91 | 1.26 | 2.51 | 65.45 | 1.54 | 3.93 | 4.68 |
| GPT-oss-120B (Agarwal et al., 2025) | 48.03 | 1.33 | 2.66 | 30.14 | 0.78 | 1.85 | 3.29 |
| GPT-oss-20B (Agarwal et al., 2025) | 44.83 | 1.20 | 2.44 | 33.68 | 0.88 | 2.14 | 3.40 |
| Seed-Coder-8B (Seed et al., 2025) | 31.79 | 0.86 | 1.74 | 58.16 | 1.25 | 3.21 | 3.61 |
| VisCoder-7B (Ni et al., 2025) | 25.97 | 0.67 | 1.47 | 51.57 | 1.00 | 2.98 | 3.01 |
| Qwen3-Coder-480B-A35B (Hui et al., 2024) | 61.39 | 1.57 | 3.24 | 74.49 | 1.78 | 4.54 | 5.50 |
| Qwen3-Coder-30B-A3B (Hui et al., 2024) | 53.34 | 1.34 | 2.84 | 72.93 | 1.53 | 4.20 | 4.90 |
| **PlotCraftor-30B-A3B** (Ours) | **64.72** | **1.72** | **4.15** | **78.38** | **1.74** | **4.74** | **6.10** |

*Table 11.* Quantitative results on PlotCraft for 16 primary LLMs by `GPT-4o` judge.

| Model | Single-Turn Generation | | | Multi-Turn Refinement | | | AVG score |
|---|---|---|---|---|---|---|---|
| | Pass Rate (%) | Task-Comp. | Quality | Pass Rate (%) | Task-Comp. | Quality | |
| *Closed-source LLMs* | | | | | | | |
| Claude-4.2-Opus (Anthropic, 2023) | **75.00** | **1.92** | **4.16** | **82.34** | **2.07** | **5.32** | **6.66** |
| Claude-4-Sonnet (Anthropic, 2023) | 68.43 | 1.75 | 4.07 | 78.39 | 1.90 | 4.89 | 6.22 |
| Gemini-2.5-Pro (Team, 2024) | 41.01 | 1.14 | 2.36 | 58.13 | 1.50 | 3.84 | 4.38 |
| ChatGPT-4o-Latest (OpenAI, 2023) | 62.76 | 1.59 | 3.28 | 70.37 | 1.53 | 4.26 | 5.23 |
| GPT-5 (OpenAI, 2025) | 69.53 | 1.74 | 2.86 | 74.49 | 1.77 | 4.32 | 5.33 |
| Grok-4 (xAI, 2025) | 65.60 | 1.64 | 3.70 | 71.52 | 1.60 | 4.41 | 5.57 |
| *Open-source LLMs* | | | | | | | |
| Kimi-K2 (Team et al., 2025b) | 60.83 | 1.53 | 3.38 | 60.70 | 1.50 | 4.11 | 5.27 |
| DeepSeek-V3.1 (DeepSeek-AI & etc., 2024) | 56.46 | 1.48 | 3.24 | 56.61 | 1.49 | 3.92 | 5.14 |
| GLM-4.5 (?) | 43.85 | 1.24 | 2.45 | 64.49 | 1.54 | 3.89 | 4.50 |
| GPT-oss-120B (Agarwal et al., 2025) | 48.43 | 1.31 | 2.70 | 29.24 | 0.76 | 1.89 | 3.35 |
| GPT-oss-20B (Agarwal et al., 2025) | 44.53 | 1.22 | 2.42 | 34.36 | 0.90 | 2.14 | 3.28 |
| Seed-Coder-8B (Seed et al., 2025) | 32.97 | 0.88 | 1.74 | 55.90 | 1.27 | 3.23 | 3.49 |
| VisCoder-7B (Ni et al., 2025) | 24.95 | 0.67 | 1.53 | 51.89 | 0.98 | 2.94 | 3.11 |
| Qwen3-Coder-480B-A35B (Hui et al., 2024) | 61.21 | 1.55 | 3.18 | 77.45 | 1.72 | 4.38 | 5.48 |
| Qwen3-Coder-30B-A3B (Hui et al., 2024) | 51.76 | 1.30 | 2.86 | 73.31 | 1.57 | 4.16 | 5.00 |
| **PlotCraftor-30B-A3B** (Ours) | **64.00** | **1.74** | **4.03** | **75.84** | **1.78** | **4.74** | **6.22** |

*Table 12.* Quantitative results on PlotCraft for 16 primary LLMs by `Gemini-2.5-Flash` Judge.

---

**Prompt H.1: Generation Prompt**

```
You are an expert Python data visualization developer specializing in
    matplotlib and seaborn. Your task is to generate high-quality, executable
    Python code for data visualization based on the given task description and
    dataset information.
Output format:
- Provide only the Python code wrapped in ```python and ``` markers
- Ensure the code can run independently
```

# I. Baseline Models

To benchmark the performance of PlotCraftor, we selected 23 representative baselines divided into proprietary and open-weight categories.

**Proprietary Models**     We include six state-of-the-art closed-source systems:

- **Anthropic:** Claude-4.1-Opus and Claude-4-Sonnet (Anthropic, 2023).

- **OpenAI:** GPT-5 and the latest version of GPT-4o (OpenAI, 2025).

- **Google:** Gemini 2.5 Pro (Team, 2024).

- **xAI:** Grok-4 (xAI, 2025).

**Open-Weight Models**     We evaluate 17 competitive open-source models spanning a diverse range of architectures and sizes:

- **Large-Scale Models (¿100B):** Kimi-K2 (1T, A32B) (Team et al., 2025b), DeepSeek-V3.1 (671B, A37B) (DeepSeek-AI & etc., 2024), GLM-4.5 (355B, A32B) (Team et al., 2025a), GPT-oss-120B (A5.1B) (Agarwal et al., 2025), and Qwen3-Coder-480B (A22B) (Hui et al., 2024).

- **Mid-to-Small Scale Models (¡100B):** GPT-oss-20B (A3.6B) (Agarwal et al., 2025), Qwen3-Coder-30B (A3B) (Hui et al., 2024), Qwen2.5-Coder (Hui et al., 2024), DeepSeek-Coder (Guo et al., 2024), and Qwen3 (Yang et al., 2025a).

- **Comparable Baselines:** To provide a direct comparison with PlotCraftor (30B, A3B), we specifically include Seed-Coder-8B (Seed et al., 2025) and VisCoder-7B (Ni et al., 2025).

# J. Discussion

## J.1. Model Performance Comparison

**Comparison 1** For the task detailed in Instruction J.1, Figure 23 compares the outputs of several leading models. Our model, PlotCraftor, successfully generates a visualization that meets all specified requirements. In contrast, the base model, Qwen3-Coder-30B-A3B, exhibits both chart type and factual errors and uses an unprofessional default gray background. GPT-5 clutters the visualization with excessive legends, gridlines, and text, leading to poor clarity and element overlap. Similarly, GLM-4.5 and Cluade-4.1-Opus produce charts with unpolished gray backgrounds and poor color choices, with the latter also failing on chart type compliance. Gemini-2.5-Pro fails to generate any output, resulting in a blank image.

---

**Instruction J.1: Comparison - 1**

```
Create a comprehensive temporal analysis of Kaggle tweet engagement patterns
    from 2010-2021. Design a 2x2 subplot grid where each subplot combines
    multiple visualization elements: (1) Top-left: A line chart showing
    yearly tweet volume trends overlaid with a bar chart displaying average
    engagement metrics (likes + retweets) per year, (2) Top-right: An area
    chart depicting the cumulative distribution of tweet languages over time
    with stacked areas for the top 5 languages, (3) Bottom-left: A dual-axis
    plot combining a line chart of monthly tweet frequency patterns overlaid
    with a scatter plot showing seasonal engagement spikes, and (4) Bottom-
    right: A time series decomposition showing trend, seasonal, and residual
    components of daily tweet activity across the entire dataset period. Each
     subplot should include appropriate legends, annotations for significant
    events or patterns, and use consistent color schemes to highlight the
    evolution of Kaggle's social media presence and community engagement over
     the decade.
```

**Instruction J.2: Comparison - 2**

Create a comprehensive 3x2 subplot grid analyzing profit margin deviations
    and pricing strategies across multiple e-commerce platforms. Each subplot
     should be a composite visualization combining multiple chart types:

Top row (3 subplots):
1. Left: Create a diverging bar chart showing the deviation of each platform'
    s MRP from the average MRP, overlaid with error bars representing the
    standard deviation of pricing across different style categories
2. Center: Design a dumbbell plot comparing TP1 vs TP2 costs for different
    product categories, with a secondary y-axis line plot showing the profit
    margin percentage deviation from the overall average margin
3. Right: Build a slope chart showing MRP changes from "MRP Old" to "Final
    MRP Old" for top 10 style IDs, combined with scatter points indicating
    the magnitude of price adjustment

Bottom row (2 subplots):
4. Left: Construct a diverging lollipop chart displaying how each platform's
    MRP deviates from the baseline Amazon MRP, with horizontal reference
    lines showing +=10% and +=20% deviation thresholds
5. Right: Generate a radar chart comparing normalized pricing metrics (TP1,
    TP2, various platform MRPs) for the top 5 most frequent style categories,
     overlaid with area fill showing the deviation range from the category
    median

Use a consistent color scheme where positive deviations are shown in green
    tones and negative deviations in red tones. Include proper titles,
    legends, and annotations highlighting the most significant deviations.
    The visualization should reveal pricing inconsistencies, profit margin
    variations, and strategic pricing patterns across different e-commerce
    platforms.

**Instruction J.3: Comparison - 3**

Create a composite visualization showing the composition of cosmetic products
     by chemical content. Design a subplot with two complementary charts: (1)
     a stacked bar chart displaying the top 8 companies by total product
    count, with each bar segment colored by primary category to show the
    product type distribution within each company, and (2) a pie chart
    showing the overall market share of these top 8 companies based on their
    total number of reported products. Include proper legends, titles, and
    ensure the color schemes are consistent between both charts.

**Instruction J.4: Comparison - 4**

```
Create a comprehensive 3x3 subplot grid analyzing the temporal evolution of
    London Underground station usage from 2007-2017. Each subplot should be a
     composite visualization combining multiple chart types:

Top row (2007-2009): For each year, create a scatter plot showing the
    relationship between weekday entries and annual usage (in millions), with
     bubble sizes representing weekend activity levels (Saturday + Sunday
    entries), overlaid with a regression line and confidence intervals.

Middle row (2010-2012): For each year, create a dual-axis plot combining a
    histogram of annual usage distribution with a KDE curve overlay, and add
    vertical lines marking the 25th, 50th, and 75th percentiles of usage.

Bottom row (2013-2015): For each year, create a combination plot showing both
     a box plot of weekday vs weekend entry ratios by borough (grouped by top
     10 boroughs by station count) and overlay violin plots to show the
    distribution density.

Each subplot should include year-specific titles, appropriate color schemes
    that evolve across the timeline, and statistical annotations (correlation
     coefficients for scatter plots, percentile values for histograms, and
    median values for box plots). The overall visualization should reveal how
     station usage patterns, distributions, and borough-level variations
    evolved during this decade.
```

**Instruction J.5: Comparison - 5**

```
Create a composite visualization showing the temporal evolution of CO2
    emissions across different countries. Design a subplot layout with two
    complementary charts: (1) a line chart displaying CO2 emissions trends
    over time (1990-2020) for the top 3 countries by total emissions, and (2)
     a stacked area chart showing the cumulative contribution of these same
    countries to global CO2 emissions over the same time period. Both charts
    should highlight how emission patterns have changed over the three-decade
     span and reveal which countries have been the dominant contributors to
    global CO2 emissions.
```

**Comparison 2**   The task in Instruction J.1 requires a sophisticated 3x2 grid with composite charts (3 subplots for top row and 2 for bottom row). As shown in Figure 24, PlotCraftor correctly renders this complex layout. The other models struggle significantly: Qwen3-Coder-30B-A3B implements an incorrect layout and suffers from extensive element overlap. GPT-5 produces plots with distorted aspect ratios, awkward typography, and significant text overlap. GPT-oss-120B also fails to generate the correct 3x2 layout. While Cluade-4.1-Opus generates a mostly correct visualization, it misplaces the legend, affecting the overall composition. Gemini-2.5-Pro again produces a blank image.

**Comparison 3**   The task in Instruction J.1, which requires a simpler composite chart, demonstrates that many high-performing models can handle less complex requests. As seen in Figure 25, PlotCraftor, Kimi-K2, Cluade-4-Sonnet, and Qwen3-Coder-480B-A22B all produce satisfactory results. This comparison highlights the specific failure modes of other models on what should be a manageable task: Qwen3-Coder-30B-A3B uses an incorrect chart type, while GPT-5 fails to produce any visualization.

**Comparison 4**    For the demanding 3x3 temporal grid specified in Instruction J.1, Figure 26 shows that PlotCraftor is the only model to produce a fully correct visualization. The base model, Qwen3-Coder-30B-A3B, fails to generate any output. The other models, including GPT-5, GLM-4.5, Cluade-4-Sonnet, and Gemini-2.5-Pro, all manage to generate visualizations but fail to adhere to the prompt, exhibiting various chart type errors across the subplots.

**Comparison 5**    The task in Instruction J.1 requires a composite plot with line and stacked area charts. Figure 27 illustrates that PlotCraftor successfully generates the required visualization with professional formatting. In contrast, Qwen3-Coder-30B-A3B generates an incorrect chart type, failing to meet the core requirement. GPT-5 produces a visualization that, while functionally similar, suffers from a cramped and poorly organized layout. Other powerful models like ChatGPT-4o, Cluade-4.1-Opus, and Gemini-2.5-Pro also attempted the task with varying degrees of success and failure as depicted.

### J.2. Scaling Comparison

Figure 28 and Figure 29 present scatter plots of average model scores as a function of model size on PlotCraft's Easy and Hard tasks, respectively. These plots visually confirm that the benefits of model scaling are highly dependent on task difficulty. For models under 100B parameters, performance on Easy tasks scales rapidly with size, while performance on Hard tasks remains flat, only improving for models beyond the 100B threshold. This disparity is mirrored in supervised fine-tuning (SFT): smaller models can be fine-tuned to near-proprietary levels on Easy tasks, yet SFT provides minimal benefit for Hard tasks. This indicates that solving complex visualization challenges may rely more on the emergent reasoning abilities that come with scale than on task-specific fine-tuning.

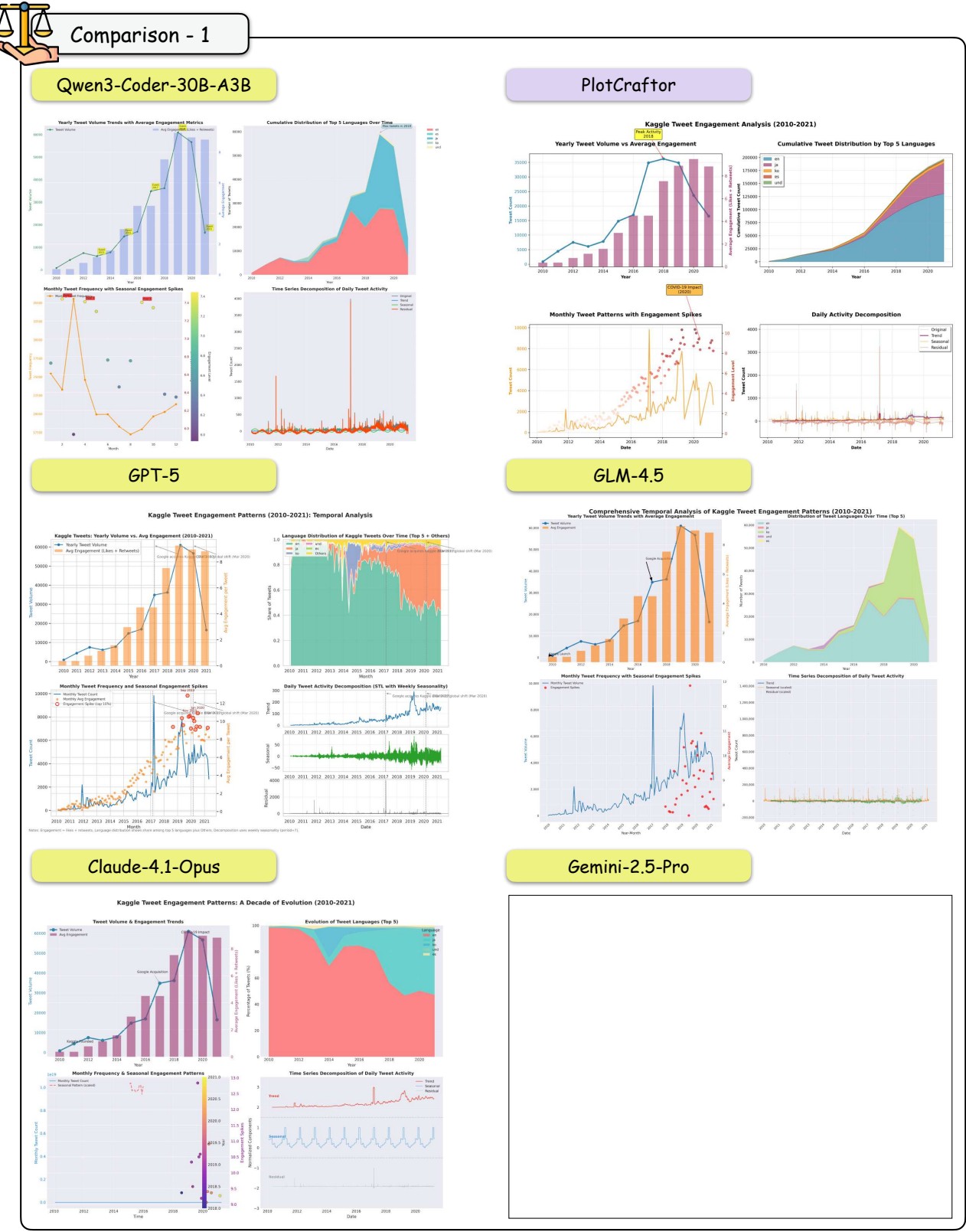

*Figure 23.* Qualitative comparison for the task in Instruction J.1. PlotCraftor produces a correct and complete visualization, while other models exhibit a range of failures, including incorrect chart types (Qwen3-Coder, Cluade-4.1-Opus), excessive clutter (GPT-5), poor formatting (GLM-4.5), and a blank output (Gemini-2.5-Pro).

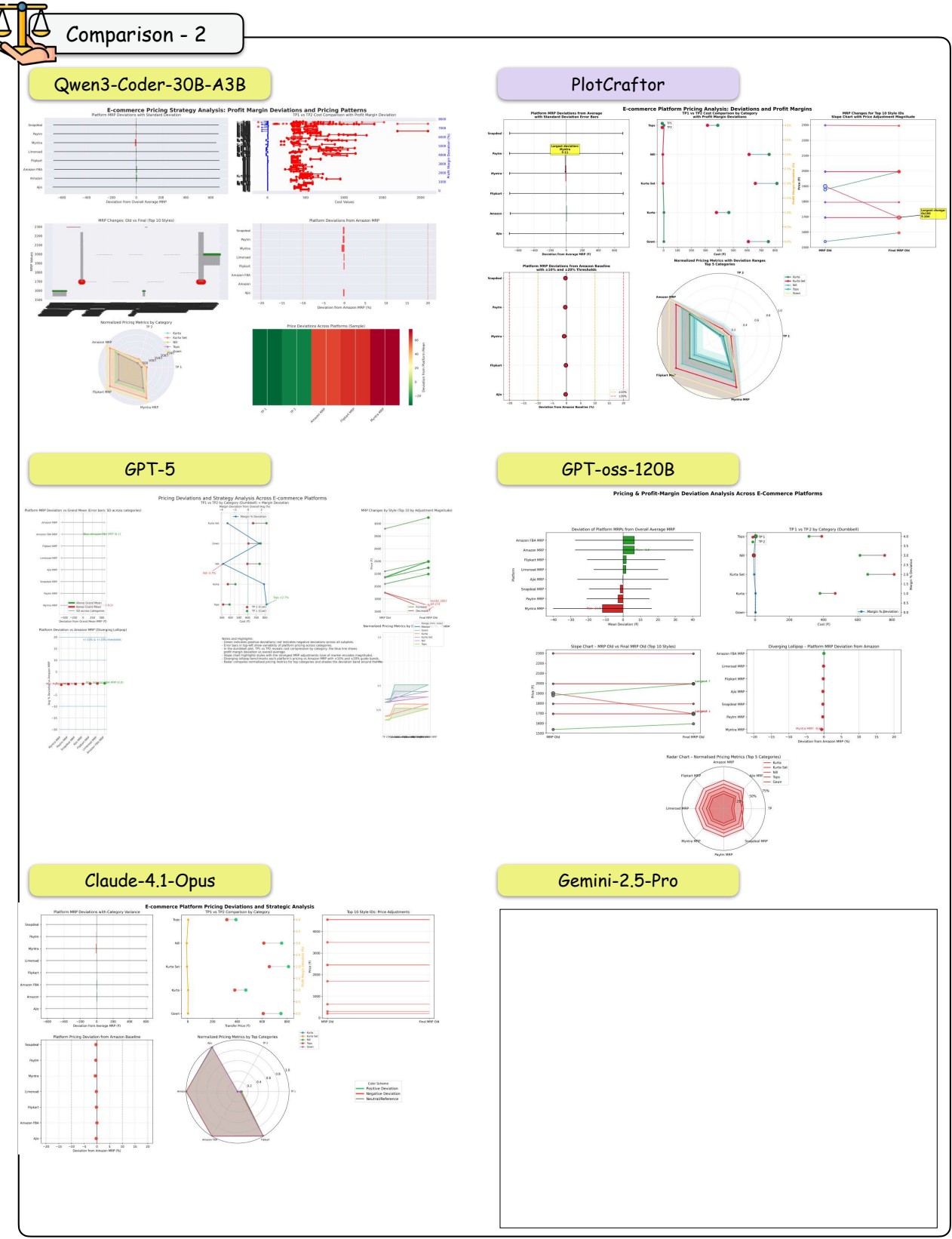

*Figure 24.* Qualitative comparison for the task in Instruction J.1. PlotCraftor correctly generates the complex 3x2 composite grid, whereas other models fail on layout generation (Qwen3-Coder, GPT-oss-120B), produce distorted outputs (GPT-5), have minor compositional flaws (Cluade-4.1-Opus), or fail completely (Gemini-2.5-Pro).

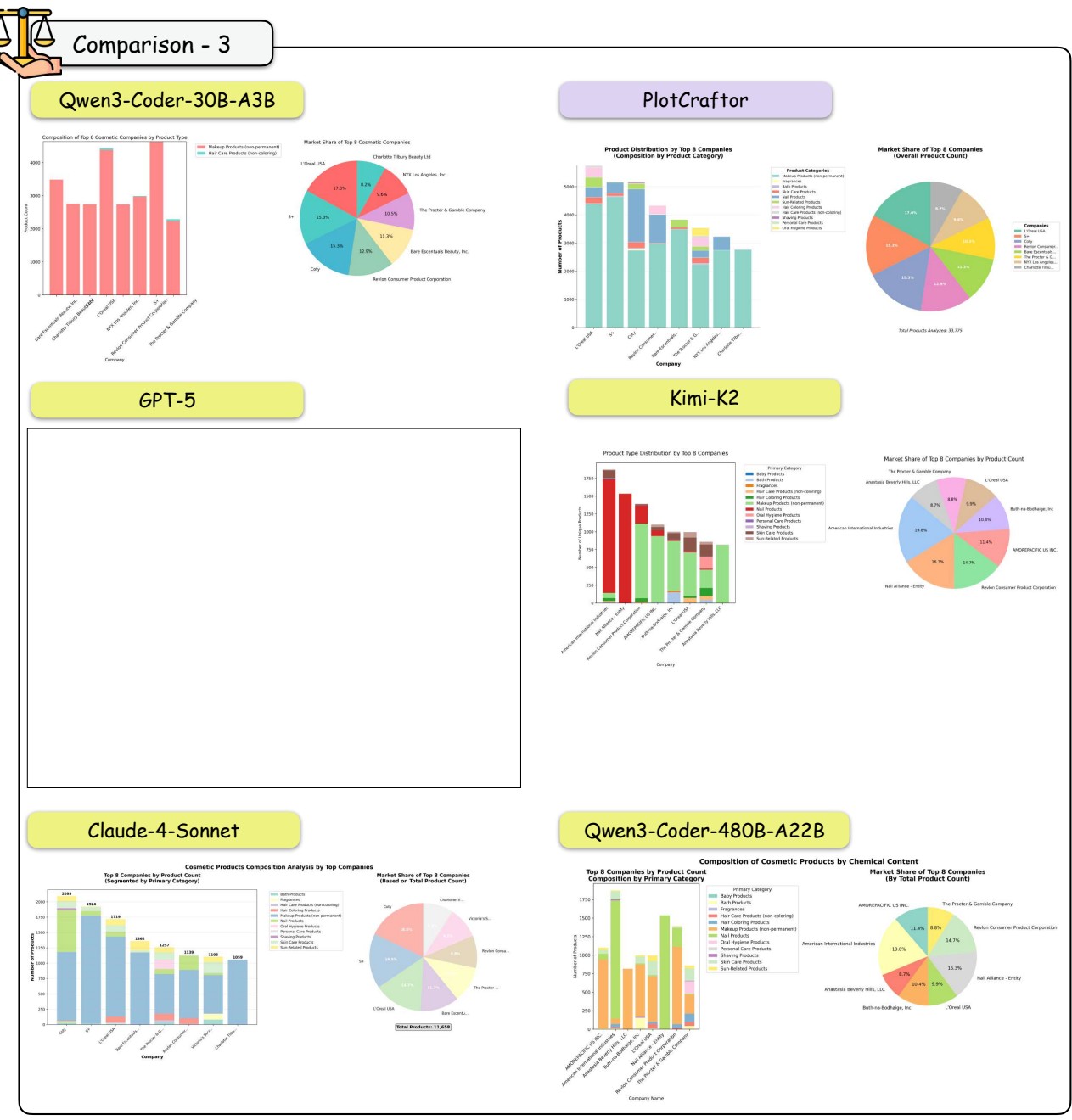

*Figure 25.* Qualitative comparison for the simpler task in Instruction J.1. While PlotCraftor and several other capable models generate correct visualizations, this figure highlights key failures, including an incorrect chart type from Qwen3-Coder-30B-A3B and a blank output from GPT-5.

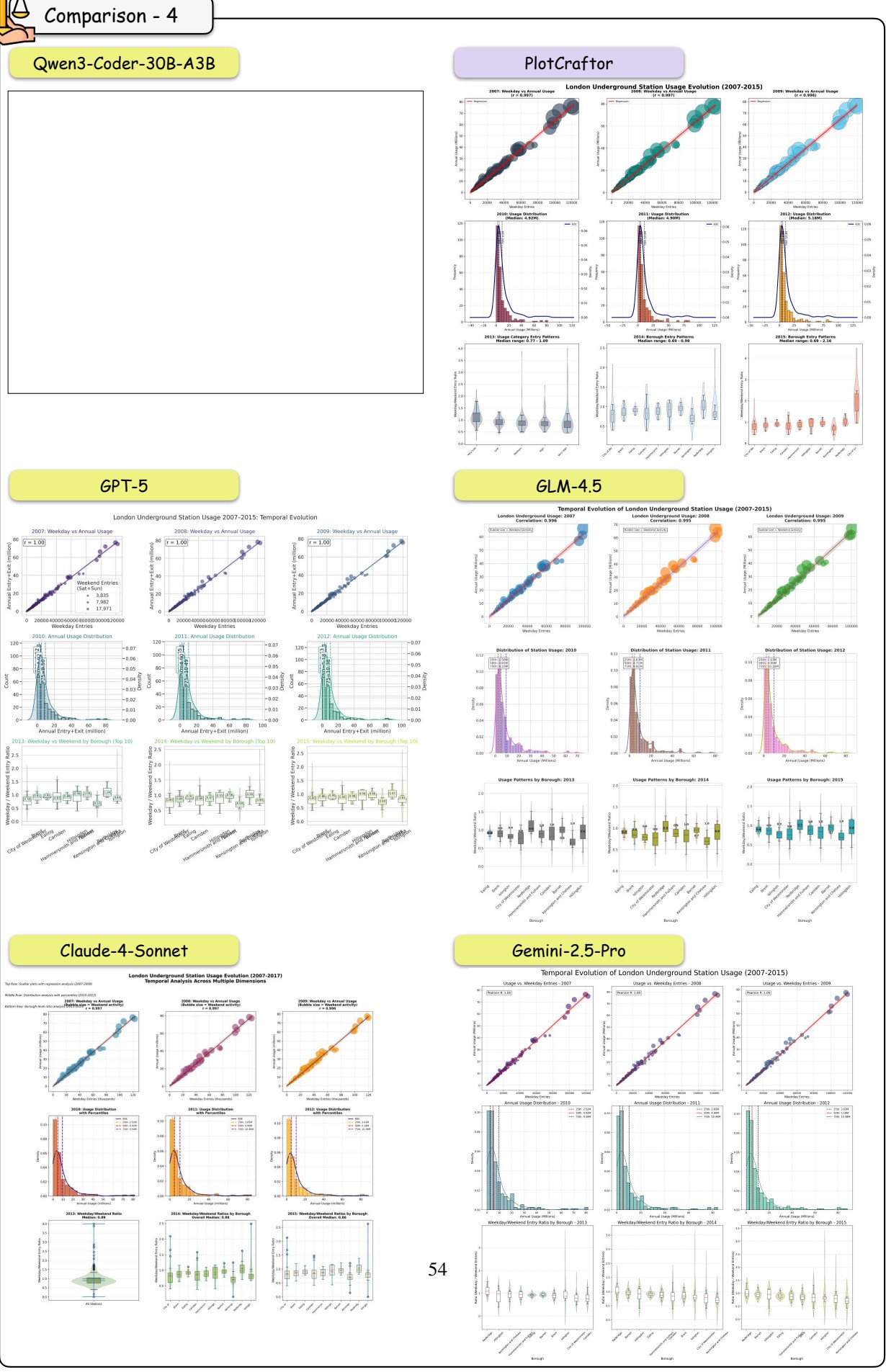

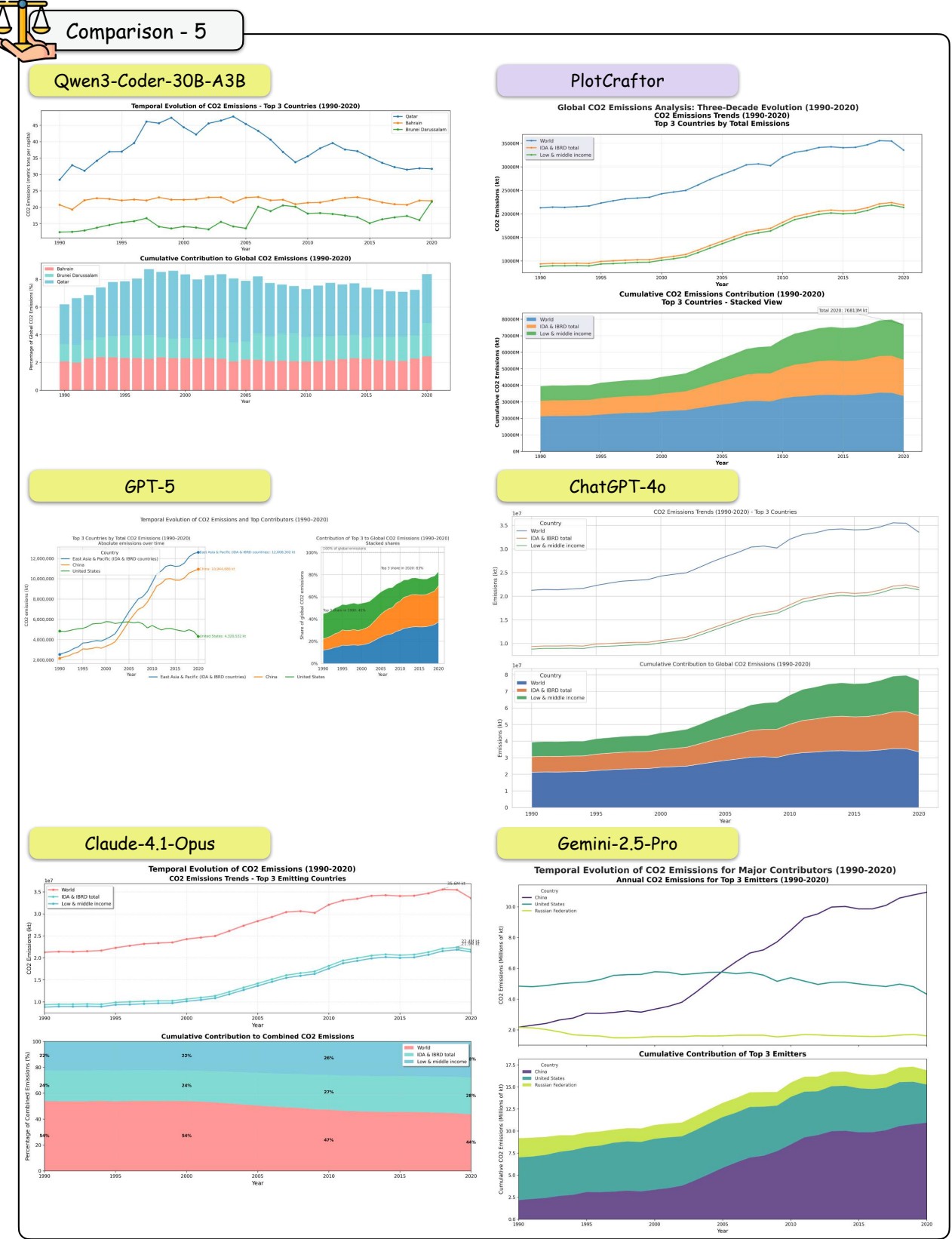

*Figure 27.* Qualitative comparison for the composite time-series task in Instruction J.1. PlotCraftor's correct and well-formatted output is contrasted with other models that produced incorrect chart types (Qwen3-Coder-30B-A3B) or suffered from poor, cramped layouts (GPT-5).

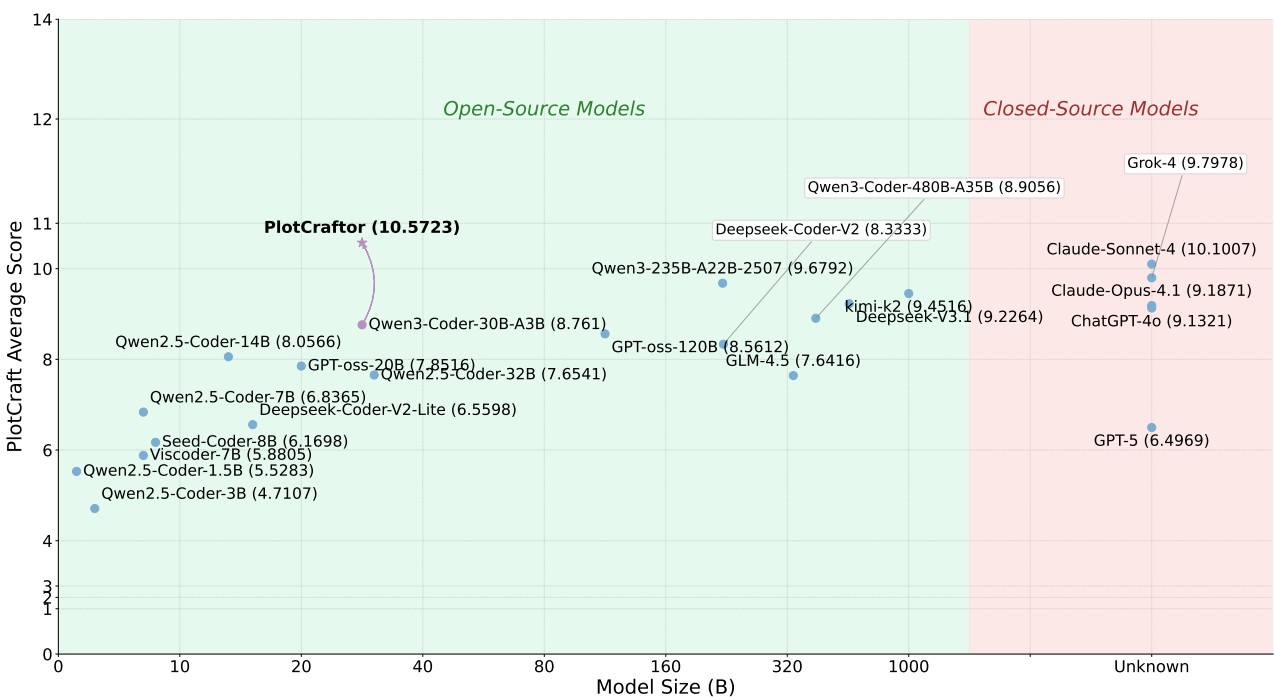

*Figure 28.* Model performance versus model size on **Easy** tasks from the PlotCraft benchmark. The plot shows a clear positive correlation, where performance scales effectively with the number of parameters across the full range of model sizes.

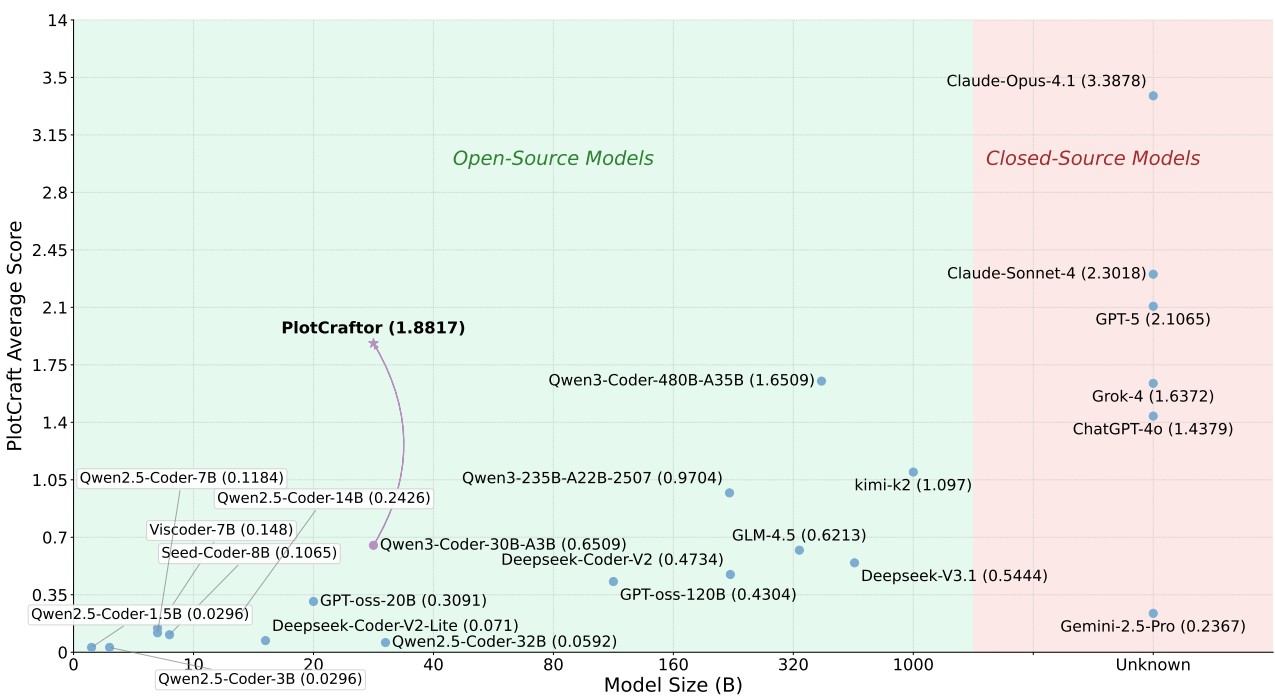

*Figure 29.* Model performance versus model size on **Hard** tasks from the PlotCraft benchmark. The plot illustrates that performance remains largely stagnant for models under 100B parameters, with a notable improvement only emerging with models that surpass this size threshold.

| Compliance Metrics | | | | Quality Metrics | | | | |
|---|---|---|---|---|---|---|---|---|
| Layout | Type | Visual | Task | Clarity | Layout | Color | Text | Format |
| 0.61 | 0.58 | 0.41 | 0.50 | 0.39 | 0.51 | 0.63 | 0.58 | 0.70 |

*Table 13.* Cohen's Kappa scores for agreement between our Claude-4-Sonnet judge and human evaluations, categorized by Compliance and Quality metrics.

| Compliance Metrics | | | | Quality Metrics | | | | |
|---|---|---|---|---|---|---|---|---|
| Layout | Type | Visual | Task | Clarity | Layout | Color | Text | Format |
| 0.64 | 0.62 | 0.53 | 0.55 | 0.36 | 0.52 | 0.54 | 0.61 | 0.67 |

*Table 14.* Cohen's Kappa scores for agreement between our ChatGPT-4o judge and human evaluations, categorized by Compliance and Quality metrics.

## K. Correlation with Human Evaluation Details

In our study, we also evaluated the reliability of other prominent models, Claude-4-Sonnet and ChatGPT-4o, as potential automated judges. The agreement between these models and our human evaluations, as measured by Cohen's Kappa scores, is presented in Table 13 and Table app-tab:gpt-judge, respectively.

The results reveal a consistent and critical weakness in both models' visual analysis capabilities. While they achieve moderate to substantial agreement on most compliance and formatting metrics, their reliability drops sharply on the Clarity metric, which is specifically designed to assess element overlap. With Kappa scores of only 0.39 for Claude-4-Sonnet and 0.36 for ChatGPT-4o on this metric, it is evident that both models struggle to effectively identify and penalize overlapping elements. This deficiency often leads them to assign inaccurately high quality scores to charts with significant readability issues, limiting their viability as standalone judges for complex data visualizations.

## L. Error Analysis

### Error Case 1: Severe Element Overlap

**This case demonstrates a critical failure in spatial reasoning, resulting in severe overlap between chart elements and text.** As shown in Figure 30, the generated visualization suffers from widespread element occlusion that renders it largely unreadable. Specific failures include: overlapping pie chart labels in subplot (0,1); annotations clashing with the line plot in subplot (0,0); indecipherable stacked text in subplot (0,2); a legend obscuring the plot in subplot (1,0); and a complete overlap of two distinct charts in subplot (1,2). These errors indicate a fundamental inability of the model to manage spatial allocation within a complex multi-plot layout.

### Error Case 2: Conflicting Layout Managers

**This case illustrates a technical failure where incompatible layout management commands lead to a complete collapse of the plotting canvas.** The model-generated code produces a blank image because of a conflict between Matplotlib's `constrained_layout` engine and the subsequent addition of figure-level elements, particularly `fig.legend()`.

The core of the issue lies in the subplot initialization:

```
fig, (ax_top, ax_bottom) = plt.subplots(
2, 1, figsize=(12, 16), sharex=True, \textt{constrained\_layout=True},
gridspec_kw=dict(height_ratios=[1, 1])
)
```

Here, `constrained_layout=True` activates a sophisticated automatic layout engine designed to prevent the overlap of axes labels and titles by adjusting subplot positions. However, this engine has known limitations when interacting with elements added directly to the figure canvas, such as:

```
fig.suptitle(...)
fig.text(...)
fig.legend(...)
fig.text(...)
```

The conflict arises because `constrained_layout` is primarily designed to manage subplots (Axes) and their immediate decorations. It cannot properly account for the space consumed by figure-level objects like `fig.legend()`, which are placed in the figure's coordinate system independently of the subplot grid.

The failure proceeds as follows:

1. **Engine Activation**: `constrained_layout=True` instructs Matplotlib to manage all subplot layouts automatically.

2. **Content Preparation**: The plotting functions successfully prepare the bar charts and text within the `ax_top` and `ax_bottom` axes objects in memory.

3. **Conflict Introduction**: The code then adds a `fig.legend()` directly to the figure. The layout engine does not natively know how to reserve space for this object while also optimizing the subplot positions.

4. **Layout Calculation Failure**: When `plt.show()` is called, the rendering backend executes the `constrained_layout` algorithm. It attempts to find a solution that accommodates the subplots, their labels, and the "external" figure-level elements. Unable to converge on a stable solution, the algorithm fails.

5. **Canvas Collapse**: A common outcome of this failure is that the layout engine allocates zero (or a near-zero) height and width to the subplots in a misguided attempt to make space for the other elements. Consequently, the primary drawing areas vanish.

The final rendered output is a blank figure canvas. While the figure title or legend might be present, the core subplots are invisible because their dimensions have been reduced to zero, as shown in Figure 31.

**Error Case 3: Critical Rendering and Layout Failures** This case, generated by Gemini-2.5-Pro, demonstrates a combination of critical rendering failures and severe layout issues that render the chart unusable, as shown in Figure 32.

The most significant issue is a rendering failure in the main plot, which displays as a solid gray area instead of the intended visualization, indicating a fundamental error in the code's data-to-visual mapping. In addition to this critical failure, the chart suffers from severe clarity problems. All x-axis tick labels are collapsed and stacked on top of one another, making them completely illegible. Furthermore, the legend in the supplementary plot overlaps with the chart's content, obscuring key information and making that part of the visualization difficult to interpret.

**Reference Visualization for Error Case 3** Figure 33 displays the ground truth for the task previously shown in Figure 32. Unlike the model's output, which suffered from severe overlapping, the reference code achieves a clean and readable layout by strategically selecting top cities based on the data description. This demonstrates that the layout issues were due to the model's lack of reasoning capability rather than task infeasibility.

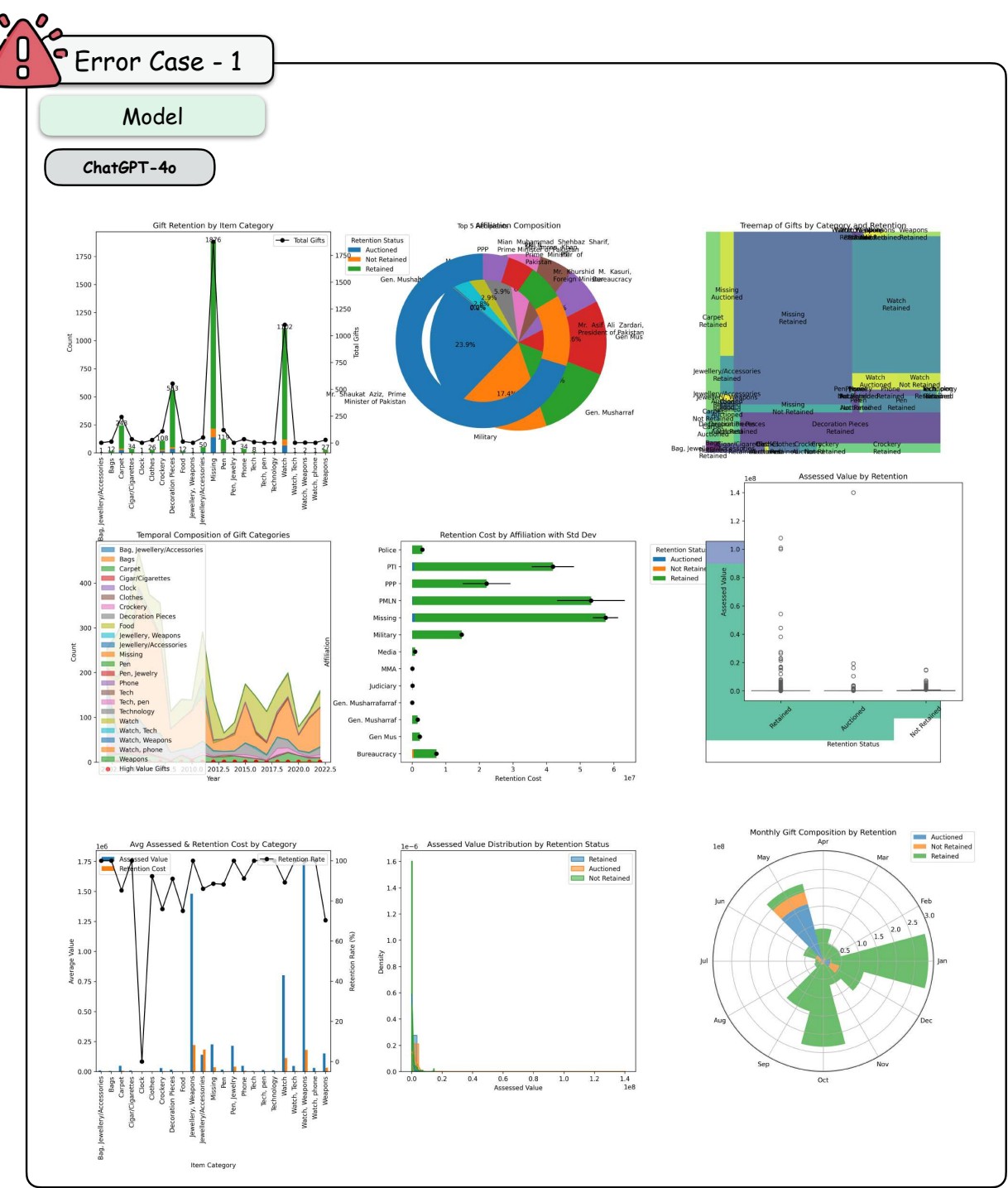

*Figure 30.* An error case generated by **ChatGPT-4o**, characterized by severe and pervasive element overlap. The visualization fails due to multiple instances of text, labels, legends, and entire subplots occluding one another, making the chart uninterpretable and highlighting a deficiency in layout management.

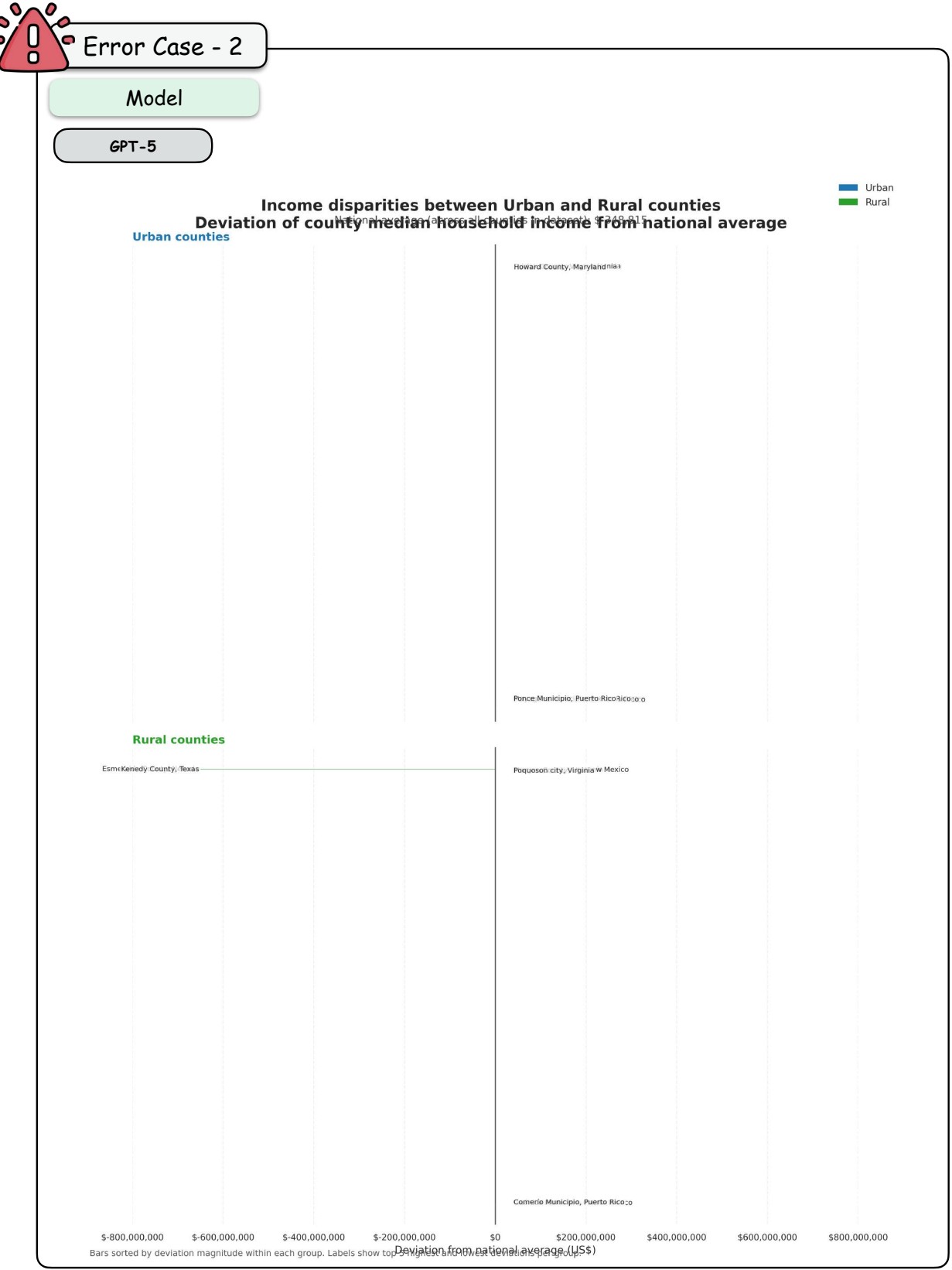

*Figure 31.* A failure case generated by **GPT-5**, resulting from a conflict between Matplotlib's layout managers. The use of `constrained_layout=True` in conjunction with figure-level elements like `fig.legend()` causes the layout engine to fail, collapsing the subplot dimensions to zero and producing a blank image.

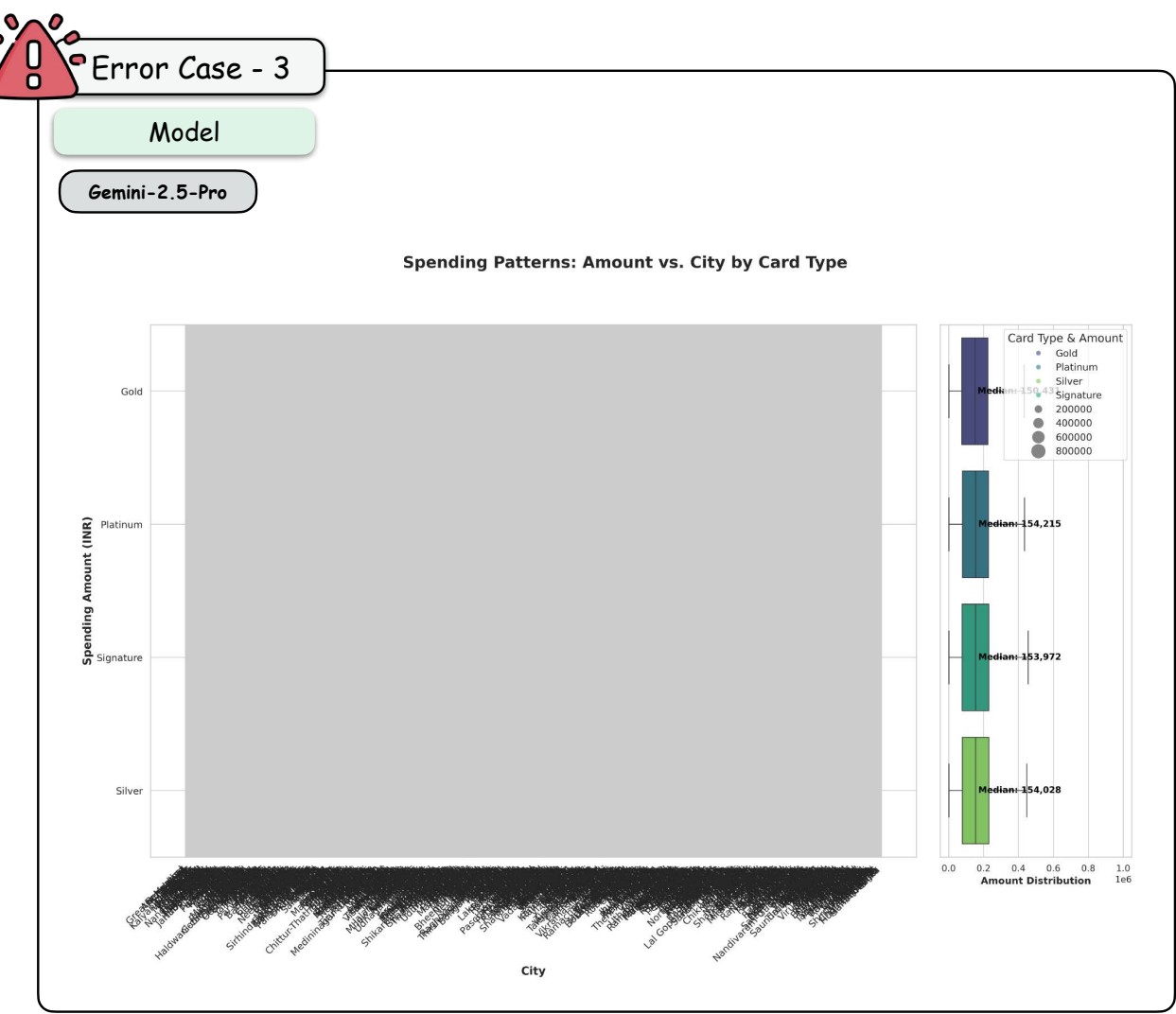

*Figure 32.* An error case generated by **Gemini-2.5-Pro** exhibiting both a critical rendering failure and severe layout issues. The main plot is incorrectly rendered as a solid gray block, while overlapping x-axis labels and legends make other parts of the visualization unreadable.

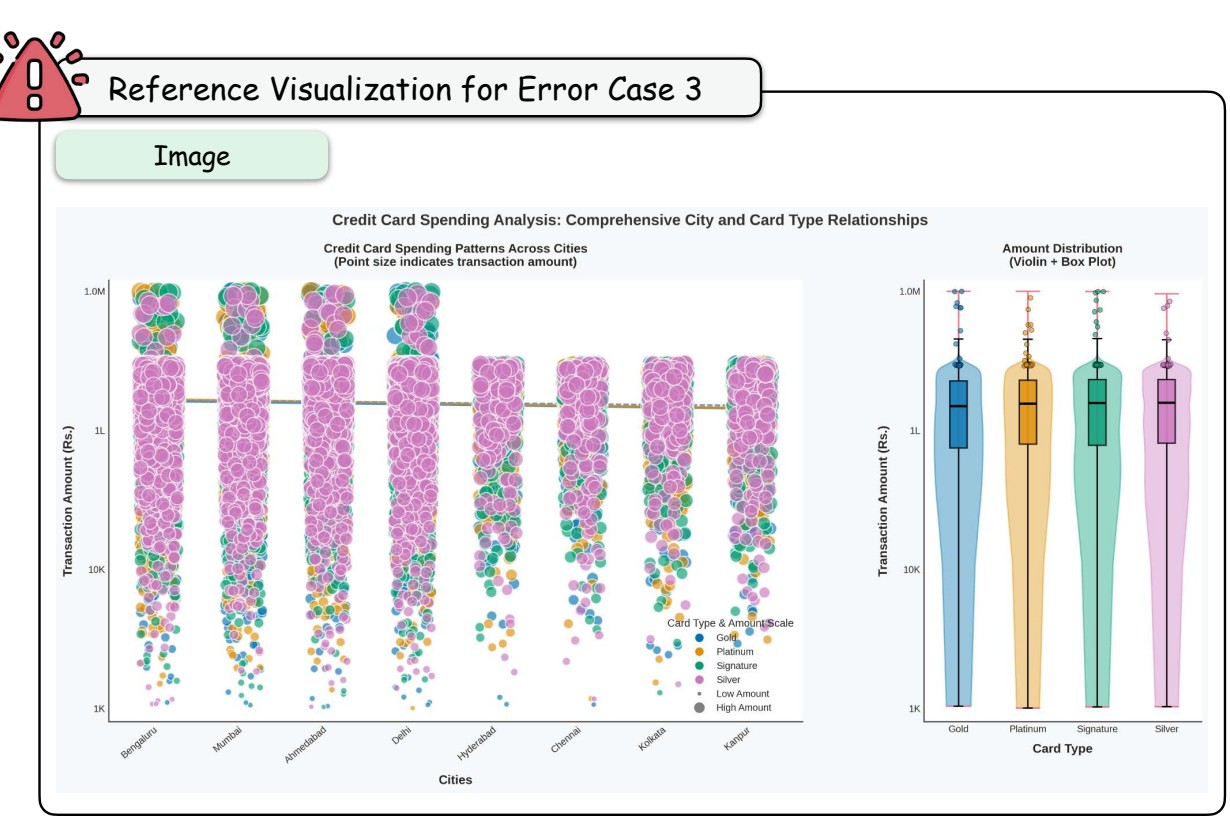

*Figure 33.* Human-written reference visualization for Error Case 3.

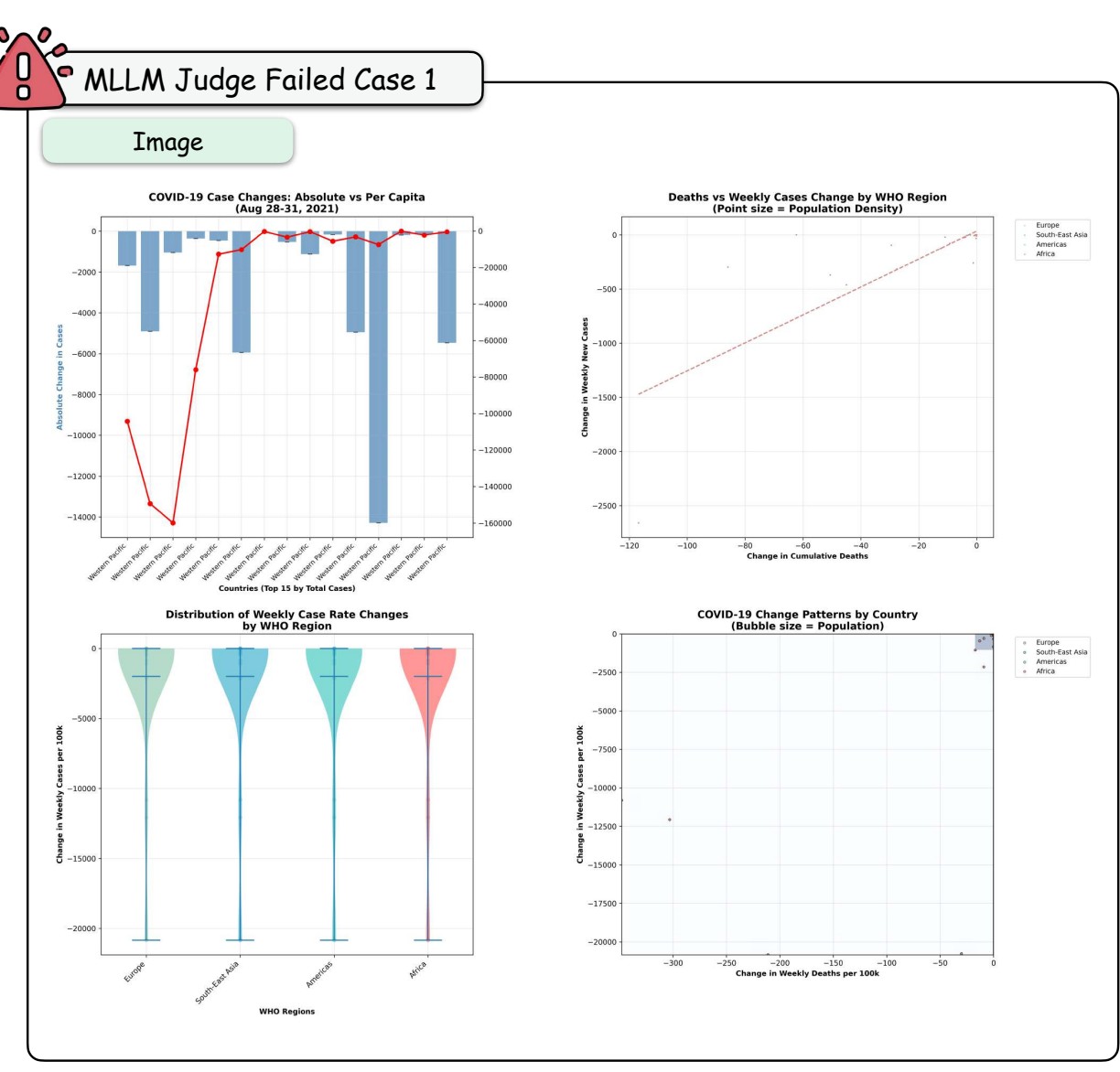

*Figure 34.* MLLM visual judge failed case 1

*Table 15.* **Quantitative results on PlotCraft across three difficulty settings: Hard, Medium, and Easy.** The presented scores are the average results over 5 independent evals. Task-Comp. and Quality denote the total scores for the Task Compliance (out of a maximum of 4) and Chart Quality (out of a maximum of 10) sub-metrics, respectively. The metrics are presented separately for Single-Turn Generation and Multi-Turn Refinement.

| Model | Single-Turn Generation | | | Multi-Turn Refinement | | |
|---|---|---|---|---|---|---|
| | Pass Rate (%) | Task-Comp. | Quality | Pass Rate (%) | Task-Comp. | Quality |
| **HARD** | | | | | | |
| *Closed-source LLMs* | | | | | | |
| Claude-4.1-Opus (Anthropic, 2023) | 64.85 | $0.79 \pm 0.02$ | $2.59 \pm 0.01$ | 68.26 | $0.89 \pm 0.01$ | $2.73 \pm 0.02$ |
| Claude-4-Sonnet (Anthropic, 2023) | 46.75 | $0.59 \pm 0.01$ | $1.71 \pm 0.03$ | 57.40 | $0.65 \pm 0.04$ | $2.17 \pm 0.02$ |
| Gemini-2.5-Pro (Team, 2024) | 5.33 | $0.07 \pm 0.09$ | $0.17 \pm 0.08$ | 28.99 | $0.29 \pm 0.02$ | $1.30 \pm 0.11$ |
| ChatGPT-4o-Latest (OpenAI, 2023) | 34.32 | $0.31 \pm 0.01$ | $1.12 \pm 0.03$ | 47.34 | $0.37 \pm 0.05$ | $1.86 \pm 0.02$ |
| GPT-5 (OpenAI, 2025) | 44.97 | $0.98 \pm 0.01$ | $1.12 \pm 0.01$ | 53.85 | $0.98 \pm 0.02$ | $2.03 \pm 0.01$ |
| *Open-source LLMs* | | | | | | |
| Kimi-K2 (Team et al., 2025b) | 29.70 | $0.28 \pm 0.01$ | $0.82 \pm 0.06$ | 32.93 | $0.40 \pm 0.01$ | $1.37 \pm 0.01$ |
| DeepSeek-V3.1 (DeepSeek-AI & etc., 2024) | 11.83 | $0.14 \pm 0.02$ | $0.40 \pm 0.00$ | 10.06 | $0.11 \pm 0.03$ | $0.43 \pm 0.05$ |
| GLM-4.5 (Team et al., 2025a) | 15.98 | $0.18 \pm 0.02$ | $0.44 \pm 0.04$ | 43.79 | $0.51 \pm 0.01$ | $1.64 \pm 0.01$ |
| GPT-oss-120B (Agarwal et al., 2025) | 12.12 | $0.12 \pm 0.01$ | $0.32 \pm 0.02$ | 13.17 | $0.13 \pm 0.03$ | $0.42 \pm 0.04$ |
| GPT-oss-20B (Agarwal et al., 2025) | 8.48 | $0.10 \pm 0.01$ | $0.21 \pm 0.02$ | 12.57 | $0.11 \pm 0.01$ | $0.31 \pm 0.03$ |
| Seed-Coder-8B (Seed et al., 2025) | 3.55 | $0.02 \pm 0.00$ | $0.08 \pm 0.02$ | 37.28 | $0.30 \pm 0.01$ | $1.22 \pm 0.01$ |
| VisCoder-7B (Ni et al., 2025) | 5.33 | $0.02 \pm 0.00$ | $0.13 \pm 0.00$ | 50.30 | $0.38 \pm 0.01$ | $2.04 \pm 0.01$ |
| Qwen3-Coder-480B-A35B (Hui et al., 2024) | 39.64 | $0.48 \pm 0.02$ | $1.17 \pm 0.06$ | 56.21 | $0.63 \pm 0.01$ | $1.98 \pm 0.07$ |
| Qwen3-Coder-30B-A3B (Hui et al., 2024) | 19.53 | $0.18 \pm 0.01$ | $0.47 \pm 0.02$ | 56.80 | $0.50 \pm 0.01$ | $1.76 \pm 0.06$ |
| **PlotCraftor-30B-A3B** (Ours) | 36.09 | $0.43 \pm 0.01$ | $1.45 \pm 0.01$ | 54.44 | $0.63 \pm 0.01$ | $2.11 \pm 0.01$ |
| **MEDIUM** | | | | | | |
| *Closed-source LLMs* | | | | | | |
| Claude-4.1-Opus (Anthropic, 2023) | 72.33 | $1.97 \pm 0.02$ | $4.01 \pm 0.02$ | 82.61 | $2.06 \pm 0.02$ | $5.20 \pm 0.02$ |
| Claude-4-Sonnet (Anthropic, 2023) | 72.39 | $1.58 \pm 0.02$ | $3.39 \pm 0.02$ | 85.89 | $1.96 \pm 0.02$ | $4.74 \pm 0.02$ |
| Gemini-2.5-Pro (Team, 2024) | 39.26 | $1.04 \pm 0.01$ | $1.77 \pm 0.01$ | 67.48 | $1.74 \pm 0.02$ | $4.07 \pm 0.02$ |
| ChatGPT-4o-Latest (OpenAI, 2023) | 65.64 | $1.48 \pm 0.01$ | $2.96 \pm 0.02$ | 74.85 | $1.48 \pm 0.02$ | $4.10 \pm 0.02$ |
| GPT-5 (OpenAI, 2025) | 77.30 | $2.15 \pm 0.02$ | $3.29 \pm 0.02$ | 80.37 | $2.04 \pm 0.02$ | $4.76 \pm 0.02$ |
| *Open-source LLMs* | | | | | | |
| Kimi-K2 (Team et al., 2025b) | 62.89 | $1.38 \pm 0.01$ | $2.99 \pm 0.02$ | 68.94 | $1.47 \pm 0.02$ | $4.05 \pm 0.02$ |
| DeepSeek-V3.1 (DeepSeek-AI & etc., 2024) | 53.99 | $1.15 \pm 0.01$ | $2.54 \pm 0.01$ | 42.33 | $1.01 \pm 0.01$ | $2.50 \pm 0.01$ |
| GLM-4.5 (Team et al., 2025a) | 44.17 | $1.20 \pm 0.01$ | $1.96 \pm 0.01$ | 68.10 | $1.50 \pm 0.02$ | $3.66 \pm 0.02$ |
| GPT-oss-120B (Agarwal et al., 2025) | 51.57 | $1.14 \pm 0.01$ | $2.13 \pm 0.01$ | 32.30 | $0.76 \pm 0.01$ | $1.69 \pm 0.01$ |
| GPT-oss-20B (Agarwal et al., 2025) | 45.91 | $1.08 \pm 0.01$ | $1.94 \pm 0.01$ | 36.65 | $0.93 \pm 0.01$ | $1.99 \pm 0.01$ |
| Seed-Coder-8B (Seed et al., 2025) | 28.83 | $0.60 \pm 0.01$ | $1.14 \pm 0.01$ | 50.31 | $0.99 \pm 0.01$ | $2.50 \pm 0.01$ |
| VisCoder-7B (Ni et al., 2025) | 11.04 | $0.17 \pm 0.00$ | $0.47 \pm 0.00$ | 39.88 | $0.66 \pm 0.01$ | $1.92 \pm 0.01$ |
| Qwen3-Coder-480B-A35B (Hui et al., 2024) | 60.12 | $1.35 \pm 0.01$ | $2.63 \pm 0.01$ | 82.82 | $1.66 \pm 0.02$ | $4.31 \pm 0.02$ |
| Qwen3-Coder-30B-A3B (Hui et al., 2024) | 53.37 | $1.07 \pm 0.01$ | $2.27 \pm 0.01$ | 74.85 | $1.48 \pm 0.02$ | $3.71 \pm 0.02$ |
| **PlotCraftor-30B-A3B** (Ours) | 68.10 | $1.72 \pm 0.02$ | $3.54 \pm 0.02$ | 83.80 | $1.79 \pm 0.02$ | $4.44 \pm 0.02$ |
| **EASY** | | | | | | |
| *Closed-source LLMs* | | | | | | |
| Claude-4.1-Opus (Anthropic, 2023) | 92.26 | $3.08 \pm 0.03$ | $6.10 \pm 0.03$ | 94.27 | $3.29 \pm 0.03$ | $7.89 \pm 0.03$ |
| Claude-4-Sonnet (Anthropic, 2023) | 88.68 | $3.09 \pm 0.03$ | $7.01 \pm 0.03$ | 93.08 | $3.11 \pm 0.03$ | $7.67 \pm 0.03$ |
| Gemini-2.5-Pro (Team, 2024) | 81.76 | $2.43 \pm 0.02$ | $5.16 \pm 0.02$ | 81.76 | $2.58 \pm 0.02$ | $6.18 \pm 0.03$ |
| ChatGPT-4o-Latest (OpenAI, 2023) | 92.45 | $3.08 \pm 0.03$ | $6.06 \pm 0.03$ | 86.79 | $2.73 \pm 0.02$ | $6.88 \pm 0.03$ |
| GPT-5 (OpenAI, 2025) | 88.68 | $2.20 \pm 0.02$ | $4.30 \pm 0.02$ | 89.31 | $2.43 \pm 0.02$ | $6.34 \pm 0.03$ |
| *Open-source LLMs* | | | | | | |
| Kimi-K2 (Team et al., 2025b) | 89.68 | $2.99 \pm 0.03$ | $6.46 \pm 0.03$ | 82.80 | $2.66 \pm 0.02$ | $6.92 \pm 0.03$ |
| DeepSeek-V3.1 (DeepSeek-AI & etc., 2024) | 88.68 | $2.97 \pm 0.03$ | $6.25 \pm 0.03$ | 59.75 | $1.96 \pm 0.02$ | $4.69 \pm 0.02$ |
| GLM-4.5 (Team et al., 2025a) | 71.70 | $2.44 \pm 0.02$ | $5.20 \pm 0.02$ | 84.28 | $2.67 \pm 0.02$ | $6.60 \pm 0.03$ |
| GPT-oss-120B (Agarwal et al., 2025) | 83.23 | $2.79 \pm 0.02$ | $5.77 \pm 0.02$ | 44.59 | $1.46 \pm 0.01$ | $3.59 \pm 0.01$ |
| GPT-oss-20B (Agarwal et al., 2025) | 81.94 | $2.55 \pm 0.02$ | $5.30 \pm 0.02$ | 54.14 | $1.68 \pm 0.02$ | $4.25 \pm 0.02$ |
| Seed-Coder-8B (Seed et al., 2025) | 66.67 | $2.04 \pm 0.02$ | $4.13 \pm 0.02$ | 84.91 | $2.57 \pm 0.02$ | $6.09 \pm 0.03$ |
| VisCoder-7B (Ni et al., 2025) | 61.64 | $1.86 \pm 0.02$ | $4.02 \pm 0.02$ | 65.41 | $2.00 \pm 0.02$ | $5.02 \pm 0.02$ |
| Qwen3-Coder-480B-A35B (Hui et al., 2024) | 85.53 | $2.92 \pm 0.03$ | $5.98 \pm 0.03$ | 89.94 | $3.03 \pm 0.03$ | $7.24 \pm 0.03$ |
| Qwen3-Coder-30B-A3B (Hui et al., 2024) | 86.79 | $2.77 \pm 0.02$ | $5.99 \pm 0.03$ | 88.68 | $2.74 \pm 0.02$ | $7.23 \pm 0.03$ |
| **PlotCraftor-30B-A3B** (Ours) | 90.57 | $3.13 \pm 0.03$ | $7.45 \pm 0.03$ | 94.34 | $2.94 \pm 0.03$ | $7.85 \pm 0.03$ |

