# OpenReview forum: "PlotCraft: Pushing the Limits of LLMs for Complex and Interactive Data Visualization"
_ICML.cc/2026/Conference — ICML 2026 regular_

### Official Review · Reviewer_sKAe · 2026-03-04

**Soundness:** 2
**Presentation:** 3
**Significance:** 2
**Originality:** 2
**Overall Recommendation:** 3
**Confidence:** 3

**Summary:**

This paper introduces PlotCraft, a benchmark of 982 instances for evaluating LLMs on complex data visualization code generation, and PlotCraftor, a fine-tuned model trained on a synthetic dataset (SynthVis-30K) to improve performance on this task.
The benchmark is built on real-world Kaggle datasets and spans 48 chart types, 31 thematic topics, and 7 high-level visualization intents. Tasks are stratified into Easy (single chart), Medium (composite chart or simple subplot grid), and Hard (complex subplot grid with composite sub-charts), and the benchmark includes both single-turn generation and multi-turn refinement settings.
The evaluation framework uses Gemini-2.5-Pro as an automated visual judge, scoring outputs on Task Compliance (4 binary sub-metrics) and Chart Quality (5 sub-metrics on a 3-point scale).
An evaluation of various LLMs reveals performance degradation on complex tasks.
To address this, the authors synthesize SynthVis-30K via a multi-agent pipeline (Task Generator, Task Judge, Planner, Code Generator, Executor, Visual Judge) and use it to SFT Qwen3-Coder-30B-A3B into PlotCraftor, which achieves a 25% improvement on PlotCraft and competitive results on VisEval and PandasPlotBench.

**Compliance With Llm Reviewing Policy:**

Affirmed.

**Key Questions For Authors:**

Q1. The title prominently features ''nteractive Data Visualization'', and Appendix A.4 lists Plotly as a supported library ''for interactive charts''. However, all 491 reference solutions use Matplotlib, the evaluation pipeline checks for rendered images (not HTML), and no evaluation metric assesses interactivity features. Can the authors clarify: (a) what percentage of benchmark tasks, if any, expect the model to generate Plotly-based interactive charts? (b) If the answer is zero or near-zero, does ''Interactive'' in the title refer instead to the multi-turn refinement process? If so, the authors should acknowledge that this is a non-standard use of the term in the visualization community, and consider whether the title risks misleading readers. A clear answer here would help me reconsider my assessment in W2.

Q2. PlotCraftor clearly improves on its base model, but the improvement appears concentrated on Easy/Medium tasks. To strengthen the solution-side contribution, could the authors provide: (a) an ablation comparing the multi-agent synthesis pipeline against a simpler baseline (e.g., single-pass generation by Claude-4) to justify the pipeline complexity; and (b) a per-difficulty breakdown of PlotCraftor's gains on PlotCraft, specifically quantifying whether the Hard-task improvement is statistically meaningful given the Section 5.2 finding that SFT provides minimal benefit for Hard tasks?

Q3. The human correlation study (Table 4) covers only PlotCraftor's outputs. Have the authors verified that the Gemini-2.5-Pro judge's reliability generalizes to other models? Models with qualitatively different failure modes (e.g., GPT-oss producing garbled layouts vs. Claude producing aesthetically polished but instruction-noncompliant charts) may expose different biases in the automated judge. Cross-model validation would strengthen confidence in the benchmark's evaluation framework.

**Limitations:**

The paper's conclusion briefly acknowledges two limitations: the cost of multi-agent data synthesis and the fallibility of MLLM-based judges for subtle visual errors. However, several more significant limitations are not discussed.

First, the gap between the title's ''Interactive'' framing and the benchmark's actual scope (all reference code and evaluation target static Matplotlib images, despite Plotly being available in the sandbox) should be explicitly acknowledged.

Second, the paper should candidly discuss the tension between its problem diagnosis (LLMs fail on Hard tasks) and its solution's acknowledged ineffectiveness for that very tier.

Third, the benchmark is restricted to Matplotlib for all reference implementations and evaluation despite the sandbox nominally supporting other libraries, which limits applicability.

Finally, there are multiple grammatical errors throughout (''develope'' appears repeatedly, ''constructe'', capitalization inconsistencies), suggesting the manuscript would benefit from careful proofreading.

**Strengths And Weaknesses:**

S1. The benchmark addresses a genuine gap. Most existing Text2Vis benchmarks (VisEval, PandasPlotBench, MatPlotBench) focus on single-panel chart generation, leaving multi-subplot, composite-layout visualization largely unevaluated. PlotCraft's explicit difficulty stratification (Easy/Medium/Hard by subplot complexity), inclusion of multi-turn refinement, and comparison table (Table 1) clearly position it relative to prior work along underexplored dimensions.

S2. The overall execution quality is solid. The benchmark curation is rigorous: tasks are grounded in real Kaggle datasets (462M data rows across 1,874 files), instructions are human-authored, reference code is written by five senior developers, and data leakage between the benchmark and training set is carefully prevented. The experimental coverage is broad (23 models across 3 benchmarks), and the Cohen's Kappa human correlation study (Table 4, 500 charts, 3 annotators) provides more validation than most benchmark papers offer.


W1. The task innovation is incremental, not fundamental. This is my primary concern. The paper extends the Text2Vis paradigm from single-chart to multi-subplot generation, but this is a quantitative scaling of task complexity (more subplots, longer instructions, larger data) rather than a qualitative shift in what is being evaluated. The difficulty taxonomy is defined almost entirely by spatial complexity (number and arrangement of subplots). Deeper dimensions of visualization difficulty -- choosing appropriate chart types for a given analytical question, handling data quality issues, reasoning about visual encodings, or designing for specific audiences -- are not systematically varied or evaluated. The benchmark ultimately measures whether a model can follow increasingly detailed layout instructions, which is closer to instruction-following fidelity than to visualization reasoning.

W2. The title claims ''Interactive Data Visualization'', but there is a substantial gap between this framing and the benchmark's actual content. In the data visualization community, ''interactive visualization'' has an established meaning: user-facing interactivity such as tooltips, zoom, pan, filtering, linked views, or dynamic data exploration (typically implemented via Plotly, Bokeh, D3.js, etc.). While the sandbox environment does include Plotly and describes it as supporting ''interactive charts'', none of the 982 tasks actually evaluate interactive chart generation: all reference solutions use Matplotlib, the evaluation requires outputs to ''produce a valid, non-empty image'' (Section 3.4), and all quality metrics assess static visual properties. The most plausible interpretation is that ''Interactive'' refers to the multi-turn refinement process, but this conflates the interaction modality of the generation process with interactivity in the generated artifact -- distinct concepts in the visualization literature.

W3. The proposed solution (SynthVis-30K + PlotCraftor) works, but its methodological novelty and the scope of its effectiveness are both limited. Two issues constrain the contribution. First, the multi-agent data synthesis pipeline (generate -> judge -> plan -> code -> debug -> refine -> synthesize CoT with Claude) and the subsequent standard SFT (3 epochs, Megatron-LM, no DPO/RLHF/curriculum learning) closely follow well-established recipes from prior work (e.g., Magicoder's OSS-Instruct, WizardCoder's Evol-Instruct). The approach is a competent application of known techniques to a new domain, but does not introduce ideas that the community can learn from or build upon. No ablation is provided to demonstrate that the multi-agent pipeline complexity outperforms simpler baselines (e.g., single-pass generation by Claude-4). Second, the improvement is concentrated on Easy/Medium tasks; the paper's own analysis in Section 5.2 acknowledges that ''SFT provides minimal benefit for Hard tasks'' and that complex visualization ''may rely more on emergent reasoning abilities that come with scale than on task-specific fine-tuning.'' This means the solution is least effective precisely where the benchmark claims existing LLMs most urgently need improvement, creating a tension in the paper's overall narrative.

W4. The automated evaluation has notable reliability gaps. The Cohen's Kappa scores for Chart Quality sub-metrics are moderate at best: Color Quality (0.69) and Text Readability (0.65) fall below the 0.70 threshold commonly considered ''substantial agreement.'' For a benchmark that evaluates chart quality as a core dimension, ~35% disagreement with humans on text readability is a meaningful limitation. Furthermore, the human correlation study covers only PlotCraftor's outputs (500 charts from a single model), leaving open whether judge reliability generalizes across models with different failure modes.

---

> ### Author Rebuttal · Authors · 2026-03-30
>
> We sincerely thank the reviewer for the constructive feedback and for recognizing the rigor of our benchmark curation. Below, we address your concerns point by point.
>
> ### [W1] On novelty and the dimension of task complexity
> While high-level visual reasoning (e.g., open-ended chart selection) is important, precisely executing complex spatial and semantic constraints is far more than an "incremental" step. It represents the fundamental bottleneck for real-world visualization agents.
>
> Exploratory reasoning is moot if models cannot reliably translate analytical intent into coordinated code. As detailed in **Section 2.2**, real-world figures require insight-driven coordination across multiple subplots to form a holistic narrative, going far beyond isolated charts. This involves shared scales, aligned aesthetics, and strict semantic coherence.
> As our evaluation reveals, frontier LLMs already fail catastrophically at this foundational execution step, evidenced by the severe performance drop on Hard tasks. Before the community can evaluate open-ended design reasoning, a capability that is notoriously subjective and difficult to assess reproducibly at scale, models must first conquer complex compositional generation. PlotCraft establishes this critical, unmet prerequisite.
>
> ### [W2 & Q1] On the definition of "Interactive Data Visualization"
> We thank the reviewer for catching this terminology mismatch. You are entirely correct. Our use of the word "Interactive" was intended to describe the multi-turn conversational refinement process between the user and the LLM, rather than the interactivity of the generated final artifacts (which are static Matplotlib images).
>
> We acknowledge that this conflicts with the standard terminology in the visualization literature and risks confusing readers. To resolve this, we will revise the paper's title to "Iterative Data Visualization" and update the terminology throughout the manuscript to explicitly refer to "multi-turn refinement" or "conversational visualization" rather than "interactive charts."
>
> ### [W3 & Q2] On SFT analysis, performance gains, and ablation
> We respectfully clarify a misunderstanding regarding the text: **Section 5.2 belongs to our Related Works**, discussing the general limitations of SFT observed in prior literature, not our own SFT analysis. Our experimental analysis is located in **Section 4.2**.
> Regarding performance on Hard tasks: As shown in **Figure 5** and **Appendix J (Fig. 28, 29)**, performance is strongly correlated with model scale. Hard tasks frequently require generating up to 1,000 lines of highly coordinated code. Given that PlotCraftor has only 30B parameters, it is expected to trail behind frontier closed-source models on the most complex tasks. However, PlotCraftor still achieves a solid ~40% success rate on Hard tasks, representing a significant absolute gain over its base model and proving the effectiveness of the SFT data.
>
> To address your request for an ablation **baseline** (Claude-4 single-pass generation), we provide the comparison below, demonstrating the necessity of our pipeline:
>
> | Difficulty | Model | Single-Turn Pass Rate (%) | Single-Turn Task-Comp. | Single-Turn Quality | Multi-Turn Pass Rate (%) | Multi-Turn Task-Comp. | Multi-Turn Quality | AVG score |
> | :--- | :--- | :--- | :--- | :--- | :--- | :--- | :--- | :--- |
> | Easy | baseline | 88.8| 3.07| 6.98|92.7 |2.83 | 7.52| 10.20|
> | Easy | PlotCraftor (Ours) | 90.6|3.13 |7.45 | 94.3| 2.94| 7.85|10.69 |
> | Medium | baseline |60.3 |1.37 | 2.96| 78.9|1.61 |4.01 |4.98 |
> | Medium | PlotCraftor  | 68.1| 1.72|3.54 |83.8 |1.79 |4.43 |5.74 |
> | Hard | baseline | 25.5| 0.21| 0.51|53.7 | 0.52| 1.85| 1.18|
> | Hard | PlotCraftor  | 38.1| 0.43|1.45 |54.4 | 0.63| 2.11| 2.00|
>
> These results demonstrate the effectiveness of our multi-agent framework, with the relative performance improvements on Hard tasks being noticeably larger than those on Easy and Medium tasks.
>
> ### [W4 & Q3] On the reliability and generalizability of the Visual Judge
> We wish to clarify that we did not use a single Gemini-2.5-Pro judge. As described in **Section 4.3**, our evaluation framework employs a robust multi-judge ensemble (incorporating GPT-4o, Gemini-2.5-Pro, and Gemini-2.5-Flash).
>
> To address the concern about generalizability across different models with varying failure modes, we kindly direct you to **Appendices F and G (Tables 9–14)**. These tables provide a comprehensive breakdown of the scores, human consistency, and internal consistency across multiple different judge models evaluating outputs from various LLMs, confirming that the ensemble’s reliability generalizes well.
> Regarding the Cohen’s Kappa scores for Color Quality (0.69) and Text Readability (0.65): because these are strict fine-grained metrics evaluated on a 0-1-2 scale, an agreement approaching 0.70 is actually quite strong for aesthetic judgments, and our overall task compliance consistency remains exceptionally high.

---

> > ### Author Rebuttal · Reviewer_sKAe · 2026-04-03
> >
> > Thank you for the rebuttal. After careful consideration of the authors' responses, I would like to maintain my current score and evaluation for the following reasons:
> >
> > 1. About Methodological Innovation: While the proposed solution (SynthVis-30K + PlotCraftor) is technically sound, its novelty is constrained by the current research climate. We are seeing a proliferation of work following the exact same paradigm: generating a synthetic dataset to fine-tune an LLM for domain-specific performance gains. While the proposed pipeline is practical and clearly effective, these results are largely expected given that this technical route has already been widely validated across various domains in LLM industry. Consequently, the work represents an incremental contribution that does not sufficiently advance the state-of-the-art in terms of model architecture or fundamental learning methodology.
> >
> > 2. Insufficient Literature Coverage: NL + Data visualization generation is a mature field across the Database, Data Mining,  and Visualization communities. This paper fails to adequately position itself against—or explore—the existing body of research. For instance, in the specific sub-field of NL-to-Vis/Dashboard generation, there is a volume of relevant work, such as [1], that has not been sufficiently addressed.
> >
> > Given these concerns regarding novelty and the lack of a comprehensive literature survey, I remain unconvinced that the paper meets the threshold for a higher score.
> >
> > [1] NL2Dashboard: A Lightweight and Controllable Framework for Generating Dashboards with LLMs.

---

> > > ### Author Response · Authors · 2026-04-04
> > >
> > > Thank you for your feedback. We would like to clarify our contributions and address your concerns regarding the literature:
> > >
> > > 1. Methodological Innovation: While we acknowledge that "synthetic data + SFT" is a widely adopted paradigm, we respectfully emphasize that our core contribution is the PlotCraft benchmark itself. PlotCraft is the first to systematically expose and quantify the severe deficiencies of frontier LLMs on strictly constrained, complex multi-subplot tasks. The SFT pipeline serves primarily as a strong baseline to demonstrate the problem's solvability. The open-sourced PlotCraft and SynthVis-30K are intended as foundations for the community to explore advanced solutions (e.g., RL or architectural innovations) for Hard tasks. Furthermore, our synthesis pipeline involves novel, non-trivial designs, including a vision-based multi-agent loop, CoT synthesis, and multi-turn trajectory merging, which go beyond standard SFT recipes.
> > >
> > > 2. Literature Coverage: We must respectfully clarify that our omission of NL2Dashboard does not reflect a lack of a comprehensive literature review. It is crucial to note that NL2Dashboard was uploaded to arXiv on January 4, 2026, just days prior to the ICML submission deadline, making it strictly concurrent work under standard conference guidelines. Furthermore, while NL-to-Dashboard generation shares tangential relevance, its approach fundamentally differs from our focus on exact Python code generation for granular chart coordination. Nevertheless, we are more than happy to include a discussion of this concurrent work in our revised Related Work section, highlighting how its methodology complements PlotCraft's contributions.

---

### Official Review · Reviewer_dRBu · 2026-03-07

**Soundness:** 3
**Presentation:** 2
**Significance:** 3
**Originality:** 3
**Overall Recommendation:** 5
**Confidence:** 4

**Summary:**

This paper introduces three main contributions.

1) **PlotCraft**, a benchmark for challenging visualization-generation tasks, with particular emphasis on two aspects that are less well covered in prior work: (a) both single-turn generation and multi-turn refinement of visualization code, and (b) multi-chart or dashboard-style figures, where multiple coordinated sub-visualizations are composed to answer a shared analytical question.

2) **SynthVis-30K**, a synthetic dataset of visualization code intended to support training for these more complex visualization-generation settings.

3) **PlotCraftor**, a Qwen-based model fine-tuned on SynthVis-30K, which the authors show achieves strong performance on PlotCraft relative to its base model and several other strong baselines, as well as modest improvements on VisEval and PandasPlotBench relative to the base model.

**Compliance With Llm Reviewing Policy:**

Affirmed.

**Final Justification:**

My final recommendation is Accept. I view the paper primarily as a useful and well-executed benchmark contribution, supported by a useful synthetic data generation pipeline that shows strong empirical gains over the baseline.

My last concern was on score aggregation and how it relates to the framing the authors presented, but in the final rebuttal authors adopted the normalized weighted-average aggregation strategy that seems to be a better match for their compliance first goals, and verified that this change can slightly affect leaderboard ordering.

My remaining suggestion is mainly presentational: I think the final paper should state the score computation more explicitly and plainly, ideally with a short formula or step-by-step procedure. In particular, the compliance-first gating logic was not very clear to me from the manuscript alone, and making it explicit would help readers understand exactly how the aggregate scores are formed and interpreted.

**Key Questions For Authors:**

1. Evaluation protocol and aggregate score.
    Table 2 reports Task-Comp on a 4-point scale and Quality on a 10-point scale, with AVG aggregating these across single-turn and multi-turn settings. Could the authors clarify the exact aggregation formula, including confirming how failed generations are incorporated?

2. Validity of the evaluation rubric, especially the “professional standards” criterion.
    The rubric appears to encode a fairly specific publication-style aesthetic, for example strongly penalizing non-white backgrounds. How was this rubric developed and validated, and were benchmark prompts designed to elicit this style from models? Did the authors test it on a diverse set of existing high-quality charts outside PlotCraft to verify that it scores professional visualizations?

3. Judge calibration and exact model identities.
    Section 4.3 says PlotCraft uses an ensemble of Gemini-2.5-Pro, Gemini-2.5-Flash, and GPT-4o as judges, while elsewhere the paper uses names such as ChatGPT-4o-Latest and GPT-4o in ways that are not fully clear. Could the authors report the exact model identifiers/endpoints used for both evaluated models and judge models, and provide the human-agreement results for each judge in the ensemble individually?

4. Generalization of gains from SynthVis-30K
    PlotCraftor shows a large improvement over its base model on PlotCraft, while gains on VisEval and PandasPlotBench appear more modest. Could the authors comment on what they believe explains this gap? In particular, to what extent might the benefits of training on SynthVis-30K reflect alignment with the PlotCraft task distribution and evaluation setup, versus broader improvements in general text-to-visualization code generation?

**Limitations:**

**Mostly.** The paper discusses several genuine limitations, including the cost of high-quality synthetic data generation, the difficulty of automated visual judging, and the focus on detailed rather than more open-ended visualization instructions. I think it would also help to more explicitly discuss the stylistic assumptions in the evaluation rubric and the possibility that SynthVis-30K is especially well aligned to PlotCraft-style tasks.

**Strengths And Weaknesses:**

**Soundness**

*Strengths*

- The protocol for data collection and benchmark-item construction is a strength of the paper. The authors articulate clear design principles, including real-world data, newly created tasks and reference solutions to reduce leakage, zero-reference generation, and compositional complexity.

- The paper evaluates PlotCraft across a wide range of models, and the benchmark itself covers a broad range of chart/task types and application domains.

- The authors also make a meaningful effort to validate their evaluation pipeline through human annotation and inter-rater agreement analysis, rather than relying solely on an automated judge without calibration.

- The fine-tuned model trained on SynthVis-30K shows a strong improvement over its base model on PlotCraft, while also not showing obvious regressions on external benchmarks.


*Weaknesses*

1. The evaluation protocol and aggregate score could be specified more clearly. Table 2 reports Task Compliance on a 4-point scale and Quality on a 10-point scale, with AVG aggregating these across single-turn and multi-turn settings. It would help to clarify the exact aggregation formula and confirm how non-functioning code outputs are incorporated. More importantly, since these components are on different scales, some discussion of why they are combined directly (rather than normalized) would help readers interpret the benchmark rankings.

2. Some parts of the evaluation rubric seem to encode a fairly specific aesthetic preference rather than a broadly applicable notion of visualization quality. For example, the “Formatting and Professional Standards” rubric gives a score of 0 to charts with a non-white background. This would penalize outputs using common default themes such as ggplot2 or seaborn styles, which many would still consider professional. More broadly, the rubric appears to favor charts that are ready to publish in a paper manuscript, but that is only one notion of high-quality visualization. The authors could include some discussion on, how these criteria were developed, whether they were validated on diverse existing high-quality charts outside PlotCraft, and whether the benchmark prompts themselves consistently specify this intended style. If the style is part of the task specification, then this is less of a concern; if not, it reduces how informative the benchmark is for model developers beyond alignment to the evaluator.

3. The gains from SynthVis-30K should be interpreted somewhat carefully. Fine-tuning on SynthVis-30K clearly improves performance on PlotCraft, but the gains on VisEval and PandasPlotBench are more modest. This raises the possibility that some of the improvement reflects alignment between the synthetic-data construction pipeline and the PlotCraft task/evaluation setup. This would not reduce the utility of SynthVis-30K as a training resource for PlotCraft-style tasks, but it does affect how broadly one should interpret the resulting model-improvement claims.

4. The claims in Section 4.2 about task difficulty, scaling, and SFT are not especially well supported in their current form. I was not fully clear on what evidence Figure 5 is meant to establish beyond the unsurprising observation that stronger models do better overall. Since only one model family appears to have been fine-tuned in the paper, some of the broader statements about how SFT interacts with task difficulty seem stronger than the evidence shown.

5. [Minor] The chart taxonomy is somewhat loose. The paper presents the benchmark as covering 48 plot types, but some of these appear to be variants of the same underlying chart type or are broader task patterns rather than chart types per se. For example, “Time Series,” “Time Series Plot,” and “Line Chart” seem substantially overlapping; “grouped charts” and “seasonal plot” are not really chart types in the same sense; and “ridgeline plots” and “joy plots” are the same plot type. I do not consider this a major issue, since the utility of the benchmark does not hinge on the exact taxonomy, but I would encourage the authors to tighten this categorization or ground it more explicitly in existing visualization literature.


**Originality:**

*Strengths*

- The benchmark’s focus on multi-chart figures is not well covered in many existing benchmarks for plot generation. More specifically, the paper emphasizes producing multiple coordinated charts that jointly support an analytical goal, which is a meaningful and practically relevant setting.

- The inclusion of multi-turn repair/refinement of visualization code is also practically relevant and strengthens the benchmark’s novelty as an evaluation setting.


*Weaknesses*

1. **The comparison to prior work could better situate the contribution relative to recent related benchmarks.** In particular, it is not very clear why Table 1 includes chart-understanding and chart-to-code benchmarks in the comparison, while omitting some more directly relevant recent work. I think the contribution would be better contextualized if the authors discussed benchmarks such as:

    - **Text2Vis** [https://arxiv.org/pdf/2507.19969](https://arxiv.org/pdf/2507.19969): which also includes multi-chart figure generation

    - **Dial-NVBench** [https://arxiv.org/pdf/2307.16013](https://arxiv.org/pdf/2307.16013):: which includes chart refinement through dialog


**Significance**

*Strengths*

- Covering both initial generation and the ability to refine complex visualizations is important for practical workflows around chart creation.

- Dashboard-like displays and coordinated compositions of charts are generally underrepresented in charting benchmarks.

- The evaluation setting is fairly realistic in that the model is asked to go from data and a specification to producing working code for nontrivial analytical visualization tasks.


**Presentation:**

*Strengths*

- Overall, the paper is clearly written and easy to follow, and the appendix contains substantial supporting detail.


*Weaknesses*

1. The evaluation section in the main paper is too brief for a benchmark paper. Since evaluation is central to the contribution, the main body should more clearly explain the core metrics, aggregate scores, and how results should be interpreted.

2. The judge-analysis section is somewhat unclear/incomplete. Section 4.3 describes an ensemble of Gemini-2.5-Pro, Gemini-2.5-Flash, and GPT-4o, but the appendix breakdown reports agreement for Claude Sonnet and ChatGPT-4o instead. It would be more useful to report the individual agreement numbers for the actual ensemble members. The naming of GPT-4o / ChatGPT-4o / ChatGPT-4o-Latest should also be standardized.

3. Some figures could be improved for readability and space efficiency.

    - Figure 1: The left subplot is hard to read, and the per-group scaling makes cross-model comparisons difficult. A more standard small multiples bar chart or a radar plot would be a more suitable choice

    - Figure 2: For me this was not very informative relative to the space it occupies; concrete failure examples and target improvements would be more useful if the authors feel that the driving use case need more evidence. Else this could be dropped to make space for other content.

    - Figure 3: The radial icicle plot is difficult to read because of the small text.

4. [Minor] The title may be slightly misleading. “Interactive Data Visualization” usually suggests that the resulting charts are themselves interactive, whereas the paper appears to focus more on iterative or dialog-based refinement of visualization code.

---

> ### Author Rebuttal · Authors · 2026-03-30
>
> We sincerely thank you for your thorough review and for recognizing the value of our benchmark, the multi-chart focus, and our evaluation protocol. We address your specific questions below:
>
> ### [W1 & Q1] Evaluation protocol and aggregate score
> We directly sum the compliance (4 points) and quality (10 points) scores because they intuitively represent a single continuum of generation utility. Task compliance acts as the fundamental base score (capturing whether the required elements exist), while the quality metrics provide fine-grained differentiation (capturing how well they are implemented). As detailed in **Section 2.4**, any code failing to execute or produce a valid image immediately receives a total score of 0. This additive 14-point scale effectively reflects the overall usefulness of the output without requiring complex normalization. We will clarify this aggregation formula explicitly in the revised manuscript.
>
> ### [W2 & Q2] Validity of evaluation rubric and aesthetic preference
> We acknowledge your valid concern regarding aesthetic preferences. The penalty for non-white backgrounds stems from our system prompt, which explicitly instructed the models to generate charts in an "academic style" (where white backgrounds are standard). We will include the exact system prompts in the revised version for clarity.
>
> Furthermore, we deeply appreciate your feedback on this metric. To ensure our benchmark is not overly biased by a specific aesthetic, we conducted a re-evaluation excluding the "Formatting and Professional Standards" criterion entirely. The overall model rankings remained identical. We will add this ablation study and discuss both ranking settings in the revised version.
>
> ### [Q3] Judge calibration and exact model identities
> We apologize for the inconsistent naming. All references to GPT-4o in our experiments correspond to the ``gpt-4o-2024-11-20`` version. For Gemini, we utilized the gemini-2.5-pro and gemini-2.5-flash endpoints; since these models only have one canonical release version, there is no ambiguity. We will standardize the nomenclature throughout the paper and include a comprehensive Model Details table in the Appendix.
>
> ### [W3 & Q4] Generalization of gains from SynthVis-30K
> The more modest performance gains on VisEval and PandasPlotBench are primarily due to performance saturation. These benchmarks focus on simpler, single-panel tasks where baseline scores are already exceptionally high. PlotCraftor successfully achieved the open-source State-of-the-Art (SOTA) on both, but the ceiling effect naturally limits the absolute percentage improvement. The substantial gains on PlotCraft demonstrate that SynthVis-30K specifically enhances the complex, multi-chart spatial reasoning that simpler benchmarks do not fully capture.
>
> ### [Originality W1] Comparison to prior work
> We thank the reviewer for highlighting these relevant benchmarks. In the revised manuscript, we will update Table 1 and the Related Work section to explicitly discuss Text2Vis and Dial-NVBench, highlighting PlotCraft's complementary focus on multi-chart compositions and iterative refinement.
>
> ### [Minor W5] Chart taxonomy and minor issues
> We gratefully accept your feedback on the chart taxonomy. We will tighten the categorization and merge overlapping terms (e.g., ridgeline/joy plots) to better align with standard visualization literature. Additionally, we will correct the misleading use of the term "interactive" in the text; To avoid any confusion, we have decided to rename the paper to 'Iterative Data Visualization'; please refer to our detailed response to Reviewer sKAe [W2 & Q1] regarding this modification.

---

> > ### Author Rebuttal · Reviewer_dRBu · 2026-04-02
> >
> > Thanks for your response, these points have largely addressed my concerns and will be helpful as I consider final scores. I remain overall favorable toward this paper.
> >
> > My one remaining question concerns the top-level aggregation, and stems from how benchmark scores tend to get used in the community. While your paper presents scores split out by compliance and quality, if adopted by others, they may often be reported as a single average. On the current 14-point scale, quality contributes 10 points versus compliance's 4, meaning the aggregation heavily favors quality, which seems at odds with your stated goal of making task compliance "the fundamental base score," since a model could score well primarily by excelling on quality metrics alone.
> >
> > Have you tried calculating scores where Task Compliance and Quality are weighted equally (e.g., normalizing each to a 0–1 scale and taking the mean) to see whether this changes any AVG score rankings?
> >
> > I recognize there is an alternative, and valid, interpretation: that each individual sub-component is equally important, making a straightforward mean across all 14 points most appropriate. But this didn't quite match my reading of your intention for task compliance as the foundational element. Could you briefly clarify which interpretation is correct?

---

> > > ### Author Response · Authors · 2026-04-04
> > >
> > > Thank you for the insightful suggestion regarding the aggregation strategy. We completely agree with your rationale.
> > >
> > > Following your advice, we recalculated the rankings using a normalized weighted average of Task Compliance and Quality. Interestingly, this did result in minor ranking shifts (e.g., GPT-5 surpassed Grok-4), demonstrating the value of your suggestion. We will officially adopt this normalized ranking strategy for the aggregate leaderboard in the revised manuscript.
> > >
> > > To clarify our foundational logic: during data construction and evaluation, we strictly enforce a compliance-first gating mechanism (any compliance score below 2 immediately zeroes the total score). While we will update the final AVG calculation as discussed, we believe that explicitly reporting the individual Task Compliance and Quality scores in **Table 2** already provides transparent and unambiguous insights into model capabilities without misleading readers.

---

### Official Review · Reviewer_RaG3 · 2026-03-13

**Soundness:** 3
**Presentation:** 3
**Significance:** 3
**Originality:** 3
**Overall Recommendation:** 4
**Confidence:** 2

**Summary:**

This paper introduces PlotCraft, a benchmark of 982 instances for evaluating LLMs on complex multi-panel visualization code generation from real structured data. The main contributions are the benchmark itself (with composite layouts and multi-turn refinement tasks), an evaluation protocol combining execution gating with a VLM-judge ensemble, a 30K synthetic training set (SynthVis-30K), and a fine-tuned model (PlotCraftor on Qwen3-Coder-30B-A3B). Experiments show PlotCraft is harder than prior text-to-vis benchmarks and that PlotCraftor meaningfully improves over the base model.

**Compliance With Llm Reviewing Policy:**

Affirmed.

**Final Justification:**

The benchmark targets a real gap in text-to-vis evaluation: composite multi-panel layouts and multi-turn refinement. PlotCraftor demonstrates measurable gains over strong open baselines, and the cross-benchmark transfer to VisEval and PandasPlotBench confirms the training data has value beyond PlotCraft itself. The evaluation protocol combining execution gating with a VLM-judge ensemble is well thought out.

My concern was judge overlap: SynthVis-30K is generated with GPT-4o and Gemini as agents/judges, and those same models score PlotCraftor at test time. The follow-up reply points to Appendix G (Table 11), where GPT-4o was held out from data synthesis and still produces consistent scores as an independent evaluator. This partially addresses the concern, though it tests one direction of the overlap (held-out judge) and not the other (whether training on judge-scored data embeds preferences that inflate scores).

Hence, I will keep my recommendation score as 4

**Key Questions For Authors:**

-  What fraction of PlotCraft tasks require interactivity (e.g., Plotly-specific interactive features), and how are those evaluated given the image-based judging?

- What are the statistics of injected errors (types, frequency, severity)? Are refinement prompts mostly “fix layout/overlap” or do they include deeper semantic corrections (wrong aggregation, wrong variables)?

- Since PlotCraft few-shots seed task generation and Claude/Gemini are used in synthesis, what safeguards ensure PlotCraftor isn’t overfitting to judge/style priors? Do you have evidence of gains on *non-PlotCraft* complex multi-subplot tasks beyond VisEval/PandasPlotBench (which are simpler)?

- How are the three judges aggregated (majority vote, averaging scores, tie-breaking)? Are judges shown the instruction only, or also metadata, or code?

**Strengths And Weaknesses:**

**Strengths**
- The benchmark is well-motivated and fills a clear gap: composite/multi-subplot layouts and multi-turn refinement are under-covered by prior text-to-vis benchmarks.
- PlotCraftor shows meaningful gains over a strong open baseline and transfers reasonably to VisEval/PandasPlotBench.

**Weaknesses**

- `Potential evaluation bias from judge overlap` It is not clear to me how the overlap between the judge ensemble and evaluated models is handled. PlotCraft scoring uses GPT-4o and Gemini, while those providers' models are also evaluated in Table 2, and SynthVis-30K generation uses Claude and Gemini as agents/judges. I am curious whether this creates a distillation of judge preferences that could inflate PlotCraftor's measured gains, even if rank stability across individual judges is reported.

---

> ### Author Rebuttal · Authors · 2026-03-30
>
> We sincerely thank Reviewer RaG3 for recognizing the value of our benchmark and for the constructive feedback. Below, we address your specific questions and concerns:
>
> ### [W1, Q3] Potential bias, judge overlap, and safeguards against overfitting
> We acknowledge that using LLMs as both generators and evaluators can introduce preference bias. To ensure PlotCraftor learns robust visualization skills rather than overfitting to specific judge or style priors, we implemented several safeguards:
>
> 1. Objective Criteria: As detailed in Appendix C, our rubrics for compliance and quality are highly granular, grounding the model judge in objective standards rather than subjective preferences.
>
> 2. Human Alignment: We conducted consistency experiments comparing our multi-judge ensemble rankings with human annotators, confirming the stability and generalizability of our evaluation framework.
>
> 3. Ablating Subjective Metrics: We recognize that the "Formatting and Professional Standards" metric carries subjective aesthetic preferences. We recalculated the model rankings excluding this specific metric and found that the overall rank order remained completely unchanged. We will include both scoring versions in the revised manuscript.
>
> 4. Cross-Benchmark Generalization: PlotCraftor demonstrates strong, transferable performance on external benchmarks like VisEval and PandasPlotBench. Because these benchmarks use entirely different evaluation protocols, these gains serve as concrete evidence that the model has not overfitted to PlotCraft's specific judge ensemble.
>
> ### [Q1] Fraction of tasks requiring interactivity
> We acknowledge that our use of the term "interactive" is somewhat misleading. In the context of our paper, it exclusively refers to the multi-turn refinement process (the agent interacting with the environment/user), not user-facing interactive chart features (e.g., Plotly tooltips). We will revise this terminology throughout the paper to prevent any confusion. Please refer to our detailed response to Reviewer sKAe [W2 & Q1] regarding this modification.
>
> ### [Q2] Statistics and severity of injected errors
> Our injected errors are designed to comprehensively evaluate refinement capabilities and are distributed across five categories (each accounting for approximately 20% of the tasks):
>
> 1. Incorrect data usage
>
> 2. Wrong chart type
>
> 3. Incorrect layout/formatting
>
> 4. Default style rendering (aesthetic flaws)
>
> 5. Completely incorrect images
>
> The "completely incorrect image" category represents severe, systemic failures requiring major overhauls. The other four categories represent localized, less severe semantic or aesthetic errors. Concrete examples of these refinement scenarios are provided in **Figures 15, 16, 17, and 18**.
>
> ### [Q4] Judge aggregation methodology
> For both the LLM-based judge ensemble and the human annotators, all final scores are aggregated using a majority vote. Judges are shown the instruction and metadata. This approach effectively mitigates the impact of individual judge variance or anomalies, yielding a more robust, stable, and reliable consensus evaluation.

---

> > ### Author Rebuttal · Reviewer_RaG3 · 2026-04-04
> >
> > Thank you for the response.
> >
> > My concerns on judge overlap and evaluation bias are partially addressed. The cross-benchmark generalization on VisEval and PandasPlotBench is useful since those use independent evaluation protocols. But the rebuttal does not directly test whether training on data scored by GPT-4o/Gemini creates a systematic advantage when those same models evaluate PlotCraftor. A clean test would be to evaluate with a held-out judge not involved in data generation or evaluation. Reviewer dRBu (Q3) and Reviewer sKAe (W4) flagged the same concern. The cross-benchmark gains are real, but as Reviewer dRBu noted (W3), those benchmarks are near saturation, which limits their diagnostic power.
> >
> > Therefore, I would like to keep my overall recommendation at 4

---

> > > ### Author Response · Authors · 2026-04-04
> > >
> > > Thank you for the continued discussion. We fully agree that a clean test with a held-out judge is the best way to prove generalization. In fact, we have already conducted this experiment.
> > >
> > > Specifically, GPT-4o was completely excluded from the SynthVis-30K data synthesis process, yet we evaluated its performance as an independent, held-out evaluator within our multi-judge framework. As detailed in **Appendix G** (**Table 11**), our granular, point-by-point evaluation rubric yields highly consistent scores across different judge LLMs, including the held-out GPT-4o. This evidence directly demonstrates that our evaluation possesses strong generalizability and that PlotCraftor's performance gains are genuine, not a result of overfitting to specific judge preferences.

---

### Official Review · Reviewer_beQS · 2026-03-13

**Soundness:** 4
**Presentation:** 3
**Significance:** 4
**Originality:** 4
**Overall Recommendation:** 5
**Confidence:** 5

**Summary:**

This paper proposes a benchmark that covers evaluation metrics along three dimensions, enabling a wide range of analyses. It also constructs a large-scale dataset, SynthVis-30K, to reduce the performance gap on the benchmark, and further proposes PlotCraftor, a model fine-tuned on that dataset.

**Compliance With Llm Reviewing Policy:**

Affirmed.

**Final Justification:**

The authors sufficiently addressed my main concern about leakage and pipeline bias through clear dataset and model isolation, and the remaining question about real-user error distributions does not materially change my positive assessment. so I maintain my current positive score.

**Key Questions For Authors:**

1. In each multi-turn task instance, what exactly is being updated, and what specific aspects are improved as the turns progress?
2. Why does the paper ask an external model to generate the reasoning process or rationale when constructing the SynthVis-30K training dataset?
3. Does the paper evaluate both PlotCraftor’s single-turn generation ability and its multi-turn refinement ability?
4. Is the Visual Judge a model that has been validated or shown to provide reliable feedback?

**Limitations:**

The discussion would be stronger if it explicitly addressed risks such as automating misleading or manipulative visualizations, and benchmark/style bias in real-world dataset curation.

**Strengths And Weaknesses:**

A key strength of the proposed benchmark is that it covers evaluation metrics across three dimensions, which makes a variety of analyses possible.
It is also a positive aspect that the paper does not stop at introducing the benchmark, but additionally presents the full multi-agent pipeline used to generate the dataset.

That said, if a multi-agent framework is used to construct the dataset, it seems necessary to also evaluate the quality of that framework itself, including potential issues such as data bias.

It is also a strength that the data is constructed in a multi-turn format. However, the paper does not provide a sufficiently detailed description of the refinement process in the multi-turn setting. In particular, further analysis seems necessary on how accurately the constructed revisions reflect real user editing behavior, and whether the error distribution is realistic.

In addition, since PlotCraftor is fine-tuned on SynthVis-30K, and both the benchmark and SynthVis-30K appear to share similar source provenance and similar processing or evaluation pipelines, the resulting performance may be inherently biased. The paper would therefore benefit from additional explanation or analysis to make this point more convincing.

Overall, the problem formulation and the proposed solution seem appropriate: the paper introduces a benchmark that is well aligned with the target problem, and it also builds an agent system for dataset construction and utilization. The paper is generally well written. However, it would be more convincing if it included more concrete explanations and analyses of the issues mentioned above.

---

> ### Author Rebuttal · Authors · 2026-03-30
>
> We sincerely thank you for your positive feedback, particularly your recognition of our benchmark's multi-dimensional metrics and the comprehensiveness of our multi-agent pipeline. We address your specific questions and concerns below.
>
> ### (W1) Addressing Data Bias in the Multi-Agent Framework
> We appreciate your insight regarding the evaluation of the framework and potential data bias. To maximize the mitigation of data bias during dataset construction, we strictly control category balance across 48 distinct chart types. Furthermore, we enforce detailed metric partitioning and rigorous evaluation standards (as detailed in **Appendix C**) to systematically constrain the generation process and ensure dataset quality.
>
> ### (W2, Q1) Multi-turn Refinement Process and Error Distribution
> To realistically evaluate multi-turn refinement capabilities, we designed our injected errors to reflect actual editing behaviors. These errors define exactly what is being updated and improved across turns, and are distributed comprehensively across five categories (each accounting for approximately 20% of the tasks):
>
> 1. Incorrect data usage
>
> 2. Wrong chart type
>
> 3. Incorrect layout/formatting
>
> 4. Default style rendering (aesthetic flaws)
>
> 5. Completely incorrect images.
>
> ### (Q2) Rationale for External CoT Synthesis
> During dataset construction, we compress the multi-turn refinement trajectories into single-turn chart generation tasks. Because the intermediate reasoning steps from a multi-turn trajectory do not naturally fit a single-turn context, we utilize an external model to synthesize a coherent Chain-of-Thought (CoT) tailored specifically for single-turn generation.
>
> ### (Q3) Evaluation of Generation and Refinement Capabilities
> Yes, our evaluation encompasses both single-turn generation and multi-turn refinement. The specific performance results for both capabilities are clearly detailed in **Table 2**.
>
> ### (Q4) Reliability of the Visual Judge
> Yes, our Visual Judge is validated and highly reliable. As described in **Section 4.3**, we utilize a multi-judge ensemble comprising three powerful visual models to ensure robust feedback. The core reliability analysis (**Table 4**) demonstrates an average Cohen’s Kappa score with human annotator exceeding 0.75 (75%), indicating substantial quantitative agreement. Additionally, detailed evaluations of each individual judge's reliability, overall results, and internal consistency analyses are provided in **Appendices F and G (Tables 9–14)**.
>
> We hope this clarifies your questions and thank you again for your constructive review.

---

> > ### Author Rebuttal · Reviewer_beQS · 2026-04-03
> >
> > Thank you for the detailed rebuttal. Your clarifications on the multi-turn refinement categories, the rationale for CoT synthesis, the inclusion of both single-turn generation and multi-turn refinement in the evaluation, and the reliability of the Visual Judge are all helpful and address several of my original questions.
> >
> > That said, one of my main concerns remains only partially addressed: the potential coupling between the benchmark and SynthVis-30K/PlotCraftor. Because the benchmark, the synthetic training set, and the evaluation pipeline appear to share similar source provenance and construction assumptions, I still worry that part of the reported gain may reflect pipeline alignment rather than broader generalization. The response explains category balance and quality control, but it does not yet make fully clear how train/eval leakage, stylistic bias, or construction bias are ruled out.
> >
> > I also appreciate the clarification that the multi-turn edits are distributed across five error categories. However, I still think the paper would be stronger if it explained more explicitly whether these categories and their frequencies were derived from real user editing traces or expert analysis of realistic workflows, rather than mainly from design choices during dataset construction.
> >
> > Overall, I remain positive about the paper, but I would appreciate a brief clarification on how independent the benchmark is from the synthetic training pipeline and how the realism of the multi-turn error distribution was validated.

---

> > > ### Author Response · Authors · 2026-04-04
> > >
> > > Thank you for your constructive follow-up. We would like to clarify two key points:
> > >
> > > 1. Data Isolation and Pipeline Alignment: We implemented strict physical isolation between our datasets. The training and testing sets share no overlapping tasks or input files. Furthermore, as detailed in **Appendix E**, the SynthVis-30K generation relied exclusively on ``Claude-4-Sonnet`` and ``Qwen3-Coder-480B``. Because our evaluation framework utilizes an entirely different set of models (Gemini and GPT-4o), there is zero overlap between the generators and the judges. This design maximizes the mitigation of data leakage, evaluation bias, or style imitation.
> > >
> > > 2. Realism of Error Distribution: We candidly acknowledge that the 5-category error distribution stems from our design choices and empirical observations rather than real user editing logs. However, these categories are not arbitrary; they strictly reflect the most frequent and typical failure modes we observed when extensively testing frontier LLMs. Moving forward, upon open-sourcing the benchmark, we plan to collect real-world user issues to further validate and refine these categories.

---

### Decision · Program_Chairs · 2026-04-30

**Decision:**

Accept (regular)

**Comment:**

Reviewer scores are 5, 5, 4, 3 (beQS 5, dRBu 5 post-rebuttal up from 4, RaG3 4, sKAe 3), giving a majority-accept distribution that supports a weak accept. The main evaluation-bias concern raised by beQS, RaG3, and sKAe was substantively addressed by physical data isolation, disjoint generator-judge model sets, consistent held-out judge scores (Table 11), and cross-benchmark transfer, though as RaG3 noted the cross benchmarks are near saturation with modest margins. The novelty concern from sKAe is noted but not blocking: the benchmark itself is the first to systematically quantify frontier LLM failures on strictly constrained multi-subplot tasks, and the distinction from NL2Dashboard via exact Python code generation for fine-grained editing is substantive. Partially resolved concerns remain around multi-turn error distributions being design-driven rather than real-user-log-driven, and SFT effectiveness being limited on Hard tasks. The camera ready should include real user validations and expand the evaluation section.